# Mixture of Geodesic Factor Analyzers on Riemannian Homogeneous Spaces

**Hengchao Chen** [1]  **Yuanyao Tan** [2][3]  **Chao Huang** [4]  **Hongtu Zhu** [5]  **Qiang Sun** [1][6]

## Abstract

This paper introduces Mixtures of Geodesic Factor Analyzers (MGFA) on Riemannian homogeneous spaces. MGFA uses a geodesic factor model within each mixture component, providing greater expressiveness than mixtures of Riemannian radial distributions and enabling clustering of manifold-valued data with anisotropic subpopulations. We establish root-$n$ consistency for the MGFA maximum likelihood estimator (MLE), thereby filling a theoretical gap for mixtures of Riemannian radial distributions as a special case. We also propose an iterative estimation algorithm and implement it on spheres, shape spaces, and hyperbolic spaces. Numerical experiments show that MGFA substantially outperforms competing methods in well-specified regimes while remaining robust under model misspecification. Finally, case studies on corpus callosum and left hippocampus shape datasets demonstrate MGFA's effectiveness for both 2D contour and 3D shape analysis.

## 1. Introduction

Advances in data acquisition and processing have made manifold-valued data increasingly common across many scientific domains, including shape analysis (Kendall, 1984; Srivastava & Klassen, 2016), medical imaging (Fletcher & Joshi, 2007; Tuzel et al., 2006; You & Park, 2021; Dryden et al., 2009; Zhu et al., 2023), bioinformatics (Mardia & Jupp, 2000; Banerjee et al., 2005), and network analysis (Krioukov et al., 2010). Such data often lie on non-Euclidean spaces, such as spheres, shape spaces, hyperbolic spaces, and symmetric positive definite (SPD) matrix spaces, each equipped with a Riemannian metric. A central chal-

lenge in analyzing manifold-valued data is that the intrinsic nonlinearity of these spaces can invalidate Euclidean methods. This has motivated the development of manifold statistics, which designs inferential tools that respect the underlying geometry (Marron & Dryden, 2021; Srivastava & Klassen, 2016; Dubey et al., 2024; Cornea et al., 2017; Fletcher, 2013; Yuan et al., 2012; Bhattacharya & Dunson, 2010; Fletcher et al., 2004; Bhattacharya & Patrangenaru, 2003; Chen et al., 2023; Banerjee et al., 2005; Said et al., 2018; Lin et al., 2017; 2019; Dai & Müller, 2018).

This paper develops a flexible mixture modeling framework on *Riemannian homogeneous spaces* for manifold-valued data with heterogeneous subpopulations. This class includes Euclidean spaces, spheres, hyperbolic spaces, real and complex projective spaces, tori, symmetric positive definite (SPD) matrix spaces, Grassmann manifolds, and Stiefel manifolds. These manifolds possess useful geometric properties (e.g., completeness, homogeneity, and a positive injectivity radius) that facilitate the construction of Riemannian radial distributions (Chen, 2024). Notable examples include the Riemannian Gaussian (Said et al., 2018; Chakraborty & Vemuri, 2019) and the von Mises–Fisher distribution (Banerjee et al., 2005).

We make four main contributions. First, we introduce a *geodesic factor model* for manifold-valued observations. For $x \in \mathcal{M}$, we assume

$$x = \mathrm{Exp}(\mathrm{Exp}(\alpha, Vz), \epsilon),$$

where $\alpha \in \mathcal{M}$ is a location parameter, $V = (v_1, \ldots, v_q)$ with $v_i \in T_\alpha \mathcal{M}$ is a loading matrix, $z \sim \mathcal{N}(0, I_q)$ is a latent vector, and $\epsilon$ is Riemannian noise. Here $\mathrm{Exp}(a, b)$ maps a tangent vector $b \in T_a \mathcal{M}$ to the manifold. Conditional on $z$, $x$ follows a Riemannian Gaussian $RN(\mathrm{Exp}(\alpha, Vz), \sigma)$, although other Riemannian radial distributions are also possible. This construction captures anisotropic dispersion and spatial dependence through low-dimensional latent factors, generalizing isotropic radial models. We extend this idea to clustering via *Mixtures of Geodesic Factor Analyzers (MGFA)*, which generalize mixtures of factor analyzers (Ghahramani et al., 1996) to Riemannian homogeneous spaces. Unlike standard Riemannian radial distributions, which are inherently isotropic and thus limited in modeling directional variation and latent heterogeneity (Huang et al., 2021), MGFA provides a practical geometry-aware

[1]University of Toronto [2]Florida State University [3]Eli Lilly and Company [4]University of Georgia [5]University of North Carolina at Chapel Hill [6]MBZUAI. Correspondence to: Hengchao Chen <chenhc001@gmail.com>, Qiang Sun <qsunstats@gmail.com>.

*Proceedings of the 43$^{rd}$ International Conference on Machine Learning*, Seoul, South Korea. PMLR 306, 2026. Copyright 2026 by the author(s).

alternative.

Second, we characterize the complexity of MGFA and address theoretical gaps in manifold mixture modeling. Using tools from Riemannian geometry, including Jacobi fields and volume comparison, we derive entropy bounds for MGFA under Riemannian Gaussian noise. Under mild conditions, the MGFA maximum likelihood estimator (MLE) achieves a convergence rate of

$$(Km \max\{q, 1\})^{1/2} n^{-1/2}$$

up to logarithmic factors in Hellinger distance, where $K$ denotes the number of components, $m = \dim(\mathcal{M})$, $q$ is the latent dimension, and $n$ is the sample size. This rate matches the classical Euclidean benchmark (Ghosal & Van Der Vaart, 2001). We further extend our analysis to mixtures of Riemannian radial distributions, including von Mises–Fisher distributions (Banerjee et al., 2005), and establish analogous guarantees. In contrast, existing convergence analyses in manifold learning based on empirical process arguments (Petersen & Müller, 2019; Chen & Müller, 2022; Dubey & Müller, 2019; 2020) typically rely on entropy conditions that are difficult to verify on general manifolds and do not directly apply to mixture models. Even for mixtures of radial distributions, convergence guarantees have been unavailable; our results fills this gap.

Third, we propose an iterative procedure for MGFA estimation and clustering that alternates between (i) updating model parameters given labels and (ii) updating labels given the current parameter estimates. Parameter updates follow ideas similar to principal geodesic analysis, combined with geometric estimators for the dispersion parameter $\sigma$. For label updates, we use Monte Carlo approximations to the likelihood and assign each point to the component with the largest posterior probability. To mitigate poor local optima, we use multiple random initializations and select the fit with the highest likelihood (Banerjee et al., 2005).

Finally, we evaluate MGFA on spheres, shape spaces, and hyperbolic spaces across low- and high-dimensional regimes, multiple classes, and model misspecification. MGFA consistently outperforms competing methods in well-specified settings and remains robust under misspecification. We also study two applications: a corpus callosum shape dataset and a left hippocampus shape dataset from the Alzheimer's Disease Neuroimaging Initiative (ADNI)[1]. These case studies illustrate MGFA's ability to identify

meaningful subgroups in both 2D contour and 3D shape analysis.

## 2. Preliminaries

We briefly review Riemannian homogeneous spaces and the geometric tools used in our analysis. Section 2.1 introduces Riemannian homogeneous spaces and summarizes properties used throughout the paper. Section 2.2 reviews integration on manifolds, including the $L^1$ and Hellinger distances, and points to the volume comparison theorem that underpins our theoretical results.

We assume familiarity with basic Riemannian geometry, with introductory material deferred to the Appendix. Further background is available in standard references (Do Carmo, 1992; Helgason, 1979; Petersen, 2006; Cheeger & Ebin, 1975).

### 2.1. Riemannian homogeneous spaces

We review *Riemannian homogeneous spaces* and their key properties (Helgason, 1979). A Riemannian manifold $(\mathcal{M}, g^{\mathcal{M}})$ is *homogeneous* if its isometry group $\mathrm{Iso}(\mathcal{M})$ acts transitively; that is, for any $x, y \in \mathcal{M}$, there exists an isometry $F \in \mathrm{Iso}(\mathcal{M})$ such that $F(x) = y$.

An important subclass is the family of *Riemannian symmetric spaces*. Informally, a symmetric space admits a geodesic symmetry at every point: for each $x \in \mathcal{M}$, there exists an isometry $s_x$ that fixes $x$ and reverses geodesics through $x$. Equivalently, $s_x$ is an involutive isometry (i.e., $s_x \circ s_x = \mathrm{id}$) whose differential at $x$ equals $-\mathrm{id}$. Homogeneous and symmetric spaces enjoy useful geometric properties; the ones used in our analysis are summarized in Proposition A.8 of Appendix A.3.1.

Beyond their geometric and algebraic structure, homogeneous and symmetric spaces provide natural models for diverse application-driven data types; many examples are given in Appendix A.3.2.

### 2.2. Riemannian geometry

We briefly review integration on manifolds and the distance measures used in our analysis. Let $\mathcal{M}$ be an $m$-dimensional Riemannian manifold, and let dvol denote its Riemannian volume measure. For an integrable function $f : \mathcal{M} \to \mathbb{R}$, we write its integral as $\int_{\mathcal{M}} f \, \mathrm{dvol}$.

We rely on two distances between functions on $\mathcal{M}$: the $L^1$ distance and the Hellinger distance. For integrable functions $f_1, f_2$ on $\mathcal{M}$, the $L^1$ distance is

$$d_1(f_1, f_2) = \int_{\mathcal{M}} |f_1 - f_2| \, \mathrm{dvol}.$$

For density functions $f_1, f_2$ on $\mathcal{M}$, the Hellinger distance

[1]Data used in preparation of this article were obtained from the Alzheimer's Disease Neuroimaging Initiative (ADNI) database (adni.loni.usc.edu). As such, the investigators within the ADNI contributed to the design and implementation of ADNI and/or provided data but did not participate in the analysis or writing of this report. A complete listing of ADNI investigators can be found at https://adni.loni.usc.edu/wp-content/uploads/how_to_apply/ADNI_Acknowledgement_List.pdf.

is defined by

$$d_h(f_1, f_2) = \left( \int_{\mathcal{M}} \left( \sqrt{f_1} - \sqrt{f_2} \right)^2 \mathrm{dvol} \right)^{1/2}.$$

Both distances are used throughout the theoretical analysis.

Evaluating integrals of the form $\int_{\mathcal{M}} f \, \mathrm{dvol}$ is a central task in Riemannian geometry. A key tool is the volume comparison theorem, which allows us to control volume growth and related integrals. For completeness, we state the theorem in Appendix A.2.4.

## 3. Mixtures of geodesic factor analyzers

We introduce MGFA on Riemannian homogeneous spaces. Section 3.1 presents the model, and Section 3.2 describes an efficient iterative procedure for estimation and clustering that alternates between parameter updates and label updates. Implementation details for spheres, shape spaces, and hyperbolic spaces are deferred to Appendix E.

### 3.1. Model

We begin by reviewing Riemannian Gaussian distributions in Section 3.1.1. Section 3.1.2 introduces a geodesic factor model that extends geodesic regression (Fletcher, 2013) by treating the regression coefficients as latent factors. Building on this model, Section 3.1.3 proposes MGFA as a mixture of geodesic factor models. The Appendix extends the construction to general Riemannian radial distributions, including von Mises–Fisher and Riemannian Laplace distributions.

#### 3.1.1. Riemannian Gaussian distributions

Given a location parameter $\alpha \in \mathcal{M}$ and a scale parameter $\sigma > 0$, the Riemannian Gaussian distribution $RN(\alpha, \sigma)$ (Chen, 2024; Said et al., 2017) is defined through the unnormalized density

$$\widetilde{f}(x; \alpha, \sigma) = \exp \left\{ -\frac{d_g^2(x, \alpha)}{2\sigma^2} \right\}, \quad x \in \mathcal{M}. \quad (3.1)$$

Proposition 3.1 shows that $\widetilde{f}$ is integrable on $\mathcal{M}$ and that its normalizing constant

$$Z(\alpha, \sigma) := \int_{\mathcal{M}} \widetilde{f}(x; \alpha, \sigma) \, \mathrm{dvol}(x) \quad (3.2)$$

does not depend on $\alpha$. We therefore write $Z(\sigma)$ and express the normalized density as

$$f(x; \alpha, \sigma) = \frac{1}{Z(\sigma)} \exp \left\{ -\frac{d_g^2(x, \alpha)}{2\sigma^2} \right\}. \quad (3.3)$$

We use two basic properties of this distribution throughout the paper. First, if $\{x_i\}_{i=1}^n \overset{\text{i.i.d.}}{\sim} RN(\alpha, \sigma)$, then the MLE

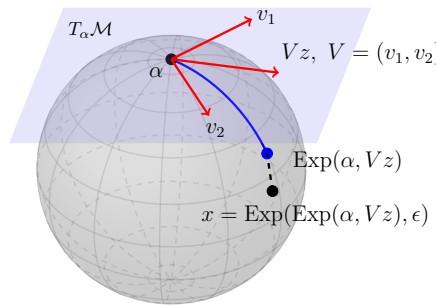

Figure 1. Visualization of the geodesic factor model in (3.5).

of $\alpha$ is the sample Fréchet mean:

$$\widehat{\alpha}^{\mathrm{MLE}} = \underset{a \in \mathcal{M}}{\arg\min} \sum_{i=1}^n d_g^2(x_i, a). \quad (3.4)$$

Second, when $\mathcal{M}$ is a Riemannian symmetric space, the population Fréchet mean under $f(\cdot; \alpha, \sigma)$,

$$\alpha^{\mathrm{FM}} := \underset{a \in \mathcal{M}}{\arg\min} \int_{\mathcal{M}} d_g^2(x, a) \, f(x; \alpha, \sigma) \, \mathrm{dvol}(x),$$

is uniquely attained at $a = \alpha$. Thus, $RN(\alpha, \sigma)$ provides a convenient isotropic location-scale model on homogeneous spaces.

**Proposition 3.1.** *Let $(\mathcal{M}, g^{\mathcal{M}})$ be a Riemannian homogeneous space.*

*(i) The function $\widetilde{f}$ defined in (3.1) is integrable on $\mathcal{M}$.*

*(ii) The normalizing constant $Z(\alpha, \sigma)$ defined in (3.2) is independent of $\alpha$ and thus $Z(\alpha, \sigma) = Z(\sigma)$.*

*(iii) If $\{x_i\}_{i=1}^n \overset{\text{i.i.d.}}{\sim} RN(\alpha, \sigma)$, then the MLE of $\alpha$ is the sample Fréchet mean in (3.4), regardless of whether $\sigma$ is known.*

*(iv) If $\mathcal{M}$ is a Riemannian symmetric space, then the population Fréchet mean under $f(\cdot; \alpha, \sigma)$ is uniquely attained at $\alpha$.*

*(v) Let $(\mathcal{N}, g^{\mathcal{N}})$ be another Riemannian homogeneous space, and equip $\mathcal{M} \times \mathcal{N}$ with the product metric $g^{\mathcal{M}} + g^{\mathcal{N}}$. Then $(x, y) \in \mathcal{M} \times \mathcal{N}$ follows $RN((\alpha, \beta), \sigma)$ if and only if $x \sim RN(\alpha, \sigma)$, $y \sim RN(\beta, \sigma)$, and $x$ and $y$ are independent.*

#### 3.1.2. Geodesic factor models

Riemannian Gaussian distributions provide an isotropic model on $\mathcal{M}$, which can be too restrictive when the data exhibit anisotropic variability. To capture such structure,

we introduce a *geodesic factor model* with low-dimensional latent factors. Specifically, we assume

$$x = \mathrm{Exp}\big(\mathrm{Exp}(\alpha, Vz), \epsilon\big), \qquad \alpha \in \mathcal{M},$$
$$V = (v_1, \ldots, v_q), \quad v_j \in T_\alpha \mathcal{M}, \qquad (3.5)$$

where $\mathrm{Exp}(a, b) := \mathrm{Exp}_a(b)$ is the exponential map, $z \sim \mathcal{N}(0, I_q)$ is a latent vector, and $\epsilon$ is Riemannian noise. Conditional on $z$, the observation follows $x \sim RN(\mathrm{Exp}_\alpha(Vz), \sigma)$. If $z$ were observed, (3.5) would reduce to geodesic regression (Fletcher et al., 2004; Cornea et al., 2017); treating $z$ as latent yields a manifold analogue of factor analysis. In the Euclidean case $\mathcal{M} = \mathbb{R}^m$, (3.5) reduces to the classical factor model

$$x = \alpha + Vz + \epsilon, \quad z \sim \mathcal{N}(0, I_q), \ \epsilon \sim \mathcal{N}(0, \sigma^2 I_m).$$

Model (3.5) is parameterized by $(\alpha, V, \sigma)$, where $V$ is identifiable only up to right orthogonal transformations: $V$ and $VO$ induce the same distribution for any orthogonal matrix $O \in \mathbb{R}^{q \times q}$. We therefore regard $V$ as an equivalence class and, with slight abuse of notation, continue to write it as $V$. Marginalizing over the latent factor $z$ gives the density $f(x; \alpha, V, \sigma)$:

$$\frac{1}{Z(\sigma)} \int_{\mathbb{R}^q} \exp\left\{ -\frac{d_g^2\big(x, \mathrm{Exp}_\alpha(Vz)\big)}{2\sigma^2} \right\} p(z) \, dz \qquad (3.6)$$

for any $x \in \mathcal{M}$, where $p(z) = (2\pi)^{-q/2} \exp\{-\|z\|^2/2\}$ is the density of $\mathcal{N}(0, I_q)$ and $Z(\sigma)$ is the normalizing constant in (3.3). In particular, $Z(\sigma)$ is independent of both $\alpha$ and $V$.

**Proposition 3.2.** *Under model* (3.5)*, the marginal density of $x$ is given by* (3.6)*.*

### 3.1.3. MIXTURES OF GEODESIC FACTOR ANALYZERS

We define MGFA as a finite mixture of geodesic factor models:

$$f(x; \Upsilon) = \sum_{k=1}^K \omega_k \, f(x; \alpha_k, V_k, \sigma_k), \qquad (3.7)$$

where $\omega = (\omega_1, \ldots, \omega_K)$ denotes the mixture weights, with $\omega_k \geq 0$ and $\sum_{k=1}^K \omega_k = 1$. Each component density $f(x; \alpha_k, V_k, \sigma_k)$ is of the form (3.6), with location $\alpha_k \in \mathcal{M}$, loading matrix $V_k = (v_{k1}, \ldots, v_{kq})$ satisfying $v_{kj} \in T_{\alpha_k} \mathcal{M}$, and scale parameter $\sigma_k > 0$. We write $\Upsilon = \{\omega_k, \alpha_k, V_k, \sigma_k\}_{k=1}^K$ for the full parameter set.

### 3.2. Algorithm

We propose an iterative procedure for MGFA estimation and clustering that alternates between updating model parameters and updating cluster labels. Given samples $\{x_i\}_{i=1}^n \subseteq$

---

**Algorithm 1** Mixture of geodesic factor analyzers (MGFA).

**input** data $\{x_i\}_{i=1}^n \subseteq \mathcal{M}$, number of components $K$, latent dimension $q$.
**output** labels $\{y_i\}_{i=1}^n \subseteq [K]$.
1: Randomly initialize labels $\{y_i\}_{i=1}^n$.
2: **for** $t \leftarrow 1$ **to** $T$ **do**
3:     Update parameters $\widehat{\Upsilon} = \{\widehat{\omega}_k, \widehat{\alpha}_k, \widehat{V}_k, \widehat{\sigma}_k\}_{k=1}^K$ given labels $\{y_i\}_{i=1}^n$.     (*parameter update*)
4:     Update labels $\{y_i\}_{i=1}^n$ given $\widehat{\Upsilon}$.     (*label update*)
5: **end for**

---

$\mathcal{M}$, the goal is to fit an MGFA model with $K$ components and $q$ latent factors. Algorithm 1 summarizes the procedure.

This algorithmic formulation also distinguishes MGFA from the mixture-of-PPGA model of Zhang et al. (2019), which extends probabilistic principal geodesic analysis (PPGA) (Zhang & Fletcher, 2013). In that work, the quantity corresponding to the component-specific latent factor in MGFA is introduced as a latent variable but later optimized as a model parameter.[2] This role mismatch obscures the objective being optimized and complicates theoretical analysis. In contrast, MGFA keeps latent factors separate from model parameters, yielding a clearer estimation procedure and supporting the theoretical analysis in Section 4.

#### 3.2.1. PARAMETER UPDATE

Fix a label assignment $\{y_i\}_{i=1}^n \subseteq [K]$, and let $\mathcal{I}_k = \{i : y_i = k\}$ denote the index set for component $k$. We estimate a geodesic factor model (3.5) separately within each component. The parameter update consists of four steps:

(a) **Location.** Estimate $\alpha_k$ by the sample Fréchet mean

$$\widehat{\alpha}_k = \underset{a \in \mathcal{M}}{\mathrm{argmin}} \sum_{i \in \mathcal{I}_k} d_g^2(x_i, a),$$

which can be computed via gradient descent on $\mathcal{M}$ (Fletcher, 2013).

(b) **Tangent projection and PCA.** For each $i \in \mathcal{I}_k$, map $x_i$ to the tangent space by $u_i = \mathrm{Log}_{\widehat{\alpha}_k}(x_i) \in T_{\widehat{\alpha}_k} \mathcal{M}$. Perform PCA on $\{u_i\}_{i \in \mathcal{I}_k}$ to obtain the leading $q$ eigenpairs $\{(\lambda_{kj}, \widehat{e}_{kj})\}_{j=1}^q$ of the empirical covariance $|\mathcal{I}_k|^{-1} \sum_{i \in \mathcal{I}_k} u_i u_i^\top$. Set

$$\widehat{V}_k = \big(\sqrt{\lambda_{k1}}\,\widehat{e}_{k1}, \ldots, \sqrt{\lambda_{kq}}\,\widehat{e}_{kq}\big) \in (T_{\widehat{\alpha}_k} \mathcal{M})^q.$$

(c) **Residuals and scale.** Let $P_k$ denote the orthogonal projection onto $\mathrm{span}(\widehat{V}_k)$, and define the residuals $u_i^\perp = u_i - P_k u_i$. Update $\sigma_k$ by

$$\widehat{\sigma}_k = \underset{\sigma > 0}{\mathrm{argmax}} \left\{ -\log Z(\sigma) - \frac{\mathrm{res}_k}{2\sigma^2} \right\}, \qquad (3.8)$$

---

[2] The corresponding notation in Zhang et al. (2019) is $x_{nk}$.

where $\mathrm{res}_k = |\mathcal{I}_k|^{-1} \sum_{i \in \mathcal{I}_k} \|u_i^\perp\|^2$.

(d) **Weights.** Update the mixture weights using empirical proportions:

$$\widehat{\omega}_k = \frac{|\mathcal{I}_k|}{n}, \qquad k = 1, \dots, K.$$

### 3.2.2. LABEL UPDATE

Given $\widehat{\Upsilon} = \{\widehat{\omega}_k, \widehat{\alpha}_k, \widehat{V}_k, \widehat{\sigma}_k\}_{k=1}^K$, update the labels by maximum a posteriori assignment:

$$y_i = \underset{k \in [K]}{\mathrm{argmax}} \ \widehat{\omega}_k \, f(x_i; \widehat{\alpha}_k, \widehat{V}_k, \widehat{\sigma}_k).$$

Because (3.6) is generally unavailable in closed form, we approximate the integral by Monte Carlo. Draw $B$ i.i.d. samples $\{z_{ij}\}_{j=1}^B \sim \mathcal{N}(0, I_q)$ and use

$$f^{\mathrm{approx}}(x_i; \alpha, V, \sigma) \qquad (3.9)$$
$$= \frac{1}{B \cdot Z(\sigma)} \sum_{j=1}^B \exp \left\{ -\frac{d_g^2(x_i, \mathrm{Exp}_\alpha(V z_{ij}))}{2\sigma^2} \right\}.$$

### 3.3. Implementation

Implementation depends on the geometry of $\mathcal{M}$. For example, updating $\sigma$ in (3.8) requires evaluating $Z(\sigma)$ and its derivative, while the label update in (3.9) also requires $Z(\sigma)$. Appendix E provides explicit expressions and numerical details for spheres, shape spaces, and hyperbolic spaces, together with initialization and model selection strategies.

## 4. Theory

Statistical analysis of manifold-valued models is important but challenging because the underlying geometry affects both likelihood behavior and model complexity. This section combines Riemannian geometry with empirical process theory to derive entropy bounds for MGFA and a non-asymptotic convergence rate for the maximum likelihood estimator (MLE). Specifically, given $n$ independent samples $\{x_i\}_{i=1}^n$ drawn from model (3.7), the MLE $\widehat{\Upsilon}$ of the parameter $\Upsilon$ is defined as

$$\widehat{\Upsilon} = \underset{\Upsilon \in \Xi}{\mathrm{argmax}} \sum_{i=1}^n \log f(x_i; \Upsilon), \qquad (4.1)$$

where $\Xi$ denotes the feasible parameter space. Our analysis establishes a $(Km \max\{q, 1\})^{1/2} n^{-1/2} \log n$ convergence rate in Hellinger distance, $d_h(f(\cdot; \widehat{\Upsilon}), f(\cdot; \Upsilon^*))$, between the estimated and true distributions. Here $\Upsilon^* \in \Xi$ denotes the true parameter set, $K$ is the number of mixture components, $m$ is the manifold dimension, and $q$ is the latent dimension. To our knowledge, this is the first convergence

theory for mixture models on general manifolds and among the first to explicitly characterize their model complexity. The Appendix extends these theoretical results to MGFA models with Riemannian radial distributions, including von Mises–Fisher and Riemannian Laplace distributions as examples.

The main technical challenge is to control the entropy of the MGFA model. This requires both integration over manifolds and bounds on geodesic deviation arising from the geodesic factor structure. We address these issues using tools from Riemannian geometry, particularly the volume comparison theorem and Jacobi field theory. Additional details are provided in Appendix C.

### 4.1. Model complexity

We now quantify the model complexity of MGFA. Let $(\mathcal{M}, g^{\mathcal{M}})$ be a Riemannian homogeneous space. Our first result gives entropy bounds for the function class $\mathcal{F}_\Xi = \{f(x; \Upsilon) \mid \Upsilon \in \Xi\}$, where $f(x; \Upsilon)$ is the MGFA density in (3.7) and $\Upsilon$ ranges over the feasible set $\Xi$. For MGFA, we define the feasible parameter set $\Xi$ as

$$\Xi = \left\{ \Upsilon = \{\omega_k, \alpha_k, V_k, \sigma_k\}_{k=1}^K \ \middle| \ \omega_k \geq 0, \ \sum_k \omega_k = 1, \right.$$
$$\alpha_k \in \mathcal{B}_\mathcal{M}(\alpha^*, D), \ \sigma_k \in [\sigma_{\min}, \sigma_{\max}],$$
$$\left. V_k = (v_{k1}, \dots, v_{kq}), \ \|V_k\|_{\mathrm{F}} \leq A \right\}, \qquad (4.2)$$

where $\mathcal{B}_\mathcal{M}(\alpha^*, D) = \{x \in \mathcal{M} \mid d_g(x, \alpha^*) \leq D\}$ denotes the geodesic ball in $\mathcal{M}$ centered at a fixed reference point $\alpha^* \in \mathcal{M}$ with radius $D$. The constants $\alpha^*, D, \sigma_{\min} > 0, \sigma_{\max}$, and $A$ are fixed. Thus, $\Xi$ imposes three constraints: the location parameters $\{\alpha_k\}$ lie in a bounded region of the manifold; the dispersion parameters $\{\sigma_k\}$ are bounded below and above, with $\sigma_k \in [\sigma_{\min}, \sigma_{\max}]$; and the factor loading matrices $\{V_k\}$ have bounded Frobenius norms. These assumptions ensure finite entropy bounds and are standard even in Euclidean mixture models.

Our first main result, Theorem 4.1, bounds the bracketing entropy of $\mathcal{F}_\Xi$ under both the $L^1$ and Hellinger metrics by $C_* K m \max\{q, 1\} \log(1/\epsilon)$, where $C_*$ is a universal constant. The factor $K m \max\{q, 1\}$ reflects the degrees of freedom in MGFA: each of the $K$ components has a location parameter on an $m$-dimensional manifold and an $mq$-dimensional loading matrix. Thus, the complexity scaling of MGFA on manifolds matches the corresponding Euclidean order.

**Theorem 4.1.** *Let $\mathcal{M}$ be an $m$-dimensional Riemannian homogeneous space, and consider the function class $\mathcal{F}_\Xi = \{f(x; \Upsilon) \mid \Upsilon \in \Xi\}$, where $\Xi$ is the parameter space in (4.2). For all sufficiently small $\epsilon > 0$, the following bracketing*

*entropy bounds hold:*

$$\log \mathcal{N}_B(\epsilon, \mathcal{F}_\Xi, d_1) \leq C_* K m \max\{q, 1\} \log(1/\epsilon),$$
$$\log \mathcal{N}_B(\epsilon, \mathcal{F}_\Xi, d_h) \leq C_* K m \max\{q, 1\} \log(1/\epsilon),$$

*where $d_1$ and $d_h$ denote the $L^1$ and Hellinger distances, respectively, and $C_*$ is a constant independent of $m, K, q, \epsilon$.*

*Proof Sketch.* The full proof of Theorem 4.1 is given in Appendix D.2; here we outline the main ideas and make explicit the notation used in the geometric step. The proof consists of three steps, each addressing a geometric difficulty that does not arise in the Euclidean setting. The first step is to construct finite nets for the parameter space $\Xi$. For the mixture weights, dispersion parameters, and loading coordinates, this follows from standard Euclidean covering arguments. For the location parameters $\alpha_k \in \mathcal{M}$, however, the covering number must be controlled on the manifold. We use the volume comparison theorem, Theorem A.7, to cover the bounded geodesic ball $\mathcal{B}_\mathcal{M}(\alpha^*, D)$ and hence obtain an $\epsilon$-net $\mathcal{S}_\Upsilon$ for $\Xi$ with cardinality of order $\exp\{CKm \max\{q, 1\} \log(1/\epsilon)\}$.

The second step converts this parameter net into a uniform approximation of the density class. Specifically, we bound

$$\max_{\Upsilon \in \Xi} \min_{\Upsilon' \in \mathcal{S}_\Upsilon} d_\infty(f(\cdot; \Upsilon), f(\cdot; \Upsilon')) \qquad (4.3)$$

where $d_\infty(g, h) = \sup_{x \in \mathcal{M}} |g(x) - h(x)|$. The main non-Euclidean issue is the dependence of $f(x; \alpha, V, \sigma)$ on the geodesic factor map $\mathrm{Exp}_\alpha(Vz)$. To measure perturbations in $(\alpha, V)$, write $V_i = (v_{i1}, \ldots, v_{iq})$ with $v_{ij} \in T_{\alpha_i}\mathcal{M}$, and define

$$d_{T\mathcal{M}}^2((\alpha_1, V_1), (\alpha_2, V_2))$$
$$= d_g^2(\alpha_1, \alpha_2) + \sum_{j=1}^q \|v_{1j} - \mathcal{P}_{\alpha_2 \to \alpha_1} v_{2j}\|^2,$$

where $\mathcal{P}_{\alpha_2 \to \alpha_1} : T_{\alpha_2}\mathcal{M} \to T_{\alpha_1}\mathcal{M}$ is parallel transport along a minimizing geodesic; if several minimizing geodesics exist, we choose one that minimizes the displayed quantity. For a single tangent vector $u_i \in T_{\alpha_i}\mathcal{M}$, the same notation denotes the corresponding quantity with the summation omitted. For any nonzero latent factor $z \in \mathbb{R}^q$, let $\xi_z = z/\|z\|$ and $t = \|z\|$, so that $\mathrm{Exp}_{\alpha_i}(V_i z) = \mathrm{Exp}_{\alpha_i}(tV_i \xi_z)$. Applying our geometric Lemma C.2, proved by Jacobi field analysis, to the tangent vectors $u_i = V_i \xi_z$ gives

$$d_g(\mathrm{Exp}_{\alpha_1}(V_1 z), \mathrm{Exp}_{\alpha_2}(V_2 z))$$
$$\leq d_{T\mathcal{M}}((\alpha_1, V_1 \xi_z), (\alpha_2, V_2 \xi_z)) e^{c\|z\|}$$
$$\leq d_{T\mathcal{M}}((\alpha_1, V_1), (\alpha_2, V_2)) e^{c\|z\|},$$

where $c = (1 + \kappa A^2)/2$ and $\kappa$ bounds the absolute sectional curvature. This lemma is the central geometric contribution in the entropy proof: it quantifies how perturbations

of both the base point and the loading directions propagate through the exponential map. Although the factor $e^{c\|z\|}$ is faster than the linear growth in Euclidean spaces, it is integrable against the Gaussian density of $z$. This yields Lipschitz continuity of $f(x; \alpha, V, \sigma)$ with respect to $(\alpha, V)$. A separate use of volume comparison controls the normalizing constant $Z(\sigma) = \int_\mathcal{M} \widetilde{f}(x; \alpha, \sigma) \mathrm{dvol}(x)$ and yields the Lipschitz bound

$$|Z(\sigma_1) - Z(\sigma_2)| \leq \int_\mathcal{M} |\widetilde{f}(x; \alpha, \sigma_1) - \widetilde{f}(x; \alpha, \sigma_2)| \mathrm{dvol}(x)$$
$$\leq C|\sigma_1 - \sigma_2| \qquad (4.4)$$

for some constant $C$, where $\widetilde{f}(x; \alpha, \sigma)$ is the unnormalized Riemannian Gaussian kernel in (3.1).

The third step constructs brackets from the uniform approximation above. We first build an envelope function for $\mathcal{F}_\Xi$ and then, using (4.3), form brackets $\{[l_i, u_i]\}$ such that $\mathcal{F}_\Xi \subseteq \cup_i [l_i, u_i]$. It remains to control the $L^1$ width of each bracket:

$$d_1(u_i, l_i) = \int_\mathcal{M} |u_i(x) - l_i(x)| \mathrm{dvol}(x).$$

This integral is not directly tractable on a general manifold. Passing to polar coordinates as in (A.2) reduces the problem to controlling the manifold volume density, but this density is generally complicated on spaces such as SPD matrix spaces, shape spaces, and Grassmann manifolds. The volume comparison theorem replaces it by the comparison function $\mathrm{sn}_\kappa(r)$ in (A.3), where $\kappa$ is a curvature bound. This reduction yields the required $L^1$ and Hellinger bracket widths and completes the entropy bound. $\square$

Building on Theorem 4.1, we derive the convergence rate of the MLE using empirical process theory. Assume that the true parameter set $\Upsilon^*$ belongs to $\Xi$, let $\widehat{\Upsilon}$ be the MLE defined in (4.1), and write $\widehat{f} = f(x; \widehat{\Upsilon})$. We show that the Hellinger distance $d_h(\widehat{f}, f^*)$ between the estimated density $\widehat{f}$ and the true density $f^* = f(x; \Upsilon^*)$ converges at rate $(Km \max\{q, 1\})^{1/2} n^{-1/2}$ up to logarithmic factors. To our knowledge, this is the first convergence result for mixture models on general manifolds. The root-$n$ dependence on sample size matches the Euclidean case, while the factor $Km \max\{q, 1\}$ captures the model complexity. When $q = 0$, the result yields theoretical guarantees for mixtures of Riemannian Gaussian distributions on Riemannian homogeneous spaces.

**Theorem 4.2.** *Let $\mathcal{M}$ be a Riemannian homogeneous space, and let $\Xi$ be defined by (4.2). Assume that the true density is $f^* = f(x; \Upsilon^*)$ for some $\Upsilon^* \in \Xi$. Given $n$ independent samples $\{x_i\}_{i=1}^n$ drawn from $f^*$, the MLE $\widehat{\Upsilon}$ defined in (4.1) satisfies, for sufficiently large $n$ and with probability at least*

$$1 - ce^{-c\log^2 n},$$

$$d_h(\widehat{f}, f^*) \leq \frac{C(Km\max\{q, 1\})^{1/2}\log n}{n^{1/2}},$$

*where $c, C > 0$ are constants independent of $K, m, q, n$, and $\widehat{f} = f(x; \widehat{\Upsilon})$ is the estimated density.*

These results highlight both the parallels and the differences between Euclidean and manifold-valued mixture models. For Riemannian Gaussian components, the Gaussian tail behavior is strong enough to recover a Euclidean-type root-$n$ rate up to logarithmic factors. For more general Riemannian radial distributions, however, geometric effects become more pronounced. For example, Laplace distributions are well-defined on Euclidean spaces and compact manifolds, but on hyperbolic and other noncompact manifolds they require additional tail conditions. The conditions needed for root-$n$ convergence may therefore depend on both the manifold geometry and the choice of radial distribution. We examine these geometric effects in more detail in the Appendix.

## 5. Synthetic data experiments

We evaluate MGFA through simulations on spheres, shape spaces, and hyperbolic spaces. The experiments cover low- and high-dimensional settings, multiple clusters, and model misspecification. We assess clustering accuracy using the Rand index (RI) (Rand, 1971) and the adjusted Rand index (ARI). Across these settings, MGFA consistently outperforms baseline methods such as distance-based Ward clustering (Ward Jr, 1963) and mixtures of Riemannian Gaussian distributions (MoRN) when the underlying subgroups exhibit latent factor structure. MGFA also remains robust under model misspecification.

Here, we present representative results for spherical, shape, and hyperbolic data clustering. Additional simulation results are deferred to Appendices F, G, and H.

### 5.1. Spherical data

Our first synthetic study considers clustering on the sphere. We compare MGFA with Ward clustering, mixtures of von Mises–Fisher distributions (movMF) (Banerjee et al., 2005; Hornik & Grün, 2014), spherical $k$-means (skmeans) (Hornik et al., 2012), and MoRN. Data are generated from the MGFA model (3.7) on $\mathbb{S}^2$ with two groups and one latent factor: (G1) $\omega_1 = 0.5$, $\alpha_1 = (1, 0, 0)$, $v_1 = (0, 1, 0)$, $\sigma_1 = 0.05$; (G2) $\omega_2 = 0.5$, $\alpha_2 = (-0.6, 0, -0.8)$, $v_2 = (0, 1, 0)$, $\sigma_2 = 0.1$.

In each trial, we draw 500 samples and apply all competing clustering methods. Figure 2 shows that MGFA more accurately identifies the latent factor structure, achieving an

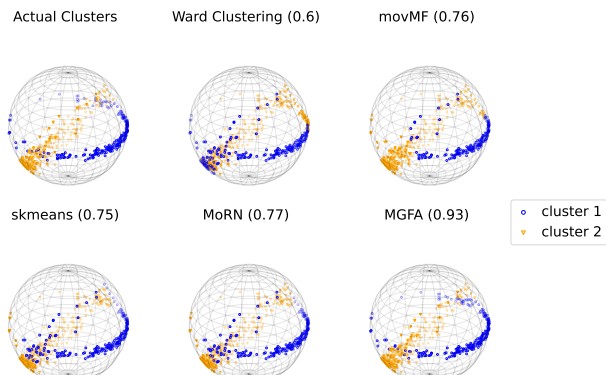

*Figure 2.* Visualization of clustering results on $\mathbb{S}^2$. The first panel shows the true clusters, and the remaining panels show the clustering results produced by different methods. Rand scores are reported in parentheses.

*Table 1.* Mean RIs and ARIs, with standard deviations in parentheses, for Ward clustering, MoRN, and MGFA on $\mathbb{CP}^3$. Bold numbers indicate the best results.

| Criteria | Ward | MoRN | MGFA |
|---|---|---|---|
| RI | 0.70 (0.08) | 0.59 (0.10) | **0.89** (0.06) |
| ARI | 0.40 (0.16) | 0.18 (0.20) | **0.79** (0.12) |

RI of 0.93 compared with at most 0.77 for the alternative methods.

### 5.2. Shape data

This subsection investigates binary clustering on the shape space $\mathbb{CP}^3$. We compare MGFA with two alternatives that do not incorporate factor structure: Ward clustering and MoRN. Performance is evaluated using RI and ARI.

We consider an MGFA model with two groups and one latent factor: (G1) $\alpha_1 = (1, 0, 0, 0) \in \mathbb{C}^4$, $v_1 = (0, 0.7, 0.7, 0) \in \mathbb{C}^4$, $\sigma_1 = 0.05$, $\omega_1 = 0.4$; and (G2) $\alpha_2 = (0, 0, 0, 1) \in \mathbb{C}^4$, $v_2 = (0, 0.7, 0.7, 0) \in \mathbb{C}^4$, $\sigma_2 = 0.05$, $\omega_2 = 0.6$. Here the equivalence class $[\alpha_k] = \{c \cdot \alpha_k \mid c \in \mathbb{S}^1\} \in \mathbb{CP}^3$ is the base point, and $\bar{v}_k \in T_{[\alpha_k]}\mathbb{CP}^3$ is the tangent vector associated with $v_k$; see Section E.3.1 for background. In each trial, we draw $n = 500$ samples from the MGFA model and apply MGFA with $q = 1$, Ward clustering, and MoRN to produce two clusters. Table 1 reports the average RIs and ARIs over 100 repeated trials. MGFA achieves an average RI of 0.89, substantially outperforming Ward clustering and MoRN, whose average RIs are at most 0.70.

### 5.3. Hyperbolic data

This section studies clustering on the hyperbolic space $\mathbb{H}^2$. We compare MGFA with Ward clustering and MoRN, nei-

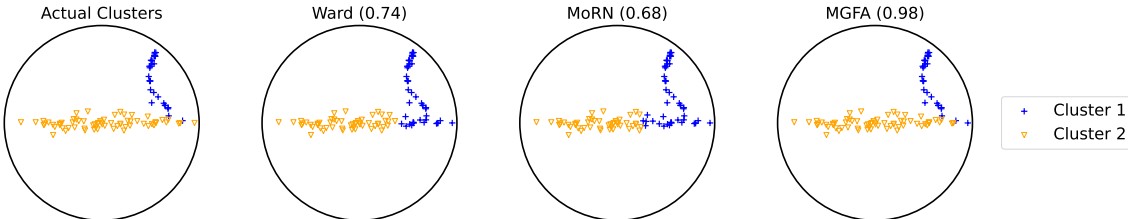

*Figure 3.* Visualization of clustering results on $\mathbb{H}^2$. The first panel shows the true clusters, and the remaining panels show the results for Ward clustering, MoRN, and MGFA, respectively. RIs are reported in parentheses. The data are displayed in the Poincaré representation.

ther of which accounts for latent factor structure, and evaluate performance using RI and ARI.

We consider an MGFA model with two groups and one latent factor: (G1) $\omega_1 = 0.4$, $\alpha_1 = (3, 2, 2)$, $v_1 = (2, 1, 2)$, $\sigma_1 = 0.1$; (G2) $\omega_2 = 0.6$, $\alpha_2 = (1, 0, 0)$, $v_2 = (0, 1, 0)$, $\sigma_2 = 0.1$; where $\alpha_k \in \{x \in \mathbb{R}^3 \mid x_1^2 = 1 + x_2^2 + x_3^2, \ x_1 > 0\}$ is represented on the hyperboloid and $v_k \in T_{\alpha_k}\mathbb{H}^2 = \{v \in \mathbb{R}^3 \mid v_1\alpha_{k1} = v_2\alpha_{k2} + v_3\alpha_{k3}\}$ is the tangent vector. In a single trial, we draw $n = 100$ samples from the model and cluster them into two groups using MGFA with $q = 1$, MoRN, or Ward clustering. Figure 3 visualizes the clustering results. MGFA recovers the latent factor structure, whereas the two competing methods fail to recover the hidden factor pattern. Consequently, MGFA achieves a substantially higher RI (0.99) than its competitors ($\leq 0.74$).

## 6. Real data analysis

We also analyze two datasets from the Alzheimer's Disease Neuroimaging Initiative (ADNI) study (Mueller et al., 2005): a corpus callosal shape dataset and a left hippocampal shape dataset. We apply MGFA to both datasets to demonstrate its ability to identify clinically meaningful subtypes of AD. The corpus callosal shape analysis is deferred to Appendix I.1.

We investigate a left hippocampal shape dataset from the ADNI study. The hippocampus is a central structure in the limbic system and plays a crucial role in memory formation and spatial navigation (Burgess et al., 2002). It is among the first brain regions affected by Alzheimer's disease (AD) and other dementias, and hippocampal atrophy is strongly associated with mild cognitive impairment (MCI) and AD (Pennanen et al., 2004; Tapiola et al., 2008; Rao et al., 2022). Previous studies have often used volumetric and morphometric measures, such as hippocampal volume (Du et al., 2001; Pennanen et al., 2004) and radial distance (Scher et al., 2007), as markers of atrophy. In contrast, we focus on hippocampal shape, which provides a more detailed representation of structural variation. Specifically, we study late mild cognitive impairment (LMCI), a transitional stage between normal cognition and AD that is marked by distinct patterns of hippocampal atrophy. Characterizing morpho-

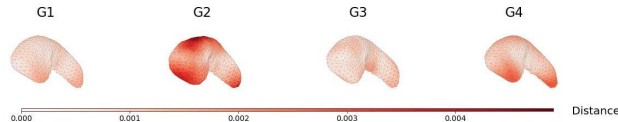

*Figure 4.* Visualization of the mean hippocampal shapes for each subgroup. The color gradient represents the surface distance between each subgroup-specific mean shape and the overall mean shape of the LMCI cohort.

logical variation in LMCI may improve understanding of cognitive decline and help predict progression from MCI to AD.

To examine these shape variations, we aim to identify meaningful subgroups within the LMCI cohort using left hippocampal shapes. After excluding subjects with missing data, the final sample contains 320 LMCI subjects; demographic information is summarized in Table 2. For each subject, we use the segmentation and registration tool `FSL-FIRST` (Patenaude et al., 2011; Patenaude, 2007) to process T1-weighted MRI and extract 3D landmark data for the left hippocampus. Each hippocampus is represented by $m_0 = 732$ landmarks in $\mathbb{R}^3$, denoted by $Y \in \mathbb{R}^{m_0 \times 3}$. We model its shape as a point on a sphere using the Helmert submatrix $H \in \mathbb{R}^{(m_0-1) \times m_0}$, whose rows are given in (I.1). Specifically, multiplication by $H$ removes location information, yielding $Y_H = HY \in \mathbb{R}^{(m_0-1) \times 3}$. We then remove scale by normalizing $Y_H$ and define $X = Y_H / \|Y_H\|_F$, which lies on the sphere $\mathbb{S}^{3(m_0-1)-1}$. Because orientation has already been removed by `FSL-FIRST`, $X$ provides a suitable shape representation for the left hippocampus, reducing the analysis to spherical data analysis. In addition to hippocampal shape and demographic information, each subject has measurements of MMSE, hippocampal volume, and intracranial volume (ICV). Our objective is to stratify the LMCI cohort into distinct subgroups and examine subgroup differences.

Using MGFA with latent dimension $q = 1$, we cluster the LMCI cohort into four groups. Figure 4 shows the mean hippocampal shape for each subgroup. The color gradient represents the surface distance from the reference shape, defined as the overall mean shape of the LMCI cohort. G2

*Table 2.* Demographic information for the LMCI cohort used in the left hippocampal shape analysis.

| Sample Size | Sex (F/M) | Age (years) | Education Length (years) |
|:---:|:---:|:---:|:---:|
| $n = 320$ | 118 / 202 | mean 75.51 (range 59.90–89.60) | mean 16.06 (range 6–20) |

shows the largest deviation from the reference shape, with the most pronounced differences in the superior and inferior regions. G4 also exhibits a notable deviation, primarily in the head and tail regions. In contrast, G1 and G3 are closer to the reference shape, suggesting that these subgroups represent more typical hippocampal morphology within the LMCI cohort.

Next, we examine subgroup demographic characteristics, reported in Table 9 of Appendix I.3. G2 and G4, which show the greatest deviations from the reference shape, also have the most distinctive demographic profiles. G2, the smallest subgroup, has the youngest age, lowest education level, lowest MMSE score, and lowest Hipp/ICV ratio. In contrast, G4 is the oldest subgroup and has the longest education duration and the second-highest MMSE score. This inverse relationship between cognitive performance and education duration has also been reported in previous studies (Huang & Zhu, 2022).

## 7. Discussion

This paper introduces MGFA for clustering data on Riemannian homogeneous spaces. By incorporating latent factor structures, MGFA is more expressive than mixtures of radial distributions and is well suited to manifold-valued data whose subpopulations exhibit anisotropic variation. On the theoretical side, we develop a geometric statistical framework that yields entropy estimates for MGFA and non-asymptotic convergence rates for the MLE. On the computational side, we propose an efficient iterative algorithm for MGFA estimation and clustering, with implementation details for spheres, shape spaces, and hyperbolic spaces. Extensive numerical studies demonstrate the advantages of MGFA over competing methods. We conclude with several directions for future research.

- **Parameter estimation and identifiability:** We establish a Hellinger convergence rate for the density induced by the MLE, but whether the underlying model parameters can be estimated at the same rate remains open. This question is closely tied to identifiability of the MGFA model. In Euclidean spaces, convergence rates for parameter estimation have been obtained under strong identifiability conditions (Ho & Nguyen, 2016a;b; Heinrich & Kahn, 2018). Extending such results to Riemannian homogeneous spaces is an important direction.

- **Model selection:** Selecting the number of clusters and latent factors is critical for applying MGFA. A central question is whether clusters identified by heuristic selection criteria correspond to genuine subgroups or are artifacts of the fitting procedure. In Euclidean settings, hypothesis-testing-based approaches have been proposed (Lo et al., 2001; Nylund et al., 2007; Liu et al., 2008). Extending these ideas to manifold-valued mixture models is a natural next step.

- **Clustering high-dimensional manifold-valued data:** High-dimensional manifold clustering raises challenges related to computational efficiency and the curse of dimensionality. Developing scalable algorithms for large, high-dimensional manifold-valued datasets is therefore essential. Another promising direction is to investigate whether regularization can mitigate the statistical and computational effects of high dimensionality.

- **Extending MGFA models:** Several extensions of MGFA are worth exploring. Incorporating covariates, such as age or sex in medical imaging studies, could improve scientific interpretability and reveal covariate-dependent subgroup structure. Another direction is to impose shared parameters across clusters, such as a common variance parameter $\sigma$, which may reduce model complexity and improve interpretability.

## Acknowledgments

We thank the anonymous reviewers for their constructive comments and suggestions, which have helped improve the clarity and presentation of this work. The bulk of the work was carried out while HC was a PhD student at University of Toronto. HC was partially supported by NSERC Grant RGPIN-2026-06888. QS was partially supported by NSERC Grant RGPIN-2026-06888, Compute Canada, and MBZUAI.

## Impact Statement

This paper presents work whose goal is to advance the field of Machine Learning. There are many potential societal consequences of our work, none which we feel must be specifically highlighted here.

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

# Appendix

## A. Geometry and statistics

This section provides an additional introduction to differential geometry and empirical process theory. For differential geometry, we introduce smooth manifolds in Section A.1 and Riemannian geometry in Section A.2, incorporating Riemannian manifolds, geodesics, curvature, Jacobi fields, and integration. Subsequently, we present properties and examples of Riemannian homogeneous spaces in Section A.3. Finally, we review concepts from empirical process theory in Section A.4.

### A.1. Smooth manifolds

An $m$-dimensional smooth manifold $\mathcal{M}$ is a topological manifold $\mathcal{M}$ equipped with a maximal family of injective mappings $\varphi_\alpha : U_\alpha \subseteq \mathcal{M} \to \mathbb{R}^m$ such that (1) $\varphi_\alpha$ is a homeomorphism from an open set $U_\alpha$ to an open subset $\varphi(U_\alpha) \subseteq \mathbb{R}^m$; (2) $\bigcup_\alpha U_\alpha = \mathcal{M}$; (3) for any $\alpha, \beta$ with a non-empty $W := U_\alpha \cap U_\beta$, the mapping $\varphi_\beta \circ \varphi_\alpha^{-1}$ is a differentiable mapping of $\varphi_\alpha(W)$ onto $\varphi_\beta(W)$. The collection $\{(U_\alpha, \varphi_\alpha)\}$ satisfying the above properties is called a smooth structure on $\mathcal{M}$. The pair $(U_\alpha, \varphi_\alpha)$ is referred to as an open chart or a local coordinate system on $\mathcal{M}$. If $x \in U_\alpha$ and $\varphi_\alpha(x) = (\mathfrak{x}_1(x), \dots, \mathfrak{x}_m(x))$, the set $U_\alpha$ is called a coordinate neighborhood of $x$ and the numbers $\mathfrak{x}_i(x)$ are called local coordinates of $x$.

Let $\mathcal{M}$ and $\mathcal{N}$ be smooth manifolds. A mapping $f : \mathcal{M} \to \mathcal{N}$ is called differentiable at $x \in \mathcal{M}$ if given a local chart $(V, \psi)$ at $f(x)$ there exists a local chart $(U, \varphi)$ at $x$ such that $f(U) \subseteq V$ and the mapping $\psi \circ f \circ \varphi^{-1} : \varphi(U) \to \psi(f(U))$ is differentiable at $\varphi(x)$. The mapping $f$ is called differentiable if it is differentiable at each point $x \in \mathcal{M}$. A differentiable mapping $f$ is called a diffeomorphism if it is bijective and its inverse $f^{-1}$ is differentiable. Let $C^\infty(\mathcal{M}, \mathcal{N})$ be the set of all differentiable mappings of $\mathcal{M}$ onto $\mathcal{N}$. In particular, when $\mathcal{N} = \mathbb{R}$, $C^\infty(\mathcal{M}, \mathbb{R})$ is the set of all differentiable functions on $\mathcal{M}$, which we denote by $C^\infty(\mathcal{M})$. In addition, we let $C_x^\infty(\mathcal{M})$ be the set of all real-valued functions on $\mathcal{M}$ that are differentiable at $x \in \mathcal{M}$.

Suppose $\mathcal{M}$ is an $m$-dimensional smooth manifold. A tangent vector at $x \in \mathcal{M}$ is a linear mapping $v : C_x^\infty(\mathcal{M}) \to \mathbb{R}$ such that $v(fg) = f(x)v(g) + g(x)v(f)$ for all $f, g \in C_x^\infty(\mathcal{M})$. The set of all tangent vectors at $x$ forms an $m$-dimensional vector space, called the tangent space to $\mathcal{M}$ at $x$ and denoted by $T_x\mathcal{M}$. Let $(U, \varphi)$ be a local chart at $x$ and $\varphi(y) = (\mathfrak{x}_1(y), \dots, \mathfrak{x}_m(y))$ be the coordinate representation. Then $\{ \frac{\partial}{\partial \mathfrak{x}_i} \big|_x \}_{1 \le i \le m}$ gives a set of basis vectors of the tangent space $T_x\mathcal{M}$, where $\frac{\partial}{\partial \mathfrak{x}_i}$ is defined as follows:

$$\frac{\partial}{\partial \mathfrak{x}_i}\bigg|_y f = \frac{\partial \left( f \circ \varphi^{-1} \right)}{\partial \mathfrak{x}_i}\bigg|_{\varphi(y)}, \quad \forall f \in C_y^\infty(\mathcal{M}), \quad \forall y \in U.$$

A tangent vector $v$ in $T_x\mathcal{M}$ can be represented as $v = \sum_{i=1}^m a_i(x) \frac{\partial}{\partial \mathfrak{x}_i}\big|_x$, which is called the coordinate representation of $v$ associated with $(U, \varphi)$. Such representation may vary across different local charts $(U, \varphi)$, while the tangent vector $v$ is independent of the choice of the local chart.

Let $\varphi : \mathcal{M} \to \mathcal{N}$ be a differentiable mapping and $x \in \mathcal{M}$. The differential of $\varphi$ at $x$ is a linear mapping $d\varphi_x : T_x\mathcal{M} \to T_{\varphi(x)}\mathcal{N}$ such that

$$d\varphi_x(v)(f) = v(f \circ \varphi)$$

holds for all $v \in T_x\mathcal{M}$ and $f \in C^\infty(\mathcal{N})$.

A vector field $X$ on a smooth manifold $\mathcal{M}$ is an assignment that assigns to each point $x \in \mathcal{M}$ a tangent vector $X_x \in T_x\mathcal{M}$. Given any $f \in C^\infty(\mathcal{M})$, $Xf : \mathcal{M} \to \mathbb{R}$ is a function given by $Xf(x) = X_x(f)$. A vector field $X$ is called smooth if $Xf \in C^\infty(\mathcal{M})$ for all $f \in C^\infty(\mathcal{M})$. Let $(U, \varphi)$ be a local chart on $\mathcal{M}$ and $\varphi(y) = (\mathfrak{x}_1(y), \dots, \mathfrak{x}_m(y))$. Then for all $y \in U$, $X_y$ can be expressed as $\sum_{i=1}^m a_i(y) \frac{\partial}{\partial \mathfrak{x}_i}\big|_y$ for some $a_i : U \to \mathbb{R}$. A vector field $X$ is smooth if and only if $\{a_i\}_{1 \le i \le m}$ are differentiable functions on $U$ for all local charts $(U, \varphi)$. We denote the set of all smooth vector fields on $\mathcal{M}$ by $\Gamma^\infty(T\mathcal{M})$.

Let $X$ and $Y$ be smooth vector fields on $\mathcal{M}$. There exists a unique smooth vector field $Z$ such that $Zf = (XY - YX)f$ for all $f \in C^\infty(\mathcal{M})$, where $(XY)f = X(Y(f))$ and $(YX)f = Y(X(f))$. We call such $Z$ as the Lie bracket $[X, Y]$ of $X$ and $Y$.

Let $\mathcal{M}$ be a smooth manifold. A family of open sets $U_\alpha \subseteq \mathcal{M}$ with $\bigcup_\alpha U_\alpha = \mathcal{M}$ is said to be locally finite if every point $x \in \mathcal{M}$ has a neighborhood $W$ such that $W \cap V_\alpha$ is non-empty for only a finite number of indices. The support of $\rho : \mathcal{M} \to \mathbb{R}$ is the closure of the set of points where $\rho$ is different from zero. We say a family $\{\rho_\alpha\}$ of differentiable functions $\rho_\alpha : \mathcal{M} \to \mathbb{R}$ is a smooth partition of unity if:
(1) For all $\alpha$, $\rho_\alpha \geq 0$ and the support of $\rho_\alpha$ is contained in a coordinate neighborhood $U_\alpha$ of a smooth structure $\{(U_\beta, \varphi_\beta)\}$ of $\mathcal{M}$.
(2) The family $\{U_\alpha\}$ is locally finite.
(3) $\sum_\alpha \rho_\alpha(x) = 1$ for all $x \in \mathcal{M}$.
The following theorem guarantees the existence of the partition of unity.

**Theorem A.1** (Theorem 5.6, Chapter 0 in Do Carmo (1992)). *A smooth manifold $\mathcal{M}$ has a smooth partition of unity if and only if every connected component of $\mathcal{M}$ is Hausdorff and has a countable basis.*

## A.2. Riemannian geometry

This subsection presents an additional introduction to Riemannian geometry. Section A.2.1 introduces Riemannian manifolds, geodesics, exponential map, logarithm map, and cut locus. Section A.2.2 reviews connection, curvature, and Jacobi fields. Section A.2.3 reviews integration on Riemannian manifolds. For a more comprehensive treatment on these topics, we suggest readers to read Do Carmo (1992); Petersen (2006); Cheeger & Ebin (1975); Helgason (1979).

A.2.1. RIEMANNIAN MANIFOLDS

A Riemannian manifold $(\mathcal{M}, g^\mathcal{M})$ is a smooth manifold $\mathcal{M}$ equipped with a Riemannian metric $g^\mathcal{M}$. The metric is a smoothly varying family of inner products $g_x^\mathcal{M} : T_x\mathcal{M} \times T_x\mathcal{M} \to \mathbb{R}$, where $T_x\mathcal{M}$ is the tangent space of $\mathcal{M}$ at $x \in \mathcal{M}$. A diffeomorphism $F : \mathcal{M} \to \mathcal{M}$ is called an isometry of $\mathcal{M}$ if it preserves the Riemannian metric $g^\mathcal{M}$, that is, $g_{F(x)}^\mathcal{M}(dF_x(v), dF_x(w)) = g_x^\mathcal{M}(v, w)$ for all $v, w \in T_x\mathcal{M}$ and $x \in \mathcal{M}$, where $dF_x$ is the differential of $F$ at $x$. We denote the set of all isometries of $\mathcal{M}$ by $\text{Iso}(\mathcal{M})$. The Riemannian metric allows us to measure geometric quantities on $\mathcal{M}$, such as the lengths of curves, distances, volumes, and curvatures. All quantities determined by the Riemannian metric are preserved under an isometry.

A mapping $\gamma : [a, b] \to \mathcal{M}$ is a piecewise smooth curve if it is continuous and there exists a partition $a = a_1 < \ldots < a_k = b$ of $[a, b]$ such that $\gamma|_{[a_i, a_{i+1}]}$ is smooth for $i = 1, \ldots, k - 1$. Given a piecewise smooth curve $\gamma$ in $\mathcal{M}$, its length is measured by integrating the norm of its tangent vectors along the curve. For any two points $x, y \in \mathcal{M}$, the distance $d_g(x, y)$ is defined as the minimum length over all possible piecewise smooth curves between $x$ and $y$. The function $d_g$ defines a metric on $\mathcal{M}$. The manifold $(\mathcal{M}, g^\mathcal{M})$ is said to be complete if it is complete as a metric space $(\mathcal{M}, d_g)$.

A curve is called a geodesic if it locally minimizes the length between points. The Hopf-Rinow theorem states that a connected manifold $\mathcal{M}$ is complete if and only if all geodesics extend indefinitely (Do Carmo, 1992). Moreover, in a connected complete manifold, any two points can be connected by a length-minimizing geodesic.

For any point $x \in \mathcal{M}$ and vector $v \in T_x\mathcal{M}$, there exists a unique geodesic $\gamma_v(t)$ with $x$ as its initial position and $v$ as its initial velocity. The exponential map $\text{Exp}_x : T_x\mathcal{M} \to \mathcal{M}$ is defined by $\text{Exp}_x(v) = \gamma_v(1)$ provided that $\gamma_v(1)$ exists. If $\mathcal{M}$ is a connected complete manifold, the exponential map is well-defined on the whole tangent space $T_x\mathcal{M}$. The segment domain of $\text{Exp}_x$ is defined as

$$\text{seg}(x) = \{v \in T_x\mathcal{M} \mid d_g(x, \gamma_v(t)) = t \|v\|, \forall t \in [0, 1]\}.$$

By the Hopf-Rinow theorem, $\mathcal{M} = \text{Exp}_x(\text{seg}(x))$. The interior of the segment domain is given by

$$\text{seg}^0(x) = \{sv \in T_x\mathcal{M} \mid s \in [0, 1), v \in \text{seg}(x)\},$$

which forms an open star-shaped domain in $T_x\mathcal{M}$. On $\text{seg}^0(x)$, the exponential map $\text{Exp}_x$ is a diffeomorphism, and its inverse is called the logarithm map, denoted by $\text{Log}_x$.

The set $\text{seg}(x) - \text{seg}^0(x)$ is known as the cut locus of $x$ in $T_x\mathcal{M}$, while the cut locus of $x$ in $\mathcal{M}$ is defined as

$$\text{Cut}(x) := \mathcal{M} - \text{Exp}_x(\text{seg}^0(x)).$$

Since $\mathcal{M}$ is complete, $\text{Cut}(x)$ is a closed set with measure zero in $\mathcal{M}$[3]. This implies that $\text{Exp}(\text{seg}^0(x))$ covers $\mathcal{M}$ except

---

[3]Please see Section A.2.3 for the definition of the volume measure (also called volume density) on a Riemannian manifold.

for a null set. The injectivity radius at $x$ is defined as

$$\mathrm{inj}(x) := d_g(x, \mathrm{Cut}(x)) = \min_{y \in \mathrm{Cut}(x)} d_g(x, y).$$

Finally, the injectivity radius of $\mathcal{M}$ is given by

$$\mathrm{inj}(\mathcal{M}) = \inf_{x \in \mathcal{M}} \mathrm{inj}(x).$$

A.2.2. CONNECTION, CURVATURE, AND JACOBI FIELDS

Let $\mathcal{M}$ be a smooth manifold, $C^\infty(\mathcal{M})$ be the set of all smooth functions on $\mathcal{M}$, and $\Gamma^\infty(T\mathcal{M})$ be the set of all smooth vector fields on $\mathcal{M}$. An affine connection $\nabla$ on $\mathcal{M}$ is a mapping

$$\nabla : \Gamma^\infty(T\mathcal{M}) \times \Gamma^\infty(T\mathcal{M}) \to \Gamma^\infty(T\mathcal{M}), \quad (X, Y) \to \nabla_X Y,$$

such that the following conditions
(1) $\nabla_{fX+gY} Z = f\nabla_X Z + g\nabla_Y Z,$
(2) $\nabla_X(Y + Z) = \nabla_X Y + \nabla_X Z,$
(3) $\nabla_X(fY) = f\nabla_X Y + (Xf)Y$
hold for all $X, Y, Z \in \Gamma^\infty(T\mathcal{M})$ and $f, g \in C^\infty(\mathcal{M})$. Suppose $(\mathcal{M}, g^\mathcal{M})$ is a Riemannian manifold. There exists a unique affine connection $\nabla$ on $\mathcal{M}$ such that the following conditions
(1) $Xg^\mathcal{M}(Y, Z) = g^\mathcal{M}(\nabla_X Y, Z) + g^\mathcal{M}(Y, \nabla_X Z),$          (compatible)
(2) $\nabla_X Y - \nabla_Y X = [X, Y]$          (torsion-free)
hold for all $X, Y, Z \in \Gamma^\infty(T\mathcal{M})$, where $[X, Y] = XY - YX$ is the Lie bracket. This connection $\nabla$ is called the Levi-Civita connection.

A vector field $X$ is said to be parallel along a curve $\gamma$ if $\nabla_{\gamma'} X \equiv 0$, where $\gamma'$ is the velocity of the curve $\gamma$ and $\nabla$ is an affine connection. By the theory of ordinary differential equations, for any $v \in T_{\gamma(0)}\mathcal{M}$, there is a unique vector field $X$ along the curve $\gamma$ such that $X_{\gamma(0)} = v$. This allows us to define the parallel transport $\mathcal{P}_{\gamma(0) \to \gamma(1), \gamma} : T_{\gamma(0)}\mathcal{M} \to T_{\gamma(1)}\mathcal{M}$ such that $\mathcal{P}_{\gamma(0) \to \gamma(1), \gamma}(v) = X_{\gamma(1)}$, where $v \in T_{\gamma(0)}\mathcal{M}$ and $X$ is the unique parallel vector field along $\gamma$ with $X_{\gamma(0)} = v$. One can show that $\mathcal{P}_{\gamma(0) \to \gamma(1), \gamma}$ is an isometry from the tangent space $T_{\gamma(0)}\mathcal{M}$ to the tangent space $T_{\gamma(1)}\mathcal{M}$.

Let $\mathcal{M}$ be a smooth manifold, $\nabla$ be an affine connection, $\Gamma^\infty(T\mathcal{M})$ be the set of smooth vector fields on $\mathcal{M}$, and $[\cdot, \cdot]$ be the Lie bracket. The curvature tensor $R$ of $\nabla$ is given by

$$R(X, Y)Z = \nabla_Y \nabla_X Z - \nabla_X \nabla_Y Z + \nabla_{[X,Y]} Z, \quad \forall X, Y, Z \in \Gamma^\infty(T\mathcal{M}).$$

Suppose $(\mathcal{M}, g^\mathcal{M})$ is a Riemannian manifold and $\nabla$ is the Levi-Civita connection. Then the Riemannian curvature tensor $Rm$ of $(\mathcal{M}, g^\mathcal{M})$ is given by

$$Rm(X, Y, Z, W) = g^\mathcal{M}(R(X, Y)Z, W), \quad \forall X, Y, Z, W \in \Gamma^\infty(T\mathcal{M}),$$

where $R$ is the curvature tensor of $\nabla$. The *sectional curvature* $K(\Pi_x)$ of $(\mathcal{M}, g^\mathcal{M})$ at $x$ with respect to the plane $\Pi_x = \mathrm{span}(X_x, Y_x) \subseteq T_x\mathcal{M}$ is given by

$$K(\Pi_x) = K(X_x, Y_x) := \frac{Rm(X_x, Y_x, X_x, Y_x)}{g^\mathcal{M}(X_x, X_x)g^\mathcal{M}(Y_x, Y_x) - g^\mathcal{M}(X_x, Y_x)^2}.$$

This is independent of the choice of the basis $\{X_x, Y_x\}$ of $\Pi_x$. Proposition A.2 shows that sectional curvatures retain all the information of a Riemannian curvature tensor.

**Proposition A.2** (Lemma 3.3, Chapter 4 in Do Carmo (1992))**.** *The values of $Rm(X_x, Y_x, X_x, Y_x)$ for all $X_x, Y_x \in T_x\mathcal{M}$ determines the tensor $Rm$ at $x$.*

**Definition A.3.** A Riemannian manifold is said to have constant (sectional) curvature if its sectional curvature $K(\Pi_x)$ is a constant, which is independent of $x \in \mathcal{M}$ and $\Pi_x \subseteq T_x\mathcal{M}$.

The Riemannian manifolds of constant curvature are the simplest among all Riemannian manifolds. Theorem A.4 shows that any complete and simply connected manifold of constant curvature is essentially one of the three cases: the Euclidean space $\mathbb{R}^m$, the sphere $\mathbb{S}^m$, or the hyperbolic space $\mathbb{H}^m$, where $m$ is the dimension of the manifold.

**Theorem A.4** (Theorem 4.1, Chapter 8 in Do Carmo (1992))**.** *Let $\mathcal{M}$ be an $m$-dimensional complete Riemannian manifold of constant curvature $\kappa$. Then the universal covering $\widetilde{\mathcal{M}}$ of $\mathcal{M}$, with the covering metric, is isometric to:*
*(a) hyperbolic space $\mathbb{H}^m$, if $\kappa = -1$;*
*(b) Euclidean space $\mathbb{R}^m$, if $\kappa = 0$;*
*(c) sphere $\mathbb{S}^m$, if $\kappa = 1$.*

Finally, let us introduce Jacobi fields as the variation field of an one-parameter family of geodesics. Let $\Gamma(t, s) : \mathcal{I}_1 \times \mathcal{I}_2 \to \mathcal{M}$ be a smooth map such that $\mathcal{I}_1, \mathcal{I}_2 \subseteq \mathbb{R}$ are intervals and for fixed $s$, $\Gamma(t, s)$ is a geodesic. Let $T = d\Gamma(\frac{\partial}{\partial t})$ and $S = d\Gamma(\frac{\partial}{\partial s})$. Then it can be shown that $\nabla_T S = \nabla_S T$, and $S$ satisfies the Jacobi equation:

$$\nabla_T \nabla_T S = R(T, S)T.$$

A vector field $S$, along a geodesic $\gamma$ with tangent vector $T$ satisfying the Jacobi equation is called a Jacobi field along $\gamma$. It can be shown that every Jacobi field along $\gamma$ is given by the geodesic variation field $S = d\Gamma(\frac{\partial}{\partial s})$ of some $\Gamma$.

A.2.3. INTEGRATION ON MANIFOLDS

Let $(\mathcal{M}, g^{\mathcal{M}})$ be a Riemannian manifold, and $A$ be a compact set contained in a local chart $(U, \varphi)$ with the coordinate $\varphi(y) = (\mathfrak{x}_1(y), \ldots, \mathfrak{x}_m(y))$. The volume of $A$ is defined to be

$$\mathrm{vol}(A) = \int_{\varphi(A)} \sqrt{G} \circ \varphi^{-1} d\mathfrak{x}^1 \cdots d\mathfrak{x}^m,$$

where $G = \det(g_{ij})$, $g_{ij} = g^{\mathcal{M}}(\frac{\partial}{\partial \mathfrak{x}_i}, \frac{\partial}{\partial \mathfrak{x}_j})$, and $d\mathfrak{x}^1 \cdots d\mathfrak{x}^m$ is the Lebesgue measure on $\mathbb{R}^m$. This definition is independent of the choice of the coordinate chart. To define the volume of a set $A$ that needs not to be in one coordinate chart, we use the partition of unity argument. More precisely, we pick a locally finite family of coordinate charts $(U_\alpha, \varphi_\alpha, \mathfrak{x}_{\alpha,1}, \ldots, \mathfrak{x}_{\alpha,m})$ with $\bigcup_\alpha U_\alpha = \mathcal{M}$ and a partition of unity $\{\rho_\alpha\}$ subordinate to this family of charts. We set

$$\mathrm{vol}(A) = \sum_\alpha \int_{\varphi_\alpha(A \cap U_\alpha)} \rho_\alpha \sqrt{G_\alpha} \circ \varphi_\alpha^{-1} d\mathfrak{x}_\alpha^1 \cdots d\mathfrak{x}_\alpha^m,$$

as long as each integral in the sum exists. This leads to the following definition.

**Definition A.5.** The Riemannian volume density on $(\mathcal{M}, g^{\mathcal{M}})$ is

$$\mathrm{dvol} = \sum_\alpha \rho_\alpha \sqrt{G_\alpha} \circ \varphi_\alpha^{-1} d\mathfrak{x}_\alpha^1 \cdots d\mathfrak{x}_\alpha^m.$$

*Remark* A.6. (1) This definition is independent of the choice of the family of local charts and the choice of the partition of unity.
(2) This definition does not assume the manifold $\mathcal{M}$ is oriented or compact. If $\mathcal{M}$ is oriented and the coordinate system is taken to be orientation-preserving, then the Riemannian volume density reduces to the Riemannian volume form.

With the Riemannian volume density $\mathrm{dvol}$, we can integrate functions on $\mathcal{M}$. Let $C_c^0(\mathcal{M})$ be the set of compactly supported continuous functions on $\mathcal{M}$. For any $f \in C_c^0(\mathcal{M})$, we can define

$$\int_{\mathcal{M}} f \mathrm{dvol} = \sum_\alpha \int_{U_\alpha} \rho_\alpha f \sqrt{G_\alpha} \circ \varphi_\alpha^{-1} d\mathfrak{x}_\alpha^1 \cdots d\mathfrak{x}_\alpha^m.$$

This integral is well-defined. Let $C_c^\infty(\mathcal{M})$ be the set of compactly supported differentiable functions on $\mathcal{M}$. For any $1 \leq p < \infty$, one can define the $L^p$ norm on $C_c^\infty(\mathcal{M})$ via

$$\|f\|_{L^p} = \left( \int_{\mathcal{M}} |f|^p \mathrm{dvol} \right)^{1/p}.$$

The completion of $C_c^\infty(\mathcal{M})$ under the $L^p$ norm defines the $L^p$ space, $L^p(\mathcal{M})$. The $L^\infty$ norm of $f$ is given by $\|f\|_\infty = \sup_{x \in \mathcal{M}} |f(x)|$ and the completion of $C_c^\infty(\mathcal{M})$ under the $L^\infty$ norm defines the $L^\infty$ space.

A.2.4. VOLUME COMPARISON THEOREM

Evaluating the integral $\int_{\mathcal{M}} f \mathrm{dvol}$ is a fundamental topic in Riemannian geometry. One of the most important tools is the volume comparison theorem, introduced in this section. Specifically, we will express the integral using normal coordinate charts $(\mathrm{Exp}_x(\mathrm{seg}^0(x)), \mathrm{Log}_x)$, where $x \in \mathcal{M}$, $\mathrm{seg}^0(x)$ is the interior of the segment domain of the exponential map $\mathrm{Exp}_x$ and $\mathrm{Log}_x$ denotes the logarithm map. Using polar coordinates $(r, \Theta)$ on this chart, we can write the volume density $\mathrm{dvol}$ as follows

$$\mathrm{dvol} = \lambda(r, \Theta) dr d\Theta. \tag{A.1}$$

where $dr$ is the radial measure, $d\Theta$ is the usual surface measure on the unit sphere $\mathbb{S}^{m-1}$, $m$ is the dimension of $\mathcal{M}$, and $\lambda(r, \Theta)$ is a function defined over $\mathrm{seg}^0(x)$. For convenience, we set $\lambda(r, \Theta) = 0$ outside $\mathrm{seg}^0(x)$. Then the integral of $f$ is given by

$$\int_{\mathcal{M}} f \mathrm{dvol} = \int_{\mathrm{Exp}_x(\mathrm{seg}^0(x))} f \mathrm{dvol} = \int_{\mathrm{seg}^0(x)} f^{\flat} \lambda(r, \Theta) dr d\Theta$$
$$= \int_{T_x \mathcal{M}} f^{\flat} \lambda(r, \Theta) dr d\Theta, \tag{A.2}$$

where $f^{\flat} = f \circ \mathrm{Exp}_x$. The first equality uses the fact that the cut locus $\mathrm{Cut}(x) := \mathcal{M} - \mathrm{Exp}_x(\mathrm{seg}^0(x))$ has measure zero. To evaluate (A.2), we use Theorem A.7 to approximate $\lambda(r, \Theta)$ with simpler functions.

**Theorem A.7** (Volume comparison theorem, Theorem 27, Chapter 6 in Petersen (2006)). *Let $(\mathcal{M}, g^{\mathcal{M}})$ be an $m$-dimensional complete Riemannian manifold whose sectional curvatures lie within $[\kappa_{\min}, \kappa_{\max}]$. Let $\lambda(r, \Theta)$ be given by equation (A.1). Then, for all $(r, \Theta) \in \mathrm{seg}^0(x)$, we have*

$$\mathrm{sn}_{\kappa_{\max}}^{m-1}(r) \leq \lambda(r, \Theta) \leq \mathrm{sn}_{\kappa_{\min}}^{m-1}(r),$$

*where $\mathrm{sn}_{\kappa}(r)$ is defined as*

$$\mathrm{sn}_{\kappa}(r) = \begin{cases} \frac{\sin(\sqrt{\kappa}r)}{\sqrt{\kappa}} \mathbb{1}_{r \leq \frac{\pi}{\sqrt{\kappa}}}, & \text{if } \kappa > 0, \\ r, & \text{if } \kappa = 0, \\ \frac{\sinh(\sqrt{-\kappa}r)}{\sqrt{-\kappa}}, & \text{if } \kappa < 0. \end{cases} \tag{A.3}$$

*Since $\lambda(r, \Theta) = 0$ outside $\mathrm{seg}^0(x)$, the upper bound $\lambda(r, \Theta) \leq \mathrm{sn}_{\kappa_{\min}}^{m-1}(r)$ holds for all $(r, \Theta)$.*

Observe that $\lambda(r, \Theta) = \mathrm{sn}_{\kappa}^{m-1}(r)$ when $\mathcal{M}$ is an $m$-dimensional complete Riemannian manifold of constant curvature $\kappa$. Thus, Theorem A.7 provides a volume comparison between general Riemannian manifolds and manifolds of constant curvature. This gives a powerful tool for the analysis on Riemannian manifolds.

## A.3. Riemannian homogeneous spaces

A.3.1. GEOMETRIC PROPERTY

This section presents geometric properties of Riemannian homogeneous spaces.

**Proposition A.8.** *A Riemannian homogeneous space $(\mathcal{M}, g^{\mathcal{M}})$ satisfies the following properties:*

- *$\mathcal{M}$ is a complete Riemannian manifold.*

- *The sectional curvatures of $\mathcal{M}$ lie within some bounded interval $[\kappa_{\min}, \kappa_{\max}]$.*

- *The injectivity radius $\mathrm{inj}(\mathcal{M})$ is positive.*

- *There exists a constant $c_0 > 0$ such that for any $\alpha_1, \alpha_2 \in \mathcal{M}$ with $d_g(\alpha_1, \alpha_2) < c_0$, $\alpha_1$ and $\alpha_2$ are connected by a unique minimizing geodesic.*

- *Let $\mathrm{dvol}$ be the volume density on $\mathcal{M}$ and $f : \mathcal{M} \to \mathbb{R}$ be an integrable function. Then,*

$$\int_{\mathcal{M}} f \mathrm{dvol} = \int_{\mathcal{M}} f \circ F \mathrm{dvol},$$

*where $F \in \mathrm{Iso}(\mathcal{M})$ is an isometry.*

*Symmetric spaces are homogeneous and therefore satisfy all the above properties. Additionally, if $\mathcal{M}$ is a symmetric space, then for any $\alpha \neq \widetilde{\alpha} \in \mathcal{M}$, there exists an isometry $F : \mathcal{M} \to \mathcal{M}$ such that $F(\alpha) = \widetilde{\alpha}$ and $F \circ F$ is the identity. For further properties of symmetric spaces, see (Helgason, 1979).*

### A.3.2. EXAMPLE

This section presents some examples of Riemannian homogeneous spaces as well as their applications. For more mathematical examples of homogeneous or symmetric spaces, one may refer to (Helgason, 1979).

**Example A.9.** The following Riemannian manifolds are Riemannian symmetric spaces (RSS):

- Simply connected complete Riemannian manifolds with constant curvature are RSS:

  1. Euclidean spaces are simply connected complete Riemannian manifolds with zero curvature. The space of symmetric positive definite (SPD) matrices, endowed with the log-Euclidean (Arsigny et al., 2007), forms a Euclidean space. This space is useful in medical imaging (Arsigny et al., 2006), brain connectivity (You & Park, 2021), and computer vision (Huang et al., 2015).
  2. Spheres are simply connected complete Riemannian manifolds with positive constant curvature. Spherical data can model $\ell_2$-normalized feature vectors and has been widely studied (Mardia & Jupp, 2000; Pewsey & García-Portugués, 2021).
  3. Hyperbolic spaces play a fundamental role in relativity (Jennings, 2012). They are also used to model complex networks and word embeddings, as they can naturally represent exponentially growing structures (Nickel & Kiela, 2017; Krioukov et al., 2010).

- Real projective spaces $\mathbb{R}P^m$ and complex projective spaces $\mathbb{C}P^m$ are RSS. Real projective spaces are useful for axial data analysis (Bhattacharya & Patrangenaru, 2005), while complex projective spaces (shape spaces) are well-suited for modeling 2D contour shapes (Kendall, 1984), with applications in archaeology, medical imaging, and biology (Dryden & Mardia, 2016).

- Grassmann manifolds, which represent linear subspaces in $\mathbb{R}^m$, are RSS and are widely applied in Riemannian optimization (Edelman et al., 1998) and computer vision (Turaga et al., 2008).

- The space of SPD matrices, endowed with the affine-invariant metric, is a symmetric space (Moakher, 2005; Said et al., 2017; Terras, 2012) and finds applications in medical imaging.

- Product spaces of RSS are also RSS. For example, the space of medial atoms, $\mathcal{M} = \mathbb{R}^3 \times \mathbb{R}^+ \times \mathbb{S}^2 \times \mathbb{S}^2$, is useful in the study of 3D geometric objects (Shi et al., 2012). Another example is the torus (Klein et al., 2020).

*Remark* A.10. Some homogeneous spaces are not symmetric, represented by Stiefel manifolds. Stiefel manifolds model frames in $\mathbb{R}^m$ and are widely applied in Riemannian optimization and computer vision (Edelman et al., 1998; Absil et al., 2008; Turaga et al., 2008).

## A.4. Empirical process theory

This section reviews two entropy concepts that are useful in empirical process theory, including metric entropy, applicable to all metric spaces, and bracketing entropy, relevant to functional spaces.

**Definition A.11** (Metric entropy). Let $(\mathcal{X}, d)$ be a metric space, and let $\mathcal{A} \subseteq \mathcal{X}$ be a compact set. An $\epsilon$-net of $\mathcal{A}$ is a finite subset $\{x_1, \ldots, x_N\} \subseteq \mathcal{X}$ such that for any $y \in \mathcal{A}$, there exists an $i \in \{1, \ldots, N\}$ satisfying $d(x_i, y) \leq \epsilon$. The $\epsilon$-covering number $\mathcal{N}(\epsilon, \mathcal{A}, d)$ is the cardinality of the smallest such $\epsilon$-net, and the $\epsilon$-metric entropy is given by

$$H(\epsilon, \mathcal{A}, d) = \log \mathcal{N}(\epsilon, \mathcal{A}, d).$$

**Definition A.12** (Bracketing entropy). Let $\mathcal{F}$ be a functional space consisting of functions $f : \mathcal{M} \to \mathbb{R}$, equipped with a distance metric $d(\cdot, \cdot)$. Given two functions $\ell, u \in \mathcal{F}$ such that $\ell \leq u$, the bracket $[\ell, u]$ is the set of functions $f \in \mathcal{F}$ satisfying $\ell \leq f \leq u$. An $\epsilon$-bracket is a bracket $[\ell, u]$ where $d(\ell, u) \leq \epsilon$. Let $\mathcal{G} \subseteq \mathcal{F}$ be a functional class. The $\epsilon$-bracketing number $\mathcal{N}_B(\epsilon, \mathcal{G}, d)$ is the minimal number of $\epsilon$-brackets required to cover $\mathcal{G}$, and the bracketing entropy is

$$H_B(\epsilon, \mathcal{G}, d) = \log \mathcal{N}_B(\epsilon, \mathcal{G}, d).$$

In empirical process theory (van der Vaart & Wellner, 1996; van de Geer, 2000), it is crucial to estimate the entropy of the studied statistical model in order to derive the convergence rate of $M$-estimators. In our paper, we first estimate entropies of the studied models and then employ empirical process theory to determine the convergence rate of the maximum likelihood estimate.

## B. Proofs for Section 3

### B.1. Proof of Proposition 3.1

*Proof.* We first establish property (i) by discussing two cases.

- When $\mathcal{M}$ is compact, any bounded and measurable function on $\mathcal{M}$ is integrable. Therefore, the function $\widetilde{f}$ is integrable on $\mathcal{M}$.

- When $\mathcal{M}$ is noncompact, we use Theorem A.7 to prove the result. As $\widetilde{f}$ is nonnegative and measurable, to prove its integrability, it suffices to show that

$$\int_{\mathcal{M}} \widetilde{f}(x; \alpha, \sigma) \mathrm{dvol}(x) < \infty.$$

To prove this, we use the polar coordinate chart at $\alpha$, and rewrite the integral as

$$\int_{\mathcal{M}} \widetilde{f}(x; \alpha, \sigma) \mathrm{dvol}(x) = \int_{T_\alpha \mathcal{M}} \widetilde{f}^\flat \lambda(r, \Theta) dr d\Theta = \int_{T_\alpha \mathcal{M}} e^{-r^2/2\sigma^2} \lambda(r, \Theta) dr d\Theta,$$

where $\widetilde{f}^\flat = \widetilde{f} \circ \mathrm{Exp}_\alpha$ and the second equality uses the definition of $\widetilde{f}$. The sectional curvatures of $\mathcal{M}$ are greater than $\kappa_{\min}$, and we assume $\kappa_{\min} < 0$ without loss of generality. It then follows from Theorem A.7 that $\lambda(r, \Theta) \leq \mathrm{sn}_{\kappa_{\min}}^{m-1}(r)$. Then we have

$$\int_{\mathcal{M}} \widetilde{f}(x; \alpha, \sigma) \mathrm{dvol}(x) \leq \int_{T_\alpha \mathcal{M}} e^{-r^2/2\sigma^2} \mathrm{sn}_{\kappa_{\min}}^{m-1}(r) dr d\Theta$$
$$= \mathrm{vol}(\mathbb{S}^{m-1}) \int_0^\infty e^{-r^2/2\sigma^2} \mathrm{sn}_{\kappa_{\min}}^{m-1}(r) dr < \infty.$$

Here $\mathrm{vol}(\mathbb{S}^{m-1})$ is the volume of the unit $(m-1)$-sphere.

Now we prove property (ii). Fix any $\alpha \in \mathcal{M}$. Then for any $\widetilde{\alpha} \in \mathcal{M}$, there exists an isometry $F \in \mathrm{Iso}(\mathcal{M})$ such that $F(\alpha) = \widetilde{\alpha}$, due to the homogeneity of $\mathcal{M}$. Consequently,

$$Z(\widetilde{\alpha}, \sigma) = \int_{\mathcal{M}} e^{-d_g^2(x, \widetilde{\alpha})/2\sigma^2} \mathrm{dvol}(x)$$
$$= \int_{\mathcal{M}} e^{-d_g^2(x, F(\alpha))/2\sigma^2} \mathrm{dvol}(x)$$
$$= \int_{\mathcal{M}} e^{-d_g^2(F^{-1}(x), \alpha)/2\sigma^2} \mathrm{dvol}(x)$$
$$= \int_{\mathcal{M}} e^{-d_g^2(x, \alpha)/2\sigma^2} \mathrm{dvol}(x) = Z(\alpha, \sigma).$$

where the third equality uses that $d_g(x, F(\alpha)) = d_g(F^{-1}(x), \alpha)$ when $F$ is an isometry and the fourth inequality uses the property of an isometry on Riemannian homogeneous spaces (Proposition A.8). Therefore, the normalizing constant is independent of the location $\alpha$. Property (iii) is an immediate result of property (ii).

To prove property (iv), it suffices to show that if

$$\int_{\mathcal{M}} d_g^p(x, \widetilde{\alpha}) f(x; \alpha, \sigma) \mathrm{dvol}(x) \leq \int_{\mathcal{M}} d_g^p(x, b) f(x; \alpha, \sigma) \mathrm{dvol}(x), \quad \forall b \in \mathcal{M}, \tag{B.1}$$

then $\widetilde{\alpha} = \alpha$. Suppose on the contrary that $\widetilde{\alpha}$ satisfies (B.1) but $\widetilde{\alpha} \neq \alpha$. Then we have

$$\int_{\mathcal{M}} d_g^p(x, \widetilde{\alpha}) f(x; \alpha, \sigma) \mathrm{dvol}(x) \leq \int_{\mathcal{M}} d_g^p(x, \alpha) f(x; \alpha, \sigma) \mathrm{dvol}(x).$$

Ignoring the normalizing constant $Z(\phi)$, we have

$$I(\widetilde{\alpha}, \alpha) \leq I(\alpha, \alpha) < \infty, \tag{B.2}$$

where

$$I(a, b) = \int_{\mathcal{M}} d_g^p(x, a) e^{-d_g^2(x, b)/2\sigma^2} \mathrm{dvol}(x), \quad \forall a, b \in \mathcal{M}.$$

By Proposition A.8, we can show that for any isometry $F \in \mathrm{Iso}(\mathcal{M})$,

$$I(a, b) = I(F(a), F(b)), \quad \forall a, b \in \mathcal{M}. \tag{B.3}$$

Since $\mathcal{M}$ is a RSS and $\widetilde{\alpha} \neq \alpha$, we can find an isometry $F \in \mathrm{Iso}(\mathcal{M})$ such that $F(\alpha) = \widetilde{\alpha}$ and $F \circ F$ is the identity, as a consequence of Proposition A.8. Using this $F$ in (B.2) and using (B.3), we have

$$I(\alpha, \widetilde{\alpha}) = I(F(\widetilde{\alpha}), F(\alpha)) = I(\widetilde{\alpha}, \alpha) \leq I(\alpha, \alpha) = I(\widetilde{\alpha}, \alpha).$$

Therefore,

$$I(\alpha, \widetilde{\alpha}) + I(\widetilde{\alpha}, \alpha) \leq I(\alpha, \alpha) + I(\widetilde{\alpha}, \widetilde{\alpha}).$$

Equivalently, we have

$$\int_{\mathcal{M}} D(x; \alpha, \widetilde{\alpha}) \mathrm{dvol}(x) \leq 0, \tag{B.4}$$

where

$$D(x; \alpha, \widetilde{\alpha}) = D_1(d_g(x, \alpha), d_g(x, \widetilde{\alpha})),$$

and

$$D_1(u, v) = u^p e^{-v^2/2\sigma^2} + v^p e^{-u^2/2\sigma^2} - u^p e^{-u^2/2\sigma^2} - v^p e^{-v^2/2\sigma^2}.$$

By calculation, we find that

$$D_1(u, v) = (u^p - v^p) \cdot (e^{-v^2/2\sigma^2} - e^{-u^2/2\sigma^2}) \geq 0, \tag{B.5}$$

where we use the fact that $\phi$ is an increasing function. Notice that the equality in (B.5) holds if and only if $u = v$. Combining this with (B.4), we obtain

$$d_g(x, \alpha) = d_g(x, \widetilde{\alpha}), \quad \forall x \in \mathcal{M},$$

where we use the continuity of the distance function $d_g(\cdot, \cdot)$. This implies $\widetilde{\alpha} = \alpha$.

Now we prove the fifth statement. Let $(\mathcal{N}, g^{\mathcal{N}})$ be a Riemannian homogeneous space, and denote by $d_{\mathcal{M} \times \mathcal{N}}$, $d_{\mathcal{M}}$, and $d_{\mathcal{N}}$ the distance function on $\mathcal{M} \times \mathcal{N}$, $\mathcal{M}$, and $\mathcal{N}$, respectively. The Riemannian Gaussian distribution $RN((\alpha, \beta), \sigma)$ with $\alpha \in \mathcal{M}$, $\beta \in \mathcal{N}$, and $\sigma > 0$ has a density proportional to

$$\widetilde{f}((\alpha, \beta), \sigma) = e^{-\frac{d_{\mathcal{M} \times \mathcal{N}}^2((x,y),(\alpha,\beta))}{2\sigma^2}} \overset{(i)}{=} e^{-\frac{d_{\mathcal{M}}^2(x,\alpha) + d_{\mathcal{N}}^2(y,\beta)}{2\sigma^2}} = \widetilde{f}(\alpha, \sigma) \cdot \widetilde{f}(\beta, \alpha), \tag{B.6}$$

where (i) uses that $d_{\mathcal{M} \times \mathcal{N}}^2((x, y), (\alpha, \beta)) = d_{\mathcal{M}}^2(x, \alpha) + d_{\mathcal{N}}^2(y, \beta)$ on a product manifold. This immediately implies the fifth property. $\qquad\square$

### B.2. Proof of Proposition 3.2

*Proof.* The joint distribution $f(x, z; \alpha, V, \sigma)$ is given by

$$f(x, z; \alpha, V, \sigma) = p(z) \cdot \frac{1}{Z(\sigma)} \exp\left\{ -\frac{d_g^2(x, \mathrm{Exp}_\alpha(Vz))}{2\sigma^2} \right\}.$$

Integrating out the latent variable $z$, we obtain the density function (3.6). $\qquad\square$

## C. Geometric lemma

This section presents a geometric lemma essential to our theoretical analysis. It characterizes the geodesic deviation over a Riemannian manifold with bounded sectional curvatures. It establishes that the deviation between distinct geodesics grows at most exponentially with time, with an order of $e^{ct}$ for some constant $c$. The proof of this lemma leverages the Jacobi field theory.

Specifically, we consider a complete, connected Riemannian manifold $\mathcal{M}$ with sectional curvature bounded in $[\kappa_{\min}, \kappa_{\max}]$. Given two points $\alpha_1$ and $\alpha_2 \in \mathcal{M}$ and two tangent vectors $v_1 \in T_{\alpha_1}\mathcal{M}$ and $v_2 \in T_{\alpha_2}\mathcal{M}$, we define the following measure of deviation between the tangent pairs $(\alpha_1, v_1)$ and $(\alpha_2, v_2)$:

$$d_{T\mathcal{M}}^2((\alpha_1, v_1), (\alpha_2, v_2)) = d_g^2(\alpha_1, \alpha_2) + \|v_1 - \mathcal{P}_{\alpha_2 \to \alpha_1} v_2\|^2, \tag{C.1}$$

where $\mathcal{P}_{\alpha_2 \to \alpha_1} : T_{\alpha_2}\mathcal{M} \to T_{\alpha_1}\mathcal{M}$ denotes the *parallel transport* along a minimizing geodesic connecting $\alpha_2$ to $\alpha_1$. This defines the initial deviation between the geodesics $\mathrm{Exp}_{\alpha_1}(tv_1)$ and $\mathrm{Exp}_{\alpha_2}(tv_2)$.

*Remark* C.1. Definition (C.1) is well-defined when $\alpha_1$ and $\alpha_2$ are sufficiently close to ensure the existence of a unique minimizing geodesic between them. In cases where multiple minimizing geodesics exist, we choose the parallel transport $\mathcal{P}_{\alpha_2 \to \alpha_1}$ that minimizes $\|v_1 - \mathcal{P}_{\alpha_2 \to \alpha_1} v_2\|$. Although $d_{T\mathcal{M}}$ does **not** define a true metric on the tangent bundle $T\mathcal{M}$—as it may violate the triangle inequality—it remains useful for stating and analyzing the following lemma.

Lemma C.2 shows that the deviation between the geodesics $\mathrm{Exp}_{\alpha_1}(tv_1)$ and $\mathrm{Exp}_{\alpha_2}(tv_2)$,

$$d_g(\mathrm{Exp}_{\alpha_1}(tv_1), \mathrm{Exp}_{\alpha_2}(tv_2)),$$

is uniformly bounded for all $t \in [0, \infty)$ by $d_{T\mathcal{M}}((\alpha_1, v_1), (\alpha_2, v_2))$ multiplying an exponential term $e^{ct}$, where the constant $c > 0$ depends on the curvature bounds. This exponential growth reflects the curved geometry of the manifold and stands in contrast to the *linear growth* observed in Euclidean spaces.

**Lemma C.2.** *Suppose $\mathcal{M}$ is a complete connected Riemannian manifold with sectional curvatures constrained within $[-\kappa, \kappa]$ for some $\kappa > 0$. Consider $(\alpha_1, v_1)$ and $(\alpha_2, v_2) \in T\mathcal{M}$, where $\alpha_1$ and $\alpha_2$ are connected by a unique minimizing geodesic. We further assume that $\max\{\|v_1\|, \|v_2\|\} \le A$ for some constant $A > 0$. Then, we have*

$$d_g(\mathrm{Exp}_{\alpha_1}(tv_1), \mathrm{Exp}_{\alpha_2}(tv_2)) \le d_{T\mathcal{M}}((\alpha_1, v_1), (\alpha_2, v_2)) \cdot e^{(1+\kappa A^2)t/2}, \quad \forall t \ge 0.$$

The proof of Lemma C.2 relies on the careful construction of a Jacobi field connecting the geodesics $\mathrm{Exp}_{\alpha_1}(tv_1)$ and $\mathrm{Exp}_{\alpha_2}(tv_2)$. By applying the Jacobi field equation, we characterize the dynamics of this field in terms of the manifold's curvature. This characterization enables us to derive a bound on the geodesic deviation rate. The complete argument is presented in Appendix D.3.1.

## D. MGFA with Riemannian radial distributions

This section provides the proofs corresponding to Section 4. Instead of limiting our analysis solely to Riemannian Gaussian distributions, we consider MGFA within the broader setting of Riemannian radial distributions. This general framework includes the Riemannian Gaussian, Riemannian Laplacian, and von Mises-Fisher distributions as special cases, thereby unifying the analysis across a wide class of distributions. Specifically, we establish Theorem 4.1 and Theorem 4.2 within this generalized context. Moreover, we provide several examples to highlight the versatility and effectiveness of our approach.

This section proceeds as follows. In Section D.1, we review Riemannian radial distributions along with the corresponding MGFA model. In Section D.2, we present the main theorem and discuss several examples, including the proofs of Theorem 4.1 and Theorem 4.2 from the main text. In Section D.3, we provide the complete proof of the main theorem stated in Section D.2.

## D.1. Model

Riemannian radial distributions unify various distributions, including Riemannian Gaussian, von Mises-Fisher, and Riemannian Laplacian distributions, into a single coherent framework (Chen, 2024). These distributions possess favorable properties, making them suitable for modeling isotropic noise on manifolds. In this section, we first review this class of distributions and then integrate them into the MGFA model. Additionally, we discuss the advantages provided by this modeling approach.

### D.1.1. RIEMANNIAN RADIAL DISTRIBUTIONS

Let $(\mathcal{M}, g^{\mathcal{M}})$ be a Riemannian homogeneous space. We shall study the following parametric family of distributions on $\mathcal{M}$:

$$\chi_\phi(x; \alpha, \beta) = \frac{1}{Z(\beta, \phi)} e^{-\beta \phi(d_g(x, \alpha))}, \quad \forall x \in \mathcal{M}, \tag{D.1}$$

where $\phi : [0, \infty) \to [0, \infty)$ is an predefined increasing function, $\alpha \in \mathcal{M}$ is the base, $\beta$ is the temperature parameter, and $Z(\beta, \phi) = \int_{\mathcal{M}} e^{-\beta \phi(d_g(x, \alpha))} \mathrm{dvol}(x)$ is the normalizing constant. We refer to it as a Riemannian radial distribution as its density only relies on the distance from the base. The function $\phi$ controls the dispersion pattern of the distribution and is referred to as the dispersion function. For example, we can regard $\phi(r) = r^2$ and $\beta = 1/2\sigma^2$ in the Riemannian Gaussian distribution $RN(\alpha, \sigma)$. We can regard $\phi(r) = 1 - \cos(r)$ and $\beta = \kappa$ in the von Mises-Fisher distribution $f_{\mathrm{vMF}}(x \mid \alpha, \kappa)$ on spheres (Banerjee et al., 2005). In addition, Riemannian Laplacian distributions correspond to the case $\phi(r) = r$. Different dispersion patterns allow us to model different tail distributions.

Riemannian radial distributions possess favorable properties. First, the homogeneity of $\mathcal{M}$ implies that the normalizing constant $Z(\beta, \phi)$ is independent of the location $\alpha$, and omitting this parameter in $Z$ is harmless. As a consequence, the MLE of the base $\alpha$ is an M-estimator defined as

$$\widehat{\alpha} = \underset{\alpha}{\operatorname{argmin}} \sum_{i=1}^{n} \phi(d_g(x_i, \alpha)),$$

where $\{x_i\}$ are independent samples drawn from $\chi_\phi(x; \alpha, \beta)$. Moreover, when $\mathcal{M}$ is a Riemannian symmetric space and $\phi$ is strictly increasing, any $L^p$ center of mass ($0 < p < \infty$) with respect to $\chi_\phi(x; \alpha, \beta)$ is uniquely given by the location parameter $\alpha$ (Chen, 2024). Particularly, the Fréchet mean is $L^2$ center of mass and thus equals to $\alpha$. Consequently, $\alpha$ models the center of the isotropic distribution $\chi_\phi(x; \alpha, \beta)$.

### D.1.2. MGFA WITH RIEMANNIAN RADIAL DISTRIBUTIONS

We now integrate Riemannian radial distributions into the MGFA model. First, we recall the geodesic factor model:

$$x = \mathrm{Exp}(\mathrm{Exp}(\alpha, Vz), \epsilon), \quad \alpha \in \mathcal{M}, V = (v_1, \ldots, v_q), v_i \in T_\alpha \mathcal{M},$$

where $\alpha$ represents the base point, $V$ is the loading matrix, $z$ denotes the latent factor, and $\epsilon$ accounts for the noise term. In Section 3, we modeled the noise distribution $\mathrm{Exp}(\cdot, \epsilon)$ using Riemannian Gaussian distributions $RN(\cdot, \sigma)$. Here, we extend this framework by employing Riemannian radial distributions $\chi_\phi(x; \cdot, \beta)$, where $\phi$ is an increasing function known *a priori*. By integrating out the latent factor $z$, we obtain the density function of this geodesic factor regression model:

$$f_\phi(x; \alpha, V, \beta) = \frac{1}{Z(\beta, \phi)} \int_{z \in \mathbb{R}^q} e^{-\beta \phi(d_g(x, \mathrm{Exp}_\alpha(Vz)))} p(z) dz, \tag{D.2}$$

where $p(z) = \frac{1}{(2\pi)^{q/2}} e^{-\frac{\|z\|^2}{2}}$ is the density of $N(0, I_q)$ and $Z(\beta, \phi)$ is the normalizing constant in (D.1). We can then formulate MGFA as a mixture of these models with Riemannian radial distributions:

$$f_\phi(x; \Upsilon) = \sum_{k=1}^{K} \omega_k f_\phi(x; \alpha_k, V_k, \beta_k), \tag{D.3}$$

where $\omega = (\omega_1, \ldots, \omega_k)$ is the weight vector with $\omega_k \geq 0$ and $\sum_k \omega_k = 1$, $f_\phi(x; \alpha_k, V_k, \beta_k)$ is the density function in (D.2), and $\Upsilon = \{\omega_k, \alpha_k, V_k, \beta_k\}_{k=1}^{K}$ is the set of parameters. This formulation enables us to handle heterogeneous datasets while accommodating diverse noise distributions. For instance, MGFA with von Mises-Fisher distributions is better-suited for spherical data clustering when the noise behaves more like von Mises-Fisher distributions than Riemannian Gaussian distributions.

## D.2. Theory

We now present the main theoretical results for MGFA with Riemannian radial distributions, followed by a discussion of specific examples, including Riemannian Gaussian, von Mises-Fisher, and Riemannian Laplacian distributions. The primary result is an entropy estimation theorem for the function class

$$\mathcal{F}_\Xi^\phi = \{f_\phi(x; \Upsilon) \mid \Upsilon \in \Xi\}, \tag{D.4}$$

where the feasible parameter set $\Xi$ is defined as

$$\Xi = \{\Upsilon = \{\alpha_k, V_k, \beta_k, \omega_k\}_{k=1}^K \mid \omega_k \geq 0, \sum_k \omega_k = 1, \alpha_k \in \mathcal{B}_\mathcal{M}(\alpha^*, D), \beta_k \in [\beta_{\min}, \beta_{\max}],$$
$$V_k = (v_{k1}, \ldots, v_{kq}), \|V_k\|_\mathrm{F} \leq A\}, \tag{D.5}$$

and $\alpha^*, D, \beta_{\min} > 0, \beta_{\max}, A$ are constants. The parameter space $\Xi$ incorporates several constraints: the weights $\omega_k$ are non-negative and sum to unity, the location parameters $\alpha_k$ lie within a bounded geodesic ball $\mathcal{B}_\mathcal{M}(\alpha^*, D)$ in the manifold $\mathcal{M}$, the scale parameters $\beta_k$ are bounded between positive constants $\beta_{\min}$ and $\beta_{\max}$, and the loading matrices $V_k$ have bounded Frobenius norms. Under mild conditions on $\phi$, we establish upper bounds on the bracketing entropies of $\mathcal{F}_\Xi^\phi$ in Theorem D.1.

**Condition 1.** Let $\mathcal{M}$ be an $m$-dimensional Riemannian homogeneous space with sectional curvatures bounded within $[\kappa_{\min}, \kappa_{\max}]$. Let $r_\mathcal{M} = \sup_{x,y} d_g(x, y)$ be the maximum radius of $\mathcal{M}$. We assume $\phi, \beta_{\min}$ and $\beta_{\max}$ satisfy the following conditions.

1. $\phi$ is an increasing, nonnegative, and continuous function on $[0, r_\mathcal{M}]$, where $[0, r_\mathcal{M}]$ stands for $[0, \infty)$ when $r_\mathcal{M} = \infty$. Also, assume $\phi(r)$ is finite for all $r < \infty$.

2. The function $\rho(r) = e^{-\beta\phi(r)}$ has a Lipschitz constant $L$ over the interval $[0, r_\mathcal{M}]$, where $L$ is independent of $\beta$ when $\beta \in [\beta_{\min}, \beta_{\max}]$.

3. When $\mathcal{M}$ is noncompact, we assume $\kappa_{\min} < 0$ and impose the following conditions on $\phi$.

   - The following functions
   
   $$h(r) = e^{-\beta_{\min}\phi(r)}\mathrm{sn}_{\kappa_{\min}}^{m-1}(r), \quad h_1(r) = e^{-\beta_{\min}\phi(r)}\phi(r)\mathrm{sn}_{\kappa_{\min}}^{m-1}(r)$$
   
   are integrable over $[0, \infty)$, where $\mathrm{sn}_{\kappa_{\min}}(\cdot)$ is given by (A.3).
   - For $W = 1/(8(m-1)\sqrt{-\kappa_{\min}})$, it holds that
   
   $$\epsilon^{-1} \int_{W \log(1/\epsilon)}^\infty e^{-\beta_{\min}\phi(r/4)}\mathrm{sn}_{\kappa_{\min}}^{m-1}(r)dr \to 0$$
   
   as $\epsilon$ tends to zero.

**Theorem D.1.** *Consider $\mathcal{M}, \phi, \beta_{\min}, \beta_{\max}$ satisfying Condition 1. Let $\mathcal{F}_\Xi^\phi$ be the function class defined in (D.4) with $\Xi$ defined in (D.5). Then for all sufficiently small $\epsilon$, we have*

$$\log \mathcal{N}_B(\epsilon, \mathcal{F}_\Xi^\phi, d_1) \leq C_* Km \max\{q, 1\} \log(1/\epsilon),$$
$$\log \mathcal{N}_B(\epsilon, \mathcal{F}_\Xi^\phi, d_h) \leq C_* Km \max\{q, 1\} \log(1/\epsilon),$$

*where $d_1$ and $d_h$ represent the $L^1$ and Hellinger distances, and $C_*$ is a constant independent of $m, K, q, \epsilon$.*

An immediate consequence of Theorem D.1 is the convergence rate of the MLE, leveraging tools from empirical process theory. Let $\Upsilon^* \in \Xi$ be the true parameter and $\{x_i\}_{i=1}^n$ be $n$ independent samples drawn from $f_\phi(x; \Upsilon^*)$. The MLE $\widehat{\Upsilon}$ of $\Upsilon$, defined as follows,

$$\widehat{\Upsilon} = \underset{\Upsilon \in \Xi}{\mathrm{argmax}} \sum_{i=1}^n \log f_\phi(x_i; \Upsilon), \tag{D.6}$$

achieves a convergence rate of order $(Km \max\{q, 1\})^{1/2}n^{-1/2}$, up to logarithmic terms. This convergence rate is measured in the Hellinger distance, as stated in the following theorem.

**Theorem D.2.** *Suppose conditions in Theorem D.1 hold. Let $\Upsilon^* \in \Xi$ be the true parameter and $\{x_i\}_{i=1}^n$ be independent samples drawn from $f_\phi(x; \Upsilon^*)$. Then for sufficiently large $n$, the MLE $\widehat{\Upsilon}$ in (D.6) satisfies that*

$$d_h(f_\phi(x; \Upsilon^*), f_\phi(x; \widehat{\Upsilon})) \leq \frac{C(Km \max\{q, 1\})^{1/2} \log n}{n^{1/2}},$$

*holds with probability at least $1 - ce^{-c \log^2 n}$, where $c, C$ are universal constants independent of $K, q, m, n$.*

Both Theorem D.1 and Theorem D.2 incorporate a broad range of applications. Some representative examples are listed as follows.

- **Riemannian Gaussian Distributions** In this case, $\phi(r) = r^2$. The function $\rho(r) = e^{-\beta r^2}$ is Lipschitz continuous with Lipschitz constant $L = \sqrt{2\beta_{\max}/e}$ for all $\beta \in [\beta_{\min}, \beta_{\max}]$. When $\mathcal{M}$ is noncompact and $\kappa_{\min} < 0$, the following functions

$$h(r) = e^{-\beta_{\min} r^2} \mathrm{sn}_{\kappa_{\min}}^{m-1}(r), \quad e^{-\beta_{\min} r^2} r^2 \mathrm{sn}_{\kappa_{\min}}^{m-1}(r)$$

  are both integrable on $[0, \infty)$. Moreover, for $W = 1/(8(m-1)\sqrt{-\kappa_{\min}})$ and sufficiently small $\epsilon$,

$$\epsilon^{-1} \int_{W \log(1/\epsilon)}^{\infty} e^{-\beta_{\min} r^2/16} \mathrm{sn}_{\kappa_{\min}}^{m-1}(r) dr \leq \epsilon^{-1} C_1 e^{-C_2 (\log(1/\epsilon))^2},$$

  where $C_1, C_2 > 0$ are constants independent of $\epsilon$. Thus, when $\epsilon$ tends to zero,

$$\epsilon^{-1} \int_{W \log(1/\epsilon)}^{\infty} e^{-\beta_{\min} r^2/16} \mathrm{sn}_{\kappa_{\min}}^{m-1}(r) dr \to 0.$$

  This verifies Condition 1 for $\phi(r) = r^2$, and thus proves Theorem 4.1 and Theorem 4.2 as a special scenario of Theorem D.1 and Theorem D.2.

- **Power Tail** Similar to Riemannian Gaussian distributions, we can verify Condition 1 for all $\phi(r) = r^p$ with $p > 1$. Therefore, Theorem D.1 and Theorem D.2 hold for these classes of distributions.

- **Riemannian Laplacian Distribution** In this case, we have $\phi(r) = r$. The function $e^{-\beta r}$ is Lipschitz continuous with Lipschitz constant $L = \beta_{\max}$ for all $\beta \in [\beta_{\min}, \beta_{\max}]$. This demonstrates that Theorem D.1 and D.2 hold for Riemannian Laplacian distributions on all compact Riemannian homogeneous spaces. However, when $\mathcal{M}$ is noncompact, things change. First, the functions

$$h(r) = e^{-\beta_{\min} r} \mathrm{sn}_{\kappa_{\min}}^{m-1}(r), \quad e^{-\beta_{\min} r} r \mathrm{sn}_{\kappa_{\min}}^{m-1}(r)$$

  are only integrable on $[0, \infty)$ for sufficiently large $\beta_{\min}$. Otherwise, the normalizing constant $Z(\beta_{\min}, \phi)$ may be infinity and the Riemannian Laplacian distribution may not be well-defined. However, when $\beta_{\min}$ is sufficiently large, we can still verify Condition 1 for $\phi(r) = r$. Thus, when $\mathcal{M}$ is noncompact, Theorem D.1 and D.2 hold for Riemannian Laplacian distributions with a sufficiently large $\beta_{\min}$.

- **von Mises-Fisher Distributions** In this case, $\mathcal{M} = \mathbb{S}^m$ is a sphere and $\phi(r) = 1 - \cos r$. Condition 1 holds for such distributions. Thus, Theorem D.1 and D.2 apply to von Mises-Fisher distributions.

Besides, we remark that when $q = 0$, MGFA reduces to mixtures of Riemannian radial distributions. Therefore, our theory applies to mixtures of Riemannian radial distributions as a special example. This provides theoretical guarantees for mixtures of von Mises-Fisher distributions (Banerjee et al., 2005) and Riemannian Gaussian distributions (Said et al., 2017), thereby addressing existing theoretical gaps in the literature.

### D.3. Proof

This section provides the proofs of Theorem D.1 and Theorem D.2. We begin with the proof of Lemma C.2 in Section D.3.1. Then we establish entropy estimates of bounded sets in $\mathcal{M}$ in Section D.3.2. In Section D.3.3, we derive Lipschitz properties of $f_\phi(x; \alpha, V, \beta)$ with respect to its parameters. Finally, we prove Theorem D.1 in Section D.3.4 and Theorem D.2 in Section D.3.5.

### D.3.1. GEODESIC DEVIATION

This section proves the geodesic deviation bounds in Lemma C.2.

*Proof of Lemma C.2.* Let $\gamma(s)$ denote the unique minimizing geodesic connecting $\alpha_1$ and $\alpha_2$, with $\gamma(0) = \alpha_1$ and $\gamma(1) = \alpha_2$. Define vector fields $v_1(s)$ and $v_2(s)$ along $\gamma(s)$ by parallel transport such that $v_1(0) = v_1$, $v_2(1) = v_2$, and $v_i(s) \in T_{\gamma(s)}\mathcal{M}$ for all $s \in [0, 1]$. We then define the two-parameter map:

$$\Gamma(t, s) = \mathrm{Exp}_{\gamma(s)}\left(t\left[v_1(s) + s(v_2(s) - v_1(s))\right]\right).$$

For each fixed $s$, the curve $\Gamma_s(t) := \Gamma(t, s)$ is a geodesic with initial point $\gamma(s)$ and velocity vector $v_1(s) + s(v_2(s) - v_1(s))$. Therefore, the vector field $S = d\Gamma\left(\frac{\partial}{\partial s}\right)$ is a *Jacobi field* along each geodesic $\Gamma_s$. We also define $T = d\Gamma\left(\frac{\partial}{\partial t}\right)$, the tangent vector to $\Gamma_s$ with respect to $t$. By construction, the norm of $T$ satisfies $\|T\| = \|(1 - s)v_1(s) + sv_2(s)\| \leq A$, since we assume $\max\{\|v_1\|, \|v_2\|\} \leq A$.

The Jacobi field $S$ satisfies the Jacobi equation:

$$\nabla_T \nabla_T S = R(T, S)T,$$

where $R(T, S)T$ is the curvature tensor. Define $h(t, s) = \|S\|^2 + \|\nabla_T S\|^2$. Then

$$\begin{aligned}
\frac{\partial}{\partial t}h(t, s) &= 2\langle S, \nabla_T S\rangle + 2\langle \nabla_T S, \nabla_T \nabla_T S\rangle \\
&= 2\langle S, \nabla_T S\rangle + 2\langle \nabla_T S, R(T, S)T\rangle \\
&\overset{(i)}{\leq} 2\|S\| \cdot \|\nabla_T S\| + 2\kappa\|T\|^2 \cdot \|\nabla_T S\| \cdot \|S\| \\
&\overset{(ii)}{\leq} 2(1 + \kappa A^2)\|\nabla_T S\| \cdot \|S\| \\
&\leq (1 + \kappa A^2) \cdot h(t, s),
\end{aligned}$$

where $(i)$ uses the fact that sectional curvatures of $\mathcal{M}$ are bounded within $[-\kappa, \kappa]$ and $(ii)$ uses the inequality $\|T\| \leq A$. Furthermore, one can further show that

$$h(t, s) \leq h(0, s)e^{(1+\kappa A^2)t},$$

leading to $\|S\|^2 \leq h(t, s) \leq h(0, s)e^{(1+\kappa A^2)t}$.

Observe that $S(0, s) = \gamma'(s)$ and hence $\|S(0, s)\| = \|\gamma'(s)\| = d_g(\alpha_1, \alpha_2)$. By the symmetric lemma $\nabla_T S = \nabla_S T$, we have

$$\nabla_T S(0, s) = \nabla_S T(0, s) = \nabla_S(v_1(s) + s\{v_2(s) - v_1(s)\}) = v_2(s) - v_1(s),$$

where we use the fact that $v_1(s)$ and $v_2(s)$ are parallel along $\gamma$. Combining, we have

$$\begin{aligned}
h(0, s) &= \|S(0, s)\|^2 + \|\nabla_T S(0, s)\|^2 \\
&= d_g^2(\alpha_1, \alpha_2) + \|v_2(s) - v_1(s)\|^2 \\
&= d_{T\mathcal{M}}^2((\alpha_1, v_1), (\alpha_2, v_2)).
\end{aligned}$$

Therefore, $\|S\|^2 \leq d_{T\mathcal{M}}^2((\alpha_1, v_1), (\alpha_2, v_2))e^{(1+\kappa A^2)t}$.

Finally, since $\Gamma_t : s \to \Gamma(t, s)$ is a smooth curve connecting $\mathrm{Exp}_{\alpha_1}(tv_1)$ and $\mathrm{Exp}_{\alpha_2}(tv_2)$, we have

$$d_g(\mathrm{Exp}_{\alpha_1}(tv_1), \mathrm{Exp}_{\alpha_2}(tv_2)) \leq \mathrm{len}(\Gamma_t) := \int_0^1 \|S\| ds \leq d_{T\mathcal{M}}((\alpha_1, v_1), (\alpha_2, v_2)) \cdot e^{(1+\kappa A^2)t/2},$$

where the last inequality uses the upper bound on $\|S\|$. This concludes the proof. $\square$

### D.3.2. BASIC ENTROPY ESTIMATES

Lemma D.3 establishes metric entropy estimates of any bounded set in the metric space $(\mathcal{M}, d_g)$.

**Lemma D.3.** *Let $(\mathcal{M}, g^{\mathcal{M}})$ be an $m$-dimensional complete Riemannian manifold with sectional curvatures constrained within some bounded interval $[\kappa_{\min}, \kappa_{\max}]$. Here we assume without loss of generality that $\kappa_{\min} < 0$ and $\kappa_{\max} > 0$. Further we assume the injectivity radius $\mathrm{inj}(\mathcal{M})$ of $\mathcal{M}$ is positive. Let $\mathcal{X}$ be a fixed bounded set in $\mathcal{M}$. Then there is a constant $\epsilon^* > 0$ independent of $m$ such that for all $\epsilon < \epsilon^*$, we have*

$$\mathcal{N}(\epsilon, \mathcal{X}, d_g) \leq e^{Cm}\epsilon^{-m},$$

*where $d_g$ is the geodesic distance and $C$ is a constant independent of $\epsilon$ and $m$.*

*Proof.* We prove this lemma by constructing an $\epsilon$-net $\mathcal{S}$. Suppose $\mathcal{S} = \{x_1, x_2, \ldots\}$ is constructed by the following greedy procedure. Let $x_1 \in \mathcal{X}$ be chosen arbitrarily. Suppose $x_1, \ldots, x_s$ have been chosen. If the set $\{x \in \mathcal{X} \mid \min_{1 \leq i \leq s} d_g(x, x_i) \geq \epsilon\}$ is non-empty, let $x_{s+1}$ be an arbitrary element of this set. Else end the construction of $\mathcal{S}$. It is easy to check that $\mathcal{S}$ is indeed an $\epsilon$-net of $\mathcal{X}$.

The distance between any two distinct points $x_i, x_j \in \mathcal{S}$ is at least $\epsilon$. Hence, two geodesic balls $\mathcal{B}_{\mathcal{M}}(x_i, \epsilon/2)$ and $\mathcal{B}_{\mathcal{M}}(x_j, \epsilon/2)$ do not intersect, where $\mathcal{B}_{\mathcal{M}}(\alpha, r) = \{x \in \mathcal{M} \mid d_g(x, \alpha) < r\}$ is the geodesic ball in $\mathcal{M}$ with center $\alpha$ and radius $r$. Since $\mathcal{X}$ is a bounded set, we know $\mathcal{X} \subseteq \mathcal{B}_{\mathcal{M}}(x^*, D)$ for some $x^* \in \mathcal{X}$ and $D > 0$. Then we have $\bigcup_{x_i \in \mathcal{S}} \mathcal{B}_{\mathcal{M}}(x_i, \epsilon/2) \subseteq \mathcal{B}_{\mathcal{M}}(x^*, D + \epsilon/2)$ and the volume of $\bigcup_{x_i \in \mathcal{S}} \mathcal{B}_{\mathcal{M}}(x_i, \epsilon/2)$ should be smaller than that of $\mathcal{B}_{\mathcal{M}}(x^*, D + \epsilon/2)$.

Assume $\epsilon \leq \epsilon^*$, where $\epsilon^* > 0$ is a constant such that

$$\epsilon^* \leq 2\min\{1, \mathrm{inj}(\mathcal{M})\}, \text{ and } \mathrm{sn}_{\kappa_{\max}}(\epsilon^*/2) \geq \epsilon^*/4, \tag{D.7}$$

where $\mathrm{sn}_\kappa(\cdot)$ is given by (A.3). Then we can give an upper bound on $\mathrm{vol}(\mathcal{B}_{\mathcal{M}}(x, D + \epsilon/2))$ and a lower bound on $\mathrm{vol}(\mathcal{B}_{\mathcal{M}}(x, \epsilon/2))$ for all $x \in \mathcal{M}$ using the volume comparison theorem, Theorem A.7. Indeed, the volume of a $(D + \epsilon/2)$-ball is upper bounded by the volume of a $(D + 1)$-ball:

$$\mathrm{vol}(\mathcal{B}_{\mathcal{M}}(x, D + \epsilon/2)) \leq \mathrm{vol}(\mathcal{B}_{\mathcal{M}}(x, D + 1)) \overset{(i)}{=} \int_{T_x\mathcal{M}} \mathbb{1}_{r \leq D+1} \cdot \lambda(r, \Theta) dr d\Theta,$$

where (i) uses the polar coordinate representation of dvol. Since the sectional curvatures of $\mathcal{M}$ are larger than $\kappa_{\min}$, we can use Theorem A.7 to obtain the following bound:

$$\mathrm{vol}(\mathcal{B}_{\mathcal{M}}(x, D + 1)) \leq \int_{T_x\mathcal{M}} \mathbb{1}_{r \leq D+1} \cdot \mathrm{sn}_{\kappa_{\min}}^{m-1}(r) dr d\Theta = \mathrm{vol}(\mathbb{S}^{m-1}) \cdot \int_0^{D+1} \mathrm{sn}_{\kappa_{\min}}^{m-1}(r) dr,$$

where $\mathrm{vol}(\mathbb{S}^{m-1})$ is the volume of the unit $(m-1)$-sphere in $\mathbb{R}^m$. Since

$$\mathrm{sn}_{\kappa_{\min}}(r) \leq e^{\sqrt{-\kappa_{\min}}r}/(2\sqrt{-\kappa_{\min}}),$$

by calculation, we can obtain the following upper bound

$$\mathrm{vol}(\mathcal{B}_{\mathcal{M}}(x, D + \epsilon/2)) \leq \mathrm{vol}(\mathcal{B}_{\mathcal{M}}(x, D + 1)) \leq \mathrm{vol}(\mathbb{S}^{m-1}) \cdot \frac{2e^{\sqrt{-\kappa_{\min}}(m-1)(D+1)}}{(m-1)(2\sqrt{-\kappa_{\min}})^m}. \tag{D.8}$$

Now we turn to the volume of $\epsilon/2$-balls. Since $\epsilon/2 \leq \mathrm{inj}(\mathcal{M})$ and the sectional curvatures of $\mathcal{M}$ are smaller than $\kappa_{\max}$, the volume of an $\epsilon/2$-ball can be lower bounded as follows:

$$\begin{aligned}
\mathrm{vol}(\mathcal{B}_{\mathcal{M}}(x, \epsilon/2)) &= \int_{T_x\mathcal{M}} \mathbb{1}_{r \leq \epsilon/2} \cdot \lambda(r, \Theta) dr d\Theta \\
&\geq \int_{T_x\mathcal{M}} \mathbb{1}_{r \leq \epsilon/2} \cdot \mathrm{sn}_{\kappa_{\max}}^{m-1}(r) dr d\Theta \\
&= \mathrm{vol}(\mathbb{S}^{m-1}) \int_0^{\epsilon/2} \mathrm{sn}_{\kappa_{\max}}^{m-1}(r) dr.
\end{aligned}$$

where the inequality follows from Theorem A.7. Since $\epsilon^*$ satisfies (D.7) and $\mathrm{sn}_{\kappa_{\max}}(r)/r$ is a decreasing function, we have $\mathrm{sn}_{\kappa_{\max}}(r) \geq \frac{r}{2}$ for all $r \leq \epsilon^*/2$. In particular, we have

$$\mathrm{vol}(\mathcal{B}_{\mathcal{M}}(x, \epsilon/2)) \geq \mathrm{vol}(\mathbb{S}^{m-1}) \int_0^{\epsilon/2} \frac{r^{m-1}dr}{2^{m-1}} = \frac{\mathrm{vol}(\mathbb{S}^{m-1})}{m 2^{2m-1}} \cdot \epsilon^m. \tag{D.9}$$

Combining (D.8), (D.9), and the fact that $\bigcup_{x_i \in \mathcal{S}} \mathcal{B}_{\mathcal{M}}(x_i, \epsilon/2) \subseteq \mathcal{B}_{\mathcal{M}}(x^*, D+1)$, we obtain

$$|\mathcal{S}| \cdot \frac{\mathrm{vol}(\mathbb{S}^{m-1})}{m 2^{2m-1}} \cdot \epsilon^m \leq \mathrm{vol}\left(\bigcup_{x_i \in \mathcal{S}} \mathcal{B}_{\mathcal{M}}(x_i, \epsilon/2)\right) \leq \mathrm{vol}(\mathcal{B}_{\mathcal{M}}(x^*, D+1)),$$

where $|\mathcal{S}|$ is the cardinality of the set $\mathcal{S}$. This implies that

$$|\mathcal{S}| \leq e^{Cm} \epsilon^{-m},$$

where $C$ is a constant independent of $\epsilon$ and $m$. This proves the lemma. $\qquad\square$

Lemma D.4 provides a metric entropy estimate of the unit $\ell_1$-simplex in $\mathbb{R}^K$.

**Lemma D.4** (Lemma A.4 in Ghosal & Van Der Vaart (2001)). *Denote the unit $\ell_1$-simplex in $\mathbb{R}^K$ by $\Delta_K = \{(\omega_1, \ldots, \omega_K) : \omega_i \geq 0, \sum \omega_i = 1\}$. Then for $\epsilon \leq 1$, we have*

$$\mathcal{N}(\epsilon, \Delta_k, \|\cdot\|_1) \leq \left(\frac{5}{\epsilon}\right)^{K-1}.$$

### D.3.3. LIPSCHITZ PROPERTIES

This section establishes Lipschitz properties of the function $f_\phi(x; \alpha, V, \beta)$ with respect to its parameters, where $f_\phi(x; \alpha, V, \beta)$ is defined in (D.2). Lemma D.5 establishes the Lipschitz continuity of $f_\phi(x; \alpha, V, \beta)$ with respect to $(\alpha, V)$. The deviation between $(\alpha_1, V_1)$ and $(\alpha_2, V_2)$ is measured as follows:

$$d_{T\mathcal{M}}^2((\alpha_1, V_1), (\alpha_2, V_2)) = d_g^2(\alpha_1, \alpha_2) + \sum_{j=1}^q \|v_{1j} - \mathcal{P}_{\alpha_2 \to \alpha_1} v_{2j}\|^2, \tag{D.10}$$

where $\alpha_i \in \mathcal{M}$, $V_i = (v_{i1}, \ldots, v_{iq})$ with $v_{ij} \in T_{\alpha_i}\mathcal{M}$, and $\mathcal{P}_{\alpha_2 \to \alpha_1} : T_{\alpha_2}\mathcal{M} \to T_{\alpha_1}\mathcal{M}$ is the parallel transport along the minimizing geodesic between $\alpha_1$ and $\alpha_2$. This notion is well-defined when $\alpha_1$ and $\alpha_2$ are sufficiently close, since there exists a unique minimizing geodesic between them. When $\alpha_1$ and $\alpha_2$ are connected by multiple minimizing geodesics, we choose $\mathcal{P}_{\alpha_2 \to \alpha_1}$ such that $\sum_{j=1}^q \|v_{1j} - \mathcal{P}_{\alpha_2 \to \alpha_1} v_{2j}\|^2$ is minimized.

**Lemma D.5.** *Consider $\mathcal{M}, \phi, \beta_{\min}, \beta_{\max}$ satisfying Condition 1. Assume that $\alpha_1, \alpha_2$ are connected by a unique minimizing geodesic, $\|V_i\|_F \leq A$, and $\beta \in [\beta_{\min}, \beta_{\max}]$. Then it holds that*

$$d_\infty(f_\phi(x; \alpha_1, V_1, \beta), f_\phi(x; \alpha_2, V_2, \beta)) \leq \frac{2Le^{2(1+\kappa^2 A^4 + q)}}{Z(\beta_{\max}, \phi)} \cdot d_{T\mathcal{M}}((\alpha_1, V_1), (\alpha_2, V_2)),$$

*where $d_\infty$ is the $L^\infty$ distance, $L$ is the Lipschitz constant in Condition 1, and $\kappa = \max\{|\kappa_{\min}|, |\kappa_{\max}|\}$.*

*Proof.* If $q = 0$, then using the Lipschitz property of $\rho(r) = e^{-\beta\phi(r)}$ and triangle inequality, we have

$$d_\infty(f_\phi(x; \alpha_1, V_1, \beta), f_\phi(x; \alpha_2, V_2, \beta)) \leq \frac{L}{Z(\beta_{\max}, \phi)} d_{T\mathcal{M}}((\alpha_1, V_1), (\alpha_2, V_2)).$$

Now we consider $q > 0$. By definition of $f_\phi$ and the Lipschitz property of the function $\rho(r) = e^{-\beta\phi(r)}$, we have

$$|f_\phi(x; \alpha_1, V_1, \beta) - f_\phi(x; \alpha_2, V_2, \beta)| \leq \frac{1}{Z(\beta, \phi)} \int_{\mathbb{R}^q} \left| e^{-\beta\phi(u_1(z))} - e^{-\beta\phi(u_2(z))} \right| p(z)dz$$

$$\leq \frac{1}{Z(\beta, \phi)} \int_{\mathbb{R}^q} L|u_1(z) - u_2(z)|p(z)dz$$

$$\leq \frac{L}{Z(\beta, \phi)} \int_{\mathbb{R}^q} d_g(\mathrm{Exp}(\alpha_1, V_1 z), \mathrm{Exp}(\alpha_2, V_2 z))p(z)dz,$$

where $u_1(z) = d_g(x, \mathrm{Exp}(\alpha_1, V_1 z))$, $u_2(z) = d_g(x, \mathrm{Exp}(\alpha_2, V_2 z))$, $p(z)$ is the density of a $q$-dimensional standard normal distribution, $L$ is the Lipschitz constant of $\rho(r)$ in Condition 1, and the third inequality uses the triangle inequality. Since the upper bound is independent of $x$, we have

$$d_\infty(f_\phi(x; \alpha_1, V_1, \beta), f_\phi(x; \alpha_2, V_2, \beta)) \leq \frac{L}{Z(\beta_{\max}, \phi)} \int_{\mathbb{R}^q} d_g(\mathrm{Exp}(\alpha_1, V_1 z), \mathrm{Exp}(\alpha_2, V_2 z)) p(z) dz, \qquad \text{(D.11)}$$

where we use $Z(\beta, \phi) \geq Z(\beta_{\max}, \phi)$ when $\beta < \beta_{\max}$. Let $\xi_z = z/\|z\|$ and $t = \|z\|$ for any non-zero $z \in \mathbb{R}^q$. Applying Lemma C.2, we have

$$d_g(\mathrm{Exp}(\alpha_1, V_1 z), \mathrm{Exp}(\alpha_2, V_2 z)) \leq d_{T\mathcal{M}}((\alpha_1, V_1 \xi_z), (\alpha_2, V_2 \xi_z)) \cdot e^{(1 + \kappa A^2)\|z\|/2}$$
$$\leq d_{T\mathcal{M}}((\alpha_1, V_1), (\alpha_2, V_2)) \cdot e^{(1 + \kappa A^2)\|z\|/2},$$

where $d_{T\mathcal{M}}$ is defined in (D.10) and $\kappa = \max\{|\kappa_{\min}|, |\kappa_{\max}|\}$. Substituting this into (D.11), we have

$$d_\infty(f_\phi(x; \alpha_1, V_1, \beta), f_\phi(x; \alpha_2, V_2, \beta)) \leq \frac{L^*}{Z(\beta_{\max}, \phi)} \cdot d_{T\mathcal{M}}((\alpha_1, V_1), (\alpha_2, V_2)).$$

where

$$L^* = L \int_{\mathbb{R}^q} e^{(1 + \kappa A^2)\|z\|/2} p(z) dz.$$

It is easy to show that

$$\int_{\|z\| \leq 2(1 + \kappa A^2)} e^{(1 + \kappa A^2)\|z\|/2} p(z) dz \leq e^{(1 + \kappa A^2)^2} \leq e^{2(1 + \kappa^2 A^4)}.$$

Additionally,

$$\int_{\|z\| \geq 2(1 + \kappa A^2)} e^{(1 + \kappa A^2)\|z\|/2} p(z) dz = \mathrm{vol}(\mathbb{S}^{q-1}) \int_{2(1 + \kappa A^2)}^\infty e^{(1 + \kappa A^2)r/2} e^{-r^2/2} r^{q-1} dr$$
$$\leq \mathrm{vol}(\mathbb{S}^{q-1}) \int_{2(1 + \kappa A^2)}^\infty e^{-r^2/4} r^{q-1} dr$$
$$\leq 2^{q/2}.$$

Therefore, $L^* \leq 2L e^{2(1 + \kappa^2 A^4 + q)}$, concluding the proof. $\qquad \square$

Lemma D.6 establishes the Lipschitz continuity of $f_\phi(x; \alpha, V, \beta)$ with respect to $\beta$.

**Lemma D.6.** *Suppose $\mathcal{M}, \phi, \beta_{\min}, \beta_{\max}$ satisfy Condition 1 with $\kappa_{\min} < 0$. Then it holds that*

$$d_\infty(f_\phi(x; \alpha, V, \beta_1), f_\phi(x; \alpha, V, \beta_2)) \leq \frac{|\beta_1 - \beta_2|}{\beta_{\min} Z(\beta_{\max}, \phi)} \left(1 + \frac{I(\beta_{\min}, \phi)}{Z(\beta_{\max}, \phi)}\right), \quad \forall \beta_1, \beta_2 \in [\beta_{\min}, \beta_{\max}],$$

*where $I(\beta_{\min}, \phi)$ is defined as*

$$I(\beta_{\min}, \phi) = \int_{\mathcal{M}} e^{-\beta_{\min} \phi(d_g(x, \alpha))} \phi(d_g(x, \alpha)) \mathrm{dvol}(x). \qquad \text{(D.12)}$$

*Proof.* We first introduce the following quantity:

$$\mathrm{diff}(\beta_1, \beta_2, u) = \frac{e^{-\beta_1 u}}{Z(\beta_1, \phi)} - \frac{e^{-\beta_2 u}}{Z(\beta_2, \phi)},$$

where $Z(\beta, \phi)$ is the normalizing constant. Then we have

$$
\begin{aligned}
d_\infty(f_\phi(x; \alpha, V, \beta_1), f_\phi(x; \alpha, V, \beta_2)) &\leq \int_{\mathbb{R}^q} \sup_{x \in \mathcal{M}} |\text{diff}(\beta_1, \beta_2, \phi(d_g(x, \text{Exp}(\alpha, Vz))))| p(z) dz \\
&\leq \int_{\mathbb{R}^q} \sup_{u \geq 0} |\text{diff}(\beta_1, \beta_2, u)| p(z) dz \\
&= \sup_{u \geq 0} |\text{diff}(\beta_1, \beta_2, u)|,
\end{aligned}
\tag{D.13}
$$

where the last equality uses the fact that $p(z)$ is the density function of a $q$-dimensional standard normal distribution. It remains to upper bound the quantity in (D.13). By the triangle inequality, we have

$$
\begin{aligned}
|\text{diff}(\beta_1, \beta_2, u)| &\leq \left| \frac{1}{Z(\beta_1, \phi)} - \frac{1}{Z(\beta_2, \phi)} \right| e^{-\beta_1 u} + \frac{|e^{-\beta_1 u} - e^{-\beta_2 u}|}{Z(\beta_2, \phi)} \\
&\leq \frac{|Z(\beta_1, \phi) - Z(\beta_2, \phi)|}{Z(\beta_{\max}, \phi)^2} + \frac{|\beta_1 - \beta_2|}{\beta_{\min} Z(\beta_{\max}, \phi)},
\end{aligned}
\tag{D.14}
$$

where we use that $Z(\beta, \phi) \geq Z(\beta_{\max}, \phi)$ for $\beta \leq \beta_{\max}$ and $|e^{-\beta_1 u} - e^{-\beta_2 u}| \leq |\beta_1 - \beta_2|/\beta_{\min}$ for $\beta_1, \beta_2 \in [\beta_{\min}, \beta_{\max}]$. We now upper bound $|Z(\beta_1, \phi) - Z(\beta_2, \phi)|$. By the definition of $Z(\beta, \phi)$, we have

$$
\begin{aligned}
|Z(\beta_1, \phi) - Z(\beta_2, \phi)| &\leq \int_{\mathcal{M}} \left| e^{-\beta_1 \phi(d_g(x, \alpha))} - e^{-\beta_2 \phi(d_g(x, \alpha))} \right| \text{dvol}(x) \\
&\leq |\beta_1 - \beta_2| \int_{\mathcal{M}} e^{-\beta_{\min} \phi(d_g(x, \alpha))} \phi(d_g(x, \alpha)) \text{dvol}(x),
\end{aligned}
$$

where the second inequality uses the Lipschitz continuity. Substituting this into inequality (D.14), we conclude the proof. $\square$

### D.3.4. PROOF OF THEOREM D.1

We now present the proof of Theorem D.1.

*Proof of Theorem D.1.* We prove this theorem in three steps. First, we construct $\epsilon$-nets for parameters $\alpha, V, \beta$ and $\omega$. Then we use them to construct a net $\mathcal{S}_\Upsilon$ for the parameter set $\Xi$, and establish an upper bound on

$$
\max_{\Upsilon \in \Xi} \min_{\widetilde{\Upsilon} \in \mathcal{S}_\Upsilon} d_\infty(f_\phi(x; \Upsilon), f_\phi(x; \widetilde{\Upsilon})),
\tag{D.15}
$$

where $d_\infty$ is the $L^\infty$ distance and $f_\phi(x; \Upsilon)$ is the density function defined in (D.3). Using this and extra Riemannian geometry, we finally construct brackets for the function class $\mathcal{F}_\Xi^\phi$ and establish the bracketing entropy estimates.

**Step 1.** We first find an $\epsilon$-net $\mathcal{S}_\beta$ for $[\beta_{\min}, \beta_{\max}]$, an $\epsilon$-net $\mathcal{S}_\alpha$ for $\mathcal{B}_\mathcal{M}(\alpha^*, D)$ in $(\mathcal{M}, d_g)$, an $\epsilon$-net $\mathcal{S}_\omega$ for $\Delta_K$ in $(\mathbb{R}^K, \|\cdot\|_1)$. For each $\alpha \in \mathcal{S}_\alpha$, we choose an $\epsilon$-net $\mathcal{S}_{V;\alpha}$ for the set

$$
\{V = (v_1, \ldots, v_q) \mid v_j \in T_\alpha \mathcal{M}, \|V\|_\text{F} \leq A\},
$$

where $\|V\|_\text{F}$ is the Frobenius norm and the $\epsilon$-net $\mathcal{S}_{V;\alpha}$ is constructed under the norm $\|\cdot\|_\text{F}$. By Lemma D.3, Lemma D.4, and classical results, we know for sufficiently small $\epsilon$, the above $\epsilon$-nets can be chosen such that

$$
|\mathcal{S}_\beta| \leq C\epsilon^{-1}, \quad |\mathcal{S}_\alpha| \leq e^{Cm}\epsilon^{-m}, \quad |\mathcal{S}_\omega| \leq C\epsilon^{-(K-1)}, \quad |\mathcal{S}_{V;\alpha}| \leq C\epsilon^{-mq},
$$

where $C$ is a constant independent of $\epsilon, m, K, q$. Note that since $\mathcal{M}$ is homogeneous, there is a constant $\epsilon_0$ such that for any $\alpha_1, \alpha_2 \in \mathcal{M}$ with $d_g(\alpha_1, \alpha_2) \leq \epsilon_0$, there is a unique length-minimizing geodesic connecting them. Thus, we can consider sufficiently small $\epsilon \leq \epsilon_0$ in the remainder of this proof.

**Step 2.** We construct a net $\mathcal{S}_\Upsilon$ for the parameter set $\Xi$ using the above $\epsilon$-nets. Specifically, we set

$$\mathcal{S}_\Upsilon = \{\Upsilon = \{\omega_k, \alpha_k, V_k, \beta_k\}_{k=1}^K \mid \omega = (\omega_1, \ldots, \omega_K) \in \mathcal{S}_\omega, \alpha_k \in \mathcal{S}_\alpha, V_k \in \mathcal{S}_{V;\alpha_k}, \beta_k \in \mathcal{S}_\beta\}. \tag{D.16}$$

For sufficiently small $\epsilon$, the cardinality of $\mathcal{S}_\Upsilon$ satisfies

$$|\mathcal{S}_\Upsilon| \leq \epsilon^{-CKm \max\{q,1\}},$$

where $C$ is a constant independent of $\epsilon, m, K, q$. Additionally, for any $\Upsilon = \{\omega_k, \alpha_k, V_k, \beta_k\}_{k=1}^K \in \Xi$, there exists an $\widetilde{\Upsilon} = \{\widetilde{\omega}_k, \widetilde{\alpha}_k, \widetilde{V}_k, \widetilde{\beta}_k\}_{k=1}^K \in \mathcal{S}_\Upsilon$ such that

$$\|\omega - \widetilde{\omega}\|_1 \leq \epsilon, \tag{D.17}$$

$$d_g(\alpha_k, \widetilde{\alpha}_k) \leq \epsilon, \tag{D.18}$$

$$d_{T\mathcal{M}}((\alpha_k, V_k), (\widetilde{\alpha}_k, \widetilde{V}_k)) \leq \sqrt{2}\epsilon, \tag{D.19}$$

$$|\beta_k - \widetilde{\beta}_k| \leq \epsilon, \quad \forall k \leq K, \tag{D.20}$$

where $d_{T\mathcal{M}}(\cdot, \cdot)$ is defined as follows:

$$d_{T\mathcal{M}}^2((\alpha_k, V_k), (\widetilde{\alpha}_k, \widetilde{V}_k)) = d_g^2(\alpha_k, \widetilde{\alpha}_k) + \sum_j \|v_{kj} - \mathcal{P}_{\widetilde{\alpha}_k \to \alpha_k} \widetilde{v}_{kj}\|^2,$$

and $\mathcal{P}_{\widetilde{\alpha}_k \to \alpha_k} : T_{\widetilde{\alpha}_k}\mathcal{M} \to T_{\alpha_k}\mathcal{M}$ is the parallel transport along the minimizing geodesic connecting $\widetilde{\alpha}_k$ and $\alpha_k$. Here the minimizing geodesic connecting $\widetilde{\alpha}_k$ and $\alpha_k$ is unique because $d_g(\alpha_k, \widetilde{\alpha}_k) \leq \epsilon \leq \epsilon_0$.

We now upper bound the distance $d_\infty(f_\phi(x; \Upsilon), f_\phi(x; \widetilde{\Upsilon}))$, where $\Upsilon, \widetilde{\Upsilon}$ satisfy (D.17), (D.18), (D.19), and (D.20). Specifically, we observe that for any $\Upsilon \in \Xi$,

$$0 \leq f_\phi(x; \Upsilon) \leq \frac{1}{Z(\beta_{\max}, \phi)}. \tag{D.21}$$

where $Z(\beta, \phi)$ is the normalizing constant that is decreasing in $\beta$. By the triangle inequality and (D.17), we have

$$d_\infty(f_\phi(x; \Upsilon), f_\phi(x; \widetilde{\Upsilon})) \leq \frac{\epsilon}{Z(\beta_{\max}, \phi)} + \sup_k d_\infty(f_\phi(x; \alpha_k, V_k, \beta_k), f_\phi(x; \widetilde{\alpha}_k, \widetilde{V}_k, \widetilde{\beta}_k)), \tag{D.22}$$

Using the triangle inequality and Lemma D.5 and D.6, we have

$$\sup_k d_\infty(f_\phi(x; \alpha_k, V_k, \beta_k), f_\phi(x; \widetilde{\alpha}_k, \widetilde{V}_k, \widetilde{\beta}_k))$$

$$\leq \sup_k d_\infty(f_\phi(x; \alpha_k, V_k, \beta_k), f_\phi(x; \widetilde{\alpha}_k, \widetilde{V}_k, \beta_k)) + \sup_k d_\infty(f_\phi(x; \alpha_k, V_k, \beta_k), f_\phi(\alpha_k, V_k, \widetilde{\beta}_k))$$

$$\leq \frac{2Le^{2(1+\kappa^2 A^4 + q)}}{Z(\beta_{\max}, \phi)} \sup_k d_{T\mathcal{M}}((\alpha_k, V_k), (\widetilde{\alpha}_k, \widetilde{V}_k)) + \sup_k \frac{|\beta_k - \widetilde{\beta}_k|}{\beta_{\min} Z(\beta_{\max}, \phi)} \left(1 + \frac{I(\beta_{\min}, \phi)}{Z(\beta_{\max}, \phi)}\right), \tag{D.23}$$

where $L$ is the Lipschitz constant defined in Condition 1, $\kappa = \max\{|\kappa_{\min}|, |\kappa_{\max}|\}$, and $I(\beta_{\min}, \phi)$ is defined in (D.12). Then by combining (D.23) with (D.22), (D.19), and (D.20), we have that

$$d_\infty(f_\phi(x; \Upsilon), f_\phi(x; \widetilde{\Upsilon})) \leq \frac{\epsilon}{\beta_{\min} Z(\beta_{\max}, \phi)} \left(2\beta_{\min} L e^{2(1+\kappa^2 A^4 + q)} + 1 + \frac{I(\beta_{\min}, \phi)}{Z(\beta_{\max}, \phi)}\right) =: \eta(\epsilon). \tag{D.24}$$

**Step 3.** We now establish bracketing entropy estimates of $\mathcal{F}_\Xi^\phi$ utilizing $\mathcal{S}_\Upsilon$ and (D.24). We begin by determining an envelope of the function class $\mathcal{F}_\Xi^\phi$. First, we recall that the inequality (D.21) holds for all $f(x; \Upsilon)$ with $\Upsilon \in \Xi$. Then we consider $d_g(x, \alpha^*) \geq 2D, \alpha \in \mathcal{B}_\mathcal{M}(\alpha^*, D), \|V\|_F \leq A$, and $\beta \in [\beta_{\min}, \beta_{\max}]$. We have

$$f_\phi(x; \alpha, V, \beta) \leq \frac{1}{Z(\beta_{\max}, \phi)} \int_{\mathbb{R}^q} e^{-\beta\phi(d_g(x, \text{Exp}(\alpha, Vz)))} p(z) dz$$

$$\leq \frac{1}{Z(\beta_{\max}, \phi)} \int_{\mathbb{R}^q} e^{-\beta\phi(|d_g(x, \alpha) - \|Vz\||)} p(z) dz,$$

where the second inequality uses the triangle inequality. By the triangle inequality, we have

$$d_g(x, \alpha) \geq d_g(x, \alpha^*) - d_g(\alpha, \alpha^*) \geq d_g(x, \alpha^*)/2.$$

When $\|z\| \leq d_g(x, \alpha^*)/(4A)$, we have

$$d_g(x, \alpha) - \|Vz\| \geq d_g(x, \alpha^*)/2 - d_g(x, \alpha^*)/4 \geq d_g(x, \alpha^*)/4.$$

When $\|z\| \geq d_g(x, \alpha^*)/(4A) \geq D/(2A)$, we have

$$e^{-\beta\phi(|d_g(x,\alpha) - \|Vz\||)} p(z) \leq p(z).$$

Also, there exists a constant $D_0 \geq 2D$ such that for all $d_g(x, \alpha^*) \geq D_0$, the following inequality holds

$$\int_{\|z\| \geq d_g(x,\alpha^*)/(4A)} p(z)dz = \mathrm{vol}(\mathbb{S}^{q-1}) \int_{d_g(x,\alpha^*)/(4A)}^{\infty} e^{-r^2/2} r^{q-1} dr$$
$$\leq e^{-d_g^2(x,\alpha^*)/(128A^2)}.$$

Therefore, we can show that when $d_g(x, \alpha^*) \geq D_0$, the following holds

$$f_\phi(x; \alpha, V, \beta) \leq \frac{1}{Z(\beta_{\max}, \phi)} \left( e^{-\beta_{\min}\phi(d_g(x,\alpha^*)/4)} + e^{-d_g^2(x,\alpha^*)/(128A^2)} \right) =: H_0(x). \tag{D.25}$$

This implies the following envelope function of $f_\phi(x; \alpha, V, \beta)$ in $\mathcal{F}_{\Xi}^{\phi}$:

$$f_\phi(x; \alpha, V, \beta) \leq H(x) := \begin{cases} H_0(x), & \text{if } d_g(x, \alpha^*) \geq D_0, \\ \frac{1}{Z(\beta_{\max}, \phi)}, & \text{otherwise.} \end{cases}$$

Denote $\mathcal{S}_{\mathcal{F}} = \{f(x; \Upsilon) \mid \Upsilon \in \mathcal{S}_{\Upsilon}\}$, where $\mathcal{S}_{\Upsilon}$ is given by (D.16). Then for any $f_i \in \mathcal{S}_{\mathcal{F}}$, we construct the bracket $\{l_i, u_i\}$ as follows:

$$l_i = \max\{f_i - \eta(\epsilon), 0\}, \quad u_i = \min\{f_i + \eta(\epsilon), H(x)\},$$

where $\eta(\epsilon)$ is defined in (D.24). Applying Step 2, particularly the inequality (D.24), we can immediately show that $\mathcal{F}_{\Xi}^{\phi} \subseteq \cup_i[l_i, u_i]$ and $u_i - l_i \leq \min\{2\eta(\epsilon), H(x)\}$. These brackets cover the function class $\mathcal{F}_{\Xi}^{\phi}$ and it remains to upper bound the $L^1$ distance between $u_i, l_i$. To that end, we compute for any $B > D_0$,

$$d_1(u_i, l_i) := \int_{\mathcal{M}} (u_i - l_i)\mathrm{dvol} \leq \int_{d_g(x,\alpha^*) \leq B} 2\eta(\epsilon)\mathrm{dvol}(x) + \int_{d_g(x,\alpha^*) > B} H(x)\mathrm{dvol}(x)$$
$$= 2\eta(\epsilon)\mathrm{vol}(\mathcal{B}_{\mathcal{M}}(\alpha^*, B)) + \int_{d_g(x,\alpha^*) > B} H_0(x)\mathrm{dvol}(x). \tag{D.26}$$

Then we proceed in two cases.

- **Compact** $\mathcal{M}$ In this case, we choose $B > r_{\mathcal{M}}$ where $r_{\mathcal{M}} < \infty$ is the radius of $\mathcal{M}$. Then (D.26) reduces to the following bound

$$d_1(u_i, l_i) \leq 2\eta(\epsilon)\mathrm{vol}(\mathcal{M}).$$

Additionally, on a compact manifold, we have

$$Z(\beta_{\max}, \phi) \geq e^{-\beta_{\max}\phi(r_{\mathcal{M}})}\mathrm{vol}(\mathcal{M})$$

and

$$I(\beta_{\min}, \phi) \leq e^{-\beta_{\min}\phi(0)}\phi(r_{\mathcal{M}})\mathrm{vol}(\mathcal{M}).$$

Therefore, using the definition of $\eta(\epsilon)$, we obtain

$$d_1(u_i, l_i) \leq C'\epsilon e^{C'q},$$

where $C' > 0$ is a constant independent of $q, K, m, u_i, l_i, \epsilon$. Since the cardinality of $\{[l_i, u_i]\}$ is less than $\epsilon^{-CKm \max\{q,1\}}$, we can rescale $\epsilon$ to show that for sufficiently small $\epsilon$, the following holds

$$\log \mathcal{N}_B(\epsilon, \mathcal{F}_{\Xi}^{\phi}, d_1) \leq C_* Km \max\{q, 1\} \log(1/\epsilon),$$

where $C_*$ is a constant independent of $m, K, q, \epsilon$. Using $\mathcal{N}_B(\epsilon, \mathcal{F}_{\Xi}^{\phi}, d_h) \leq \mathcal{N}_B(\epsilon^2, \mathcal{F}_{\Xi}^{\phi}, d_1)$, we conclude the proof of the theorem in the case of compact $\mathcal{M}$.

- **Noncompact** $\mathcal{M}$ In this case, we analyze the two terms in (D.26) separately.

**First Term** Since $\mathcal{M}$ is noncompact, $\text{vol}(\mathcal{B}_{\mathcal{M}}(\alpha^*, B))$ diverges as $B$ tends to infinity. Since $\eta(\epsilon)/\epsilon$ is a constant independent of $\epsilon$, there exists a sufficiently large $B_0$ independent of $\epsilon$, such that

$$\eta(\epsilon)/\epsilon \leq \text{vol}(\mathcal{B}_{\mathcal{M}}(\alpha^*, B_0)).$$

Therefore, for all $B \geq B_0$,

$$2\eta(\epsilon)\text{vol}(\mathcal{B}_{\mathcal{M}}(\alpha^*, B)) \leq 2\epsilon(\text{vol}(\mathcal{B}_{\mathcal{M}}(\alpha^*, B)))^2. \tag{D.27}$$

Since the sectional curvatures of $\mathcal{M}$ is greater than $\kappa_{\min} < 0$, we can use Theorem A.7 to show that

$$\text{vol}(\mathcal{B}_{\mathcal{M}}(\alpha^*, B)) \leq \text{vol}(\mathbb{S}^{m-1}) \int_0^B \text{sn}_{\kappa_{\min}}^{m-1}(r)dr \leq G_0 e^{(m-1)\sqrt{-\kappa_{\min}}B}$$

where $G_0$ is independent of $\epsilon$ and $B$. Thus, there is a constant $B_0'$ independent of $\epsilon$ such that

$$\text{vol}(\mathcal{B}_{\mathcal{M}}(\alpha^*, B)) \leq e^{2(m-1)\sqrt{-\kappa_{\min}}B} \tag{D.28}$$

holds for all $B \geq B_0'$. Combining (D.28) with (D.27), we have that for all $B \geq \max\{B_0, B_0'\}$,

$$2\eta(\epsilon)\text{vol}(\mathcal{B}_{\mathcal{M}}(\alpha^*, B)) \leq 2\epsilon(\text{vol}(\mathcal{B}_{\mathcal{M}}(\alpha^*, B)))^2 \leq 2\epsilon e^{4(m-1)\sqrt{-\kappa_{min}}B}.$$

Let $B = W\log(1/\epsilon)$, where $W = 1/(8(m-1)\sqrt{-\kappa_{min}})$. Then for all sufficiently small $\epsilon$,

$$2\eta(\epsilon)\text{vol}(\mathcal{B}_{\mathcal{M}}(\alpha^*, W\log(1/\epsilon))) \leq 2\sqrt{\epsilon}. \tag{D.29}$$

**Second Term** For the second term in (D.26), we employ the integral representation (A.2) and Theorem A.7 to show that

$$\int_{d_g(x,\alpha^*) \geq B} H_0(x)\text{dvol}(x) \leq \frac{\text{vol}(\mathbb{S}^{m-1})}{Z(\beta_{\max}, \phi)} \int_B^{\infty} \left( e^{-\beta_{\min}\phi(r/4)} + e^{-r^2/(128A^2)} \right) \text{sn}_{\kappa_{\min}}^{m-1}(r)dr, \tag{D.30}$$

where $\text{sn}_{\kappa_{\min}}(\cdot)$ is defined in (A.3). For sufficiently large $B$, we have

$$\int_B^{\infty} e^{-r^2/(128A^2)} \text{sn}_{\kappa_{\min}}^{m-1}(r)dr \leq \int_B^{\infty} e^{-r^2/(256A^2)}dr \leq C_1 e^{-C_2 B^2},$$

where $C_1, C_2 > 0$ are constants independent of $q, K, m, B, \epsilon$. Let $B = W\log(1/\epsilon)$ where $W = 1/(8(m-1)\sqrt{-\kappa_{\min}})$. Then for sufficiently small $\epsilon$, we have

$$\frac{\text{vol}(\mathbb{S}^{m-1})}{Z(\beta_{\max}, \phi)} \int_{W\log(1/\epsilon)}^{\infty} e^{-r^2/(128A^2)} \text{sn}_{\kappa_{\min}}^{m-1}(r)dr \leq \frac{C_1 \text{vol}(\mathbb{S}^{m-1})}{Z(\beta_{\max}, \phi)} e^{-C_2 W^2(\log(1/\epsilon))^2}$$

$$= \frac{C_1 \text{vol}(\mathbb{S}^{m-1})}{Z(\beta_{\max}, \phi)} \epsilon^{C_2 W^2 \log(1/\epsilon)} \leq \epsilon. \tag{D.31}$$

Additionally, Condition 1 states that for $W = 1/(8(m-1)\sqrt{-\kappa_{\min}})$,

$$\epsilon^{-1} \int_{W\log(1/\epsilon)}^{\infty} e^{-\beta_{\min}\phi(r/4)}\mathrm{sn}_{\kappa_{\min}}^{m-1}(r)dr \to 0$$

as $\epsilon$ tends to zero. Therefore, for sufficiently small $\epsilon$, we have

$$\frac{\mathrm{vol}(\mathbb{S}^{m-1})}{Z(\beta_{\max},\phi)} \int_{W\log(1/\epsilon)}^{\infty} e^{-\beta_{\min}\phi(r/4)}\mathrm{sn}_{\kappa_{\min}}^{m-1}(r)dr \le \epsilon.$$

Combining this with (D.31), we obtain that for sufficiently small $\epsilon$,

$$\int_{d_g(x,\alpha^*)\ge W\log(1/\epsilon)} H_0(x)\mathrm{dvol}(x) \le 2\epsilon. \tag{D.32}$$

**Combination**  Combining (D.32) with (D.29) and (D.26), we have that for sufficiently small $\epsilon$,

$$d_1(u_i, l_i) \le 2\sqrt{\epsilon} + 2\epsilon \le 4\sqrt{\epsilon}.$$

Since the cardinality of $\{[l_i, u_i]\}$ is less than $\epsilon^{-CKm\max\{q,1\}}$, we can rescale $\epsilon$ to demonstrate that

$$\log\mathcal{N}_B(\epsilon, \mathcal{F}_\Xi^\phi, d_1) \le C_* Km\max\{q,1\}\log(1/\epsilon),$$

where $C_*$ is a constant independent of $K, m, q, \epsilon$. Using $\mathcal{N}_B(\epsilon, \mathcal{F}_\Xi^\phi, d_h) \le \mathcal{N}_B(\epsilon^2, \mathcal{F}_\Xi^\phi, d_1)$, we conclude the proof of the theorem in the case of noncompact $\mathcal{M}$.

The proof is complete by combining the above two cases. $\square$

D.3.5. PROOF OF THEOREM D.2

*Proof.* Using the bracketing entropy estimates in Theorem D.1, we have

$$J_B(\delta, \mathcal{F}_\Xi^\phi, d_h) := \int_0^\delta \sqrt{\log\mathcal{N}_B(u, \mathcal{F}_\Xi^\phi, d_h)}du$$

$$\le (C_* Km\max\{q,1\})^{1/2} \int_0^\delta \log^{1/2}(1/u)du$$

$$\le (C_* Km\max\{q,1\})^{1/2}\delta\log(1/\delta),$$

for sufficiently small $\delta$. Then applying Theorem 2 in (Wong & Shen, 1995), we obtain that for sufficiently large $n$, the following inequality holds with probability at least $1 - ce^{-cKm\max\{q,1\}\log^2 n}$:

$$d_h(f(x;\Upsilon^*), f(x;\widehat{\Upsilon})) \le \frac{C(Km\max\{q,1\})^{1/2}\log n}{n^{1/2}},$$

where $c, C$ are universal constants independent of $K, q, m, n$. $\square$

# E. Implementation

This section first presents initialization and model selection methods for MGFA. Then we provide implementation details for MGFA on complex projective spaces, spheres, and hyperbolic spaces, assuming Riemannian Gaussian distributions in the MGFA model.

## E.1. Initialization

Similar to $k$-means clustering, the performance of our iterative algorithm is sensitive to the choice of initial cluster assignments. To improve robustness, we propose utilizing multiple initialization strategies and selecting the best candidate based on minimizing the total squared residual error given by $\text{res} = \sum_{k=1}^{K} |\mathcal{I}_k| \cdot \text{res}_k$. Several effective initialization approaches can be employed. One strategy leverages hierarchical clustering methods based on pairwise distances, such as agglomerative clustering with Ward's linkage (Ward Jr, 1963) or average linkage. Another common and straightforward approach is random initialization. Specifically, we implement a random initialization procedure that iteratively selects new centers from the dataset by maximizing the minimum squared distance from the previously selected centers. In practice, performing multiple random initializations and subsequently choosing the best solution according to the lowest residual error helps ensure robustness and consistency of results.

## E.2. Model selection

Implementing MGFA requires specifying the number of clusters $K$ and the latent dimension $q$. For selecting these parameters, we recommend utilizing the elbow method (Thorndike, 1953). Specifically, we fit MGFA to the dataset $\{x_i\}_{i=1}^{n}$ across a range of values for the parameter grid $\{(K, q)\}$. For each combination, we compute the total squared residual error: $\text{res} = \sum_{k=1}^{K} |\mathcal{I}_k| \cdot \text{res}_k$. Next, we plot the residual error res against different values of $(K, q)$. The optimal values of $K$ and $q$ are chosen at the "elbow" point, identified as the parameter combination beyond which increasing $K$ or $q$ yields only marginal reductions in residual error.

## E.3. Implementation - Complex projective spaces $\mathbb{CP}^m$

In this section, we specifically focus on the complex projective space $\mathcal{M} = \mathbb{CP}^m$, also known as Kendall's shape space (Kendall, 1984). We begin by reviewing the geometric structure of this manifold in Section E.3.1. Subsequently, we discuss data simulation procedures, data assignment strategies, and model estimation techniques tailored for $\mathbb{CP}^m$. Implementation details for spheres and hyperbolic spaces, which follow similar logic, are provided in the appendix.

### E.3.1. BASIC GEOMETRY OF $\mathbb{CP}^m$

The complex projective space $\mathbb{CP}^m$ is the quotient manifold $\mathbb{CS}^m/\mathbb{S}^1$, where $\mathbb{CS}^m = \{x \in \mathbb{C}^{m+1} \mid \|x\| = 1\}$ is the unit sphere in $\mathbb{C}^{m+1}$ and $\mathbb{S}^1 = \{a \in \mathbb{C} \mid |a| = 1\}$ is the unit circle in $\mathbb{C}$. The quotient structure arises from the group action of complex multiplication by elements of $\mathbb{S}^1$. Following quotient manifold theory (Absil et al., 2008), $\mathbb{CP}^m$ inherits a natural Riemannian metric, known as the Fubini–Study metric, making it a Riemannian symmetric space.

For completeness, we outline the geometry of $\mathbb{CP}^m$ briefly. Each point in $\mathbb{CP}^m$ is an equivalence class $[x] = \{c \cdot x \mid c \in \mathbb{S}^1\}$ with representative $x \in \mathbb{CS}^m$. The tangent space $T_{[\alpha]}\mathbb{CP}^m$ at $[\alpha]$ is bijectively identified with the horizontal space $\mathcal{H}_\alpha = \{v \in \mathbb{C}^{m+1} \mid \langle v, \alpha \rangle_H = 0\}$, where $\langle v, \alpha \rangle_H = \alpha^H v$ denotes the Hermitian inner product. For $v, w \in \mathcal{H}_\alpha$, the Riemannian metric $g^{\mathbb{CP}}$ satisfies $g_{[\alpha]}^{\mathbb{CP}}(\bar{v}, \bar{w}) = \langle v, w \rangle$, where $\langle v, w \rangle = w^\top v$ denotes the real inner product, treating $\mathbb{C}^{m+1}$ as $\mathbb{R}^{2(m+1)}$. The geodesic distance between two points $[x]$ and $[\alpha]$ is given by $d_g([x], [\alpha]) = \arccos(|\langle x, \alpha \rangle_H|)$. Given $\alpha \in \mathbb{CS}^m$ and $v \in \mathcal{H}_\alpha$, the exponential map $\text{Exp}_{[\alpha]}(\bar{v})$ is computed by

$$x = \cos(\theta)\alpha + \frac{\sin(\theta)}{\theta}v \quad \text{with} \quad \theta = \|v\|.$$

The logarithmic map $\text{Log}_{[\alpha]}([x])$ for $x, \alpha \in \mathbb{CS}^m$ with $\langle x, \alpha \rangle_H \neq 0$ is given by

$$v = \frac{\theta(x - \langle x, \alpha \rangle_H \alpha)}{\|x - \langle x, \alpha \rangle_H \alpha\|} \quad \text{with} \quad \theta = d_g([x], [\alpha]).$$

The segment domain of the exponential map is

$$\text{seg}^0([\alpha]) = \{\bar{v} \in T_{[\alpha]}\mathbb{CP}^m \mid v \in \mathcal{H}_\alpha, \|v\| < \pi/2\}.$$

The volume density dvol on $\mathbb{CP}^m$, expressed using the normal coordinate chart, is given by

$$\text{dvol} = 2\pi \cos(r) \sin^{2m-1}(r) \, dr \, d\Theta,$$

where $(r, \Theta) \in [0, \pi/2] \times \mathbb{CP}^{m-1}$. Using this expression, the integrals of radially symmetric functions $f([x]) = h(d_g([x], [\alpha]))$ simplify as

$$\int_{\mathbb{CP}^m} f([x]) \, d\text{vol}([x]) = 2\pi \int_{\mathbb{CP}^{m-1}} d\Theta \int_0^{\pi/2} h(r) \cos(r) \sin^{2m-1}(r) dr$$

$$= 2\pi \text{vol}(\mathbb{CP}^{m-1}) \int_0^{\pi/2} h(r) \cos(r) \sin^{2m-1}(r) dr. \tag{E.1}$$

This simplification facilitates computation of the normalizing constants for Riemannian Gaussian distributions and their derivatives.

### E.3.2. DATA SIMULATION

We now present an efficient sampling procedure for the MGFA model (3.7), establishing a simulation framework for algorithmic evaluation. Our initial step focuses on generating samples from the Riemannian Gaussian distribution $RN([\alpha], \sigma)$ defined on $\mathbb{CP}^m$, where $[\alpha] \in \mathbb{CP}^m$ is the location parameter, and $\sigma$ represents the dispersion parameter. Sampling from $RN([\alpha], \sigma)$ is equivalent to generating suitable tangent vectors in the tangent space $T_{[\alpha]}\mathbb{CP}^m$ and subsequently applying the exponential map. Leveraging the normal coordinate chart and integral representation from (E.1), we aim to sample pairs $(r, \Theta)$ from the following joint density:

$$p(r, \Theta) \propto e^{-\frac{r^2}{2\sigma^2}} \cos(r) \sin^{2m-1}(r), \quad r \in [0, \pi/2], \Theta \in \mathbb{CP}^{m-1}.$$

Since $r$ and $\Theta$ are independent, we implement a two-step sampling process:

- Sample $\{\Theta_i\}_{i=1}^n \subseteq T_{[\alpha]}\mathbb{CP}^m$ uniformly from $\mathbb{CP}^{m-1}$ in $T_{[\alpha]}\mathbb{CP}^m$.

- Generate $\{r_i\}_{i=1}^n \subseteq [0, \pi/2]$ independently from the following distribution

$$p(r) \propto e^{-\frac{r^2}{2\sigma^2}} \cos(r) \sin^{2m-1}(r), \quad r \in [0, \pi/2].$$

  This can be efficiently achieved by a rejection sampling method (Liu, 2001). Specifically, we can sample an $r$ uniformly over $[0, \pi/2]$ and then accept such sample with probability

$$p_0 = \frac{e^{-\frac{r^2}{2\sigma^2}} \cos(r) \sin^{2m-1}(r)}{e^{-\frac{r_*^2}{2\sigma^2}} \cos(r_*) \sin^{2m-1}(r_*)}, \quad \text{where } r_* = \underset{r \in [0, \pi/2]}{\text{argmax}} \, e^{-\frac{r^2}{2\sigma^2}} \cos(r) \sin^{2m-1}(r).$$

By setting $v_i = r_i \Theta_i$ and $x_i = \text{Exp}_{[\alpha]}(v_i)$, we obtain samples from $RN([\alpha], \sigma)$ on $\mathbb{CP}^m$. Building on this sampling procedure, we can easily generate data from the geodesic factor model (3.5) and the MGFA model (3.7).

### E.3.3. DATA ASSIGNMENT

This section elaborates the data assignment step in Algorithm 1, assuming the estimated parameter set $\widehat{\Upsilon}$ is known. Following the approach outlined in Section 3.2, each data point is assigned to the cluster with the highest likelihood: $\text{prob}_{ik} = \widehat{w}_k \cdot f^{\text{approx}}(x_i; \widehat{\alpha}_k, \widehat{V}_k, \widehat{\sigma}_k)$. The primary challenge lies in evaluating the normalizing constant $Z(\sigma)$ on the manifold $\mathbb{CP}^m$. To efficiently compute $Z(\sigma)$, we rewrite it using the integral formula (E.1) as follows:

$$Z(\sigma) = 2\pi \cdot \text{vol}(\mathbb{CP}^{m-1}) \int_0^{\pi/2} e^{-\frac{r^2}{2\sigma^2}} \cos(r) \sin^{2m-1}(r) \, dr.$$

With this expression, the computation of $Z(\sigma)$ simplifies to evaluating a definite integral over the bounded interval $[0, \pi/2]$. Such an integral can be efficiently approximated using numerical integration methods available in standard computational packages, such as the `quad` function in Python.

E.3.4. MODEL ESTIMATION

This section addresses model estimation in MGFA for $\mathbb{CP}^m$. The only challenge is to solve the following optimization problem:

$$\widehat{\sigma} = \underset{\sigma}{\arg\min} \log(Z(\sigma)) + \frac{\text{res}}{2\sigma^2}, \tag{E.2}$$

where $Z(\sigma)$ is the normalizing constant on $\mathbb{CP}^m$ and $\text{res} > 0$ is a known constant. By taking derivative of the objective function in (E.2), we know that

$$\ell(\sigma) = \frac{Z'(\sigma)}{Z(\sigma)} - \frac{\text{res}}{\sigma^3}$$

contain $\widehat{\sigma}$ as a root. By (E.1), we know

$$\ell_1(\sigma) = \int_0^{\pi/2} e^{-\frac{r^2}{2\sigma^2}} \cos(r) \sin^{2m-1}(r) r^2 dr - \text{res} \cdot \int_0^{\pi/2} e^{-\frac{r^2}{2\sigma^2}} \cos(r) \sin^{2m-1}(r) dr$$

also contains $\widehat{\sigma}$ as a root. Then we can use the bisection method to solve this root. When $\sigma$ or res is small, we can also use a variational method. In this case,

$$\ell_1(\sigma) \approx \int_0^{\infty} e^{-\frac{r^2}{2\sigma^2}} r^{2m+1} dr - \text{res} \cdot \int_0^{\infty} e^{-\frac{r^2}{2\sigma^2}} r^{2m-1} dr,$$

which implies that $\widehat{\sigma} \approx \sqrt{\text{res}/(2m)}$. This gives an accurate approximation when $\widehat{\sigma} \leq 0.05$.

## E.4. Spheres $\mathbb{S}^m$

This section considers the $m$-sphere $\mathcal{M} = \mathbb{S}^m$. We first recap the geometry of $\mathbb{S}^m$ in Section E.4.1. Then we delve into the implementation details of MGFA in later sections.

E.4.1. BASIC GEOMETRY OF $\mathbb{S}^m$

The $m$-sphere $\mathbb{S}^m = \{x \in \mathbb{R}^{m+1} \mid \|x\| = 1\}$ models the set of $\ell_2$ normalized vectors in $\mathbb{R}^{m+1}$. The tangent space at $\alpha \in \mathbb{S}^m$ is given by $T_\alpha \mathbb{S}^m = \{y \in \mathbb{R}^{m+1} \mid \langle \alpha, y \rangle = 0\}$, where $\langle \alpha, y \rangle = \alpha^\top y$ is the inner product in $\mathbb{R}^{m+1}$. The Riemannian metric $g^{\mathbb{S}}$ on $\mathbb{S}^m$ is given by $g_\alpha^{\mathbb{S}}(v, w) = \langle v, w \rangle$, where $\alpha \in \mathbb{S}^m$ and $v, w \in T_\alpha \mathbb{S}^m$. For any $\alpha \in \mathbb{S}^m$ and $v \in T_\alpha \mathbb{S}^m$, the exponential map $\text{Exp}_\alpha(v)$ is given by

$$\text{Exp}_\alpha(v) = \cos(\theta) \cdot \alpha + \frac{\sin(\theta)}{\theta} \cdot v, \quad \theta = \|v\|.$$

For any $\alpha, x \in T_\alpha \mathbb{S}^m$ such that $\alpha + x \neq 0$, the logarithmic map $\text{Log}_\alpha(x)$ is given by

$$\text{Log}_\alpha(x) = \frac{\theta}{\sin(\theta)} \cdot (x - \alpha \cos(\theta)), \quad \theta = \arccos(\langle \alpha, x \rangle).$$

The distance between $\alpha$ and $x$ is $d_g(\alpha, x) = \theta = \arccos(\langle \alpha, x \rangle)$. The interior of the segment domain of $\text{Exp}_\alpha$ is

$$\text{seg}^0(\alpha) = \{v \in T_\alpha \mathbb{S}^m \mid \|v\| < \pi\},$$

which is an open ball in $T_\alpha \mathbb{S}^m$ with center 0 and radius $\pi$. Let $\text{dvol}$ be the volume density on $\mathbb{S}^m$. By using the normal coordinate chart $(\text{Exp}_\alpha(\text{seg}^0(\alpha)), \text{Log}_\alpha)$, we can rewrite $\text{dvol}$ as follows

$$\text{dvol} = \sin^{m-1}(r) dr d\Theta, \tag{E.3}$$

where $(r, \Theta) \in [0, \pi) \times \mathbb{S}^{m-1}$ is the polar coordinate in $\text{seg}^0(\alpha)$. Therefore, for any function $f(x) = h(d_g(x, \alpha))$ defined on $\mathbb{S}^m$, its integral is given by

$$\int_{\mathbb{S}^m} f(x) \text{dvol}(x) = \int_{\mathbb{S}^{m-1}} d\Theta \int_0^\pi h(r) \sin^{m-1}(r) dr = \text{vol}(\mathbb{S}^{m-1}) \int_0^\pi h(r) \sin^{m-1}(r) dr, \tag{E.4}$$

where we use the coordinate representation (E.3) of $\text{dvol}$. This simplification of the integral is attributed to the rotational symmetry of the $m$-sphere, and is useful for the implementation of MGFA detailed in the following sections.

### E.4.2. DATA GENERATION

This section introduces an efficient data generation procedure, which enables us to simulate data from the MGFA model (3.7) and evaluate the performance of the MGFA algorithms in a simulated environment. The key challenge is to generate samples from the Riemannian Gaussian distribution $RN(\alpha, \sigma)$ on the $m$-sphere $\mathbb{S}^m$, where $\alpha \in \mathbb{S}^m$ is the location parameter and $\sigma > 0$ is the dispersion parameter. One observation is that sampling data $\{x_i\}_{i=1}^n$ from $RN(\alpha, \sigma)$ is equivalent to sampling suitable tangent vectors $\{v_i\}_{i=1}^n \in T_\alpha \mathbb{S}^m$ and then applying the exponential map $x_i = \mathrm{Exp}_\alpha(v_i)$. Therefore, by using the polar coordinate system on the normal coordinate chart, we know it suffices to sample $(r_i, \Theta_i) \in [0, \pi] \times \mathbb{S}^{m-1}$ from the following density function

$$p(r, \Theta) \propto e^{-\frac{r^2}{2\sigma^2}} \sin^{m-1}(r), \quad r \in [0, \pi], \Theta \in \mathbb{S}^{m-1}.$$

Since $r$ and $\Theta$ are independent, we can use a two-step sampling:

- Sample $\{\Theta_i\}_{i=1}^n \subseteq T_\alpha \mathbb{S}^m$ uniformly from the unit sphere $\mathbb{S}^{m-1}$ in $T_\alpha \mathbb{S}^m$;

- Generate $\{r_i\}_{i=1}^n \subseteq [0, \pi]$ independently from the following distribution

$$p(r) \propto e^{-\frac{r^2}{2\sigma^2}} \sin^{m-1}(r), \quad r \in [0, \pi].$$

  This step can be efficiently done via a rejection sampling method (Liu, 2001). Specifically, we can sample an $r$ uniformly over $[0, \pi]$ and then accept such sample with probability

$$p_0 = \frac{e^{-\frac{r^2}{2\sigma^2}} \sin^{m-1}(r)}{e^{-\frac{r_*^2}{2\sigma^2}} \sin^{m-1}(r_*)}, \quad \text{where } r_* = \operatorname*{argmax}_{r \in [0, \pi]} e^{-\frac{r^2}{2\sigma^2}} \sin^{m-1}(r).$$

By setting $v_i = r_i \Theta_i$ and $x_i = \mathrm{Exp}_\alpha(v_i)$, we obtain samples from $RN(\alpha, \sigma)$ on $\mathbb{S}^m$. Once we can sample data from $RN(\alpha, \sigma)$ efficiently, it is easy to sample data from the geodesic factor regression model (3.5) and the MGFA model (3.7) accordingly.

### E.4.3. DATA ASSIGNMENT

Now we discuss data assignment when provided with the estimated parameters $\widehat{\Upsilon}$. As mentioned in Section 3.2, we assign samples to the cluster with the highest likelihood, $\mathrm{prob}_{ik} = \widehat{w}_k \cdot f^{\mathrm{approx}}(x_i; \widehat{\alpha}_k, \widehat{V}_k, \widehat{\sigma}_k)$. The only remaining challenge in calculating $\mathrm{prob}_{ik}$ is the calculation of the normalizing constant $Z(\widehat{\sigma}_k)$. In this section, we address this challenge within the context of the $m$-sphere $\mathbb{S}^m$. Specifically, by definition, the normalizing constant $Z(\sigma)$ is given by

$$Z(\sigma) = \int_{\mathcal{M}} e^{-\frac{d_g^2(x, \alpha)}{2\sigma^2}} \mathrm{dvol}(x),$$

where the definition is independent of the choice of $\alpha$. Since $\mathcal{M} = \mathbb{S}^m$, we can use (E.4) to rewrite $Z(\sigma)$ as follows

$$Z(\sigma) = \mathrm{vol}(\mathbb{S}^{m-1}) \int_0^\pi e^{-\frac{r^2}{2\sigma^2}} \sin^{m-1}(r) dr. \tag{E.5}$$

The integral on the right hand side can be efficiently computed using the function `quad` in Python, since it is an integral on a bounded interval $[0, \pi]$.

### E.4.4. MODEL ESTIMATION

Now let us discuss the estimation of the geodesic factor regression models when given data $\{x_i\} \subseteq \mathbb{S}^m$. Recall that in Section 3 we propose to estimate the location $\alpha$ and vectors $V$ using the Fréchet mean of $\{x_i\}$ and principal component analysis, respectively. The Fréchet mean of $\{x_i\}$ can be computed by a gradient method (Fletcher, 2013) or simply approximated by a heuristic estimator $\frac{\sum x_i}{\|\sum x_i\|}$. Suppose we have estimated $\alpha$ and $V$. It remains to estimate $\sigma$ as follows

$$\widehat{\sigma} = \operatorname*{argmax}_\sigma \frac{1}{Z(\sigma)} e^{-\frac{\mathrm{res}}{2\sigma^2}} = \operatorname*{argmin}_\sigma \log(Z(\sigma)) + \frac{\mathrm{res}}{2\sigma^2}, \tag{E.6}$$

where $\mathrm{res} > 0$ is a known average squared residual error. Taking derivative of the objective function in (E.6), we know

$$\ell(\sigma) := \frac{Z'(\sigma)}{Z(\sigma)} - \frac{\mathrm{res}}{\sigma^3}$$

contains $\widehat{\sigma}$ as a root. By (E.5), we know

$$\ell_1(\sigma) := \int_0^\pi e^{-\frac{r^2}{2\sigma^2}} \sin^{m-1}(r)r^2 dr - \mathrm{res} \cdot \int_0^\pi e^{-\frac{r^2}{2\sigma^2}} \sin^{m-1}(r)dr$$

also contains $\widehat{\sigma}$ as a root. Since for any given $\sigma$, $\ell_1(\sigma)$ can be efficiently computed by `quad` in Python, the root $\widehat{\sigma}$ can be estimated via a bisection method. When $\sigma$ or $\mathrm{res}$ is small, we can also use a variational method. In this case,

$$\ell_1(\sigma) \approx \int_0^\infty e^{-\frac{r^2}{2\sigma^2}} r^{m+1} dr - \mathrm{res} \cdot \int_0^\infty e^{-\frac{r^2}{2\sigma^2}} r^{m-1} dr$$

$$= 2^{\frac{m-2}{2}} \sigma^m \cdot \left( 2\sigma^2 \cdot \int_0^\infty e^{-t} t^{\frac{m}{2}} dt - \mathrm{res} \cdot \int_0^\infty e^{-t} t^{\frac{m}{2}-1} dt \right),$$

which implies that $\widehat{\sigma} \approx \sqrt{\frac{\mathrm{res}}{m}}$. This is an accurate approximation when $\widehat{\sigma} \le 0.05$.

### E.5. Hyperbolic spaces $\mathbb{H}^m$

This section studies hyperbolic spaces $\mathcal{M} = \mathbb{H}^m$. We first review the geometry of $\mathbb{H}^m$ in Section E.5.1. Then we delve into the implementation details of MGFA in later sections.

#### E.5.1. BASIC GEOMETRY OF $\mathbb{H}^m$

In this paper, we will use two equivalent models to represent hyperbolic space $\mathbb{H}^m$: hyperboloid model and Poincaré ball model. We use the hyperboloid model in the computation and the Poincaré ball model in visualization. Here we review hyperboloid model and refer readers to (Benedetti & Petronio, 1992) for transformation between these two models. In the hyperboloid model, $\mathbb{H}^m$ is an unit imaginary sphere in the Minkowski space. The Minkowski space is $\mathbb{R}^{m+1}$ endowed with the Minkowski bilinear form

$$\langle x, y \rangle_M = -x_0 y_0 + \sum_{i=1}^m x_i y_i.$$

The hyperboloid is given by the unit imaginary sphere

$$\mathbb{H}^m = \{x \in \mathbb{R}^{m+1} \mid \langle x, x \rangle_M = -1, x_0 > 0\}.$$

The tangent space to $\mathbb{H}^m$ at $x$ is

$$T_x \mathbb{H}^m = \{v \in \mathbb{R}^{m+1} \mid \langle x, v \rangle = 0\}.$$

Since the Minkowski bilinear form restricted to $T_x \mathbb{H}^m$ is an inner product, this gives the Riemannian metric on $\mathbb{H}^m$. Given a base $x \in \mathbb{H}^m$ and a tangent vector $v \in T_x \mathbb{H}^m$, the exponential map is given by

$$\mathrm{Exp}_x(v) = \cosh(\|v\|)x + \sinh(\|v\|)v/\|v\|.$$

Given $x, y \in \mathbb{H}^m$, the distance between $x$ and $y$ is $d(x, y) = \mathrm{arccosh}(-\langle x, y \rangle_M)$, and the logarithm map is

$$\mathrm{Log}_x(y) = d(x, y) \cdot (y - \cosh(d(x, y))x)/\sinh(d(x, y)).$$

The segment domain of $\mathrm{Exp}_\alpha$ is the whole tangent space. With the normal coordinate chart, the volume form can be written as

$$\mathrm{dvol} = \sinh^{m-1}(r)drd\Theta,$$

where $(r, \Theta) \in [0, \infty) \times \mathbb{S}^{m-1}$ is the polar coordinate. For any function $f(x) = h(d_g(x, \alpha))$ on $\mathbb{H}^m$, its integral is given by

$$\int_{\mathbb{H}^m} f(x)\mathrm{dvol}(x) = \mathrm{vol}(\mathbb{S}^{m-1}) \int_0^\infty f(r) \sinh^{m-1}(r)dr. \tag{E.7}$$

This is similar to the integral (E.4) in the spherical case. We will utilize this formula to design our data simulation, model estimation, and data clustering procedures.

*Table 3.* The mean (and standard deviation) of the RI and ARI for Ward clustering, movMF, skmeans, MoRN, and MGFA on $\mathbb{S}^m$ in various settings. Bold numbers indicate the best results in the corresponding settings.

| Setting | Criteria | Ward clustering | movMF | skmeans | MoRN | MGFA |
|---|---|---|---|---|---|---|
| $\mathbb{S}^2$ | RI | 0.65 (0.10) | 0.59 (0.08) | 0.73 (0.09) | 0.73 (0.09) | **0.93** (0.02) |
| | ARI | 0.29 (0.19) | 0.18 (0.16) | 0.45 (0.19) | 0.46 (0.17) | **0.86** (0.04) |
| $\mathbb{S}^{300}$ | RI | 0.54 (0.04) | 0.51 (0.02) | 0.50 (0.002) | 0.50 (0.002) | **0.92** (0.05) |
| | ARI | 0.07 (0.09) | 0.01 (0.04) | 0.002 (0.004) | 0.002 (0.004) | **0.85** (0.10) |
| multi-class | RI | 0.72 (0.03) | 0.70 (0.01) | 0.71 (0.01) | 0.71 (0.01) | **0.94** (0.04) |
| | ARI | 0.30 (0.05) | 0.24 (0.03) | 0.26 (0.02) | 0.25 (0.02) | **0.83** (0.10) |
| mis-spec - A | RI | 0.99 (0.006) | 0.97 (0.07) | **1.00** (0.001) | **1.00** (0.001) | 0.99 (0.03) |
| | ARI | 0.97 (0.01) | 0.94 (0.15) | **1.00** (0.002) | **1.00** (0.002) | 0.98 (0.06) |
| mis-spec - B | RI | 0.53 (0.05) | 0.51 (0.02) | 0.50 (0.004) | 0.50 (0.004) | **0.75** (0.03) |
| | ARI | 0.07 (0.09) | 0.02 (0.05) | 0.005 (0.008) | 0.005 (0.008) | **0.50** (0.07) |
| mis-spec - C | RI | 0.55 (0.06) | 0.51 (0.03) | 0.50 (0.005) | 0.50 (0.006) | **0.93** (0.05) |
| | ARI | 0.09 (0.11) | 0.02 (0.05) | 0.005 (0.01) | 0.007 (0.01) | **0.86** (0.10) |

### E.5.2. DATA GENERATION

In this section, we present a data sampling procedure based on inverse cumulative distribution function. Similar to the cases of spheres and shape spaces, the key challenge in simulation is to sample data from the Riemannian Gaussian distribution $RN(\alpha, \sigma)$. In this work, we propose to use the polar coordinate system on the normal coordinate chart, and then it suffices to sample $(r, \Theta)$ from the following density function:

$$p(r, \Theta) \propto e^{\frac{-r^2}{2\sigma^2}} \sinh^{m-1} r, \quad r \in [0, \infty), \Theta \in \mathbb{S}^{m-1}.$$

Since $r$ and $\Theta$ are independent, we can first sample $\Theta$ uniformly from the unit sphere $\mathbb{S}^{m-1}$ and then sample $r$ using the inverse cumulative distribution function method. Then we can combine $r$ and $\Theta$ to find the desired data. In practice, this is effective when the dimension $m$ is low.

### E.5.3. DATA ASSIGNMENT AND MODEL ESTIMATION

Similar to the spherical case, it suffices to compute the normalizing constant $Z(\sigma)$ for data assignment. We shall use the integral formula (E.7) and the python function `quad` for the computation. As for model estimation, we will use similar strategies as in Section E.4.4. In particular, when $\sigma$ is small, we will estimate $\sigma$ by $\widehat{\sigma} \approx \sqrt{\text{res}/m}$, same as the spherical case.

## F. Simulation on Spheres

We evaluate MGFA for spherical data clustering under diverse scenarios, comparing its performance with established methods including Ward clustering, mixture of von Mises-Fisher distributions (movMF) (Banerjee et al., 2005; Hornik & Grün, 2014), spherical $k$-means (skmeans) (Hornik et al., 2012), and MoRN. For each experiment, we measure clustering accuracy using the Rand index (RI) (Rand, 1971) and adjusted Rand index (ARI).

### F.1. 2-dimensional spherical data clustering

For visual clarity, we first cluster data points sampled from the MGFA model (3.7) on $\mathbb{S}^2$, with two groups and a single latent factor:

$$
\begin{array}{llll}
\text{(G1)} & \alpha_1 = (1, 0, 0), & v_1 = (0, 1, 0), & \sigma_1 = 0.05, \quad \omega_1 = 0.4, \\
\text{(G2)} & \alpha_2 = (-0.6, 0, -0.8), & v_2 = (0, 1, 0), & \sigma_2 = 0.1, \quad \omega_2 = 0.6.
\end{array}
$$

We draw $n = 500$ samples per trial and cluster them using each method. A representative result is depicted in Figure 2. The average RI and ARI scores over 100 trials (Table 3) demonstrate MGFA's superior capability in accurately identifying latent structures compared to other methods, with an RI of 0.93 versus $\leq 0.73$ for alternatives.

## F.2. High-dimensional spherical data clustering

We next assess MGFA performance on high-dimensional spheres ($\mathbb{S}^{300}$) using two clusters with one latent factor:

$$
\begin{aligned}
\text{(G1)} \quad & \alpha_1 = (1, 0, \ldots), & v_1 = (0, 0, 0.8, 0.8, 0, \ldots), & \quad \sigma_1 = 0.03, & \omega_1 = 0.4, \\
\text{(G2)} \quad & \alpha_2 = (0, 1, 0, \ldots), & v_2 = (0, 0, 0.8, 0.8, 0, \ldots), & \quad \sigma_1 = 0.03, & \omega_2 = 0.6.
\end{aligned}
$$

After 100 trials with $n = 1000$ samples per trial, MGFA consistently outperformed other methods with an RI of 0.92, significantly surpassing the best alternative RI ($\leq 0.54$) (Table 3).

## F.3. Multi-class clustering

In a multi-class setting on $\mathbb{S}^{100}$, we test MGFA with four distinct groups and a single latent factor:

$$
\begin{aligned}
\text{(G1)} \quad & \alpha_1 = (1, 0, \ldots), & v_1 = (0, 0, 0.8, 0.8, 0, \ldots), & \quad \sigma_1 = 0.05, & \omega_1 = 0.2, \\
\text{(G2)} \quad & \alpha_2 = (0, 1, 0, \ldots), & v_2 = (0, 0, 0.8, 0.8, 0, \ldots), & \quad \sigma_1 = 0.05, & \omega_2 = 0.3, \\
\text{(G3)} \quad & \alpha_3 = (0, 0, 1, 0, \ldots), & v_3 = (0, 0, 0, 0.8, 0.8, 0, \ldots), & \quad \sigma_1 = 0.03, & \omega_1 = 0.2, \\
\text{(G4)} \quad & \alpha_4 = (-0.6, -0.8, 0, \ldots), & v_4 = (0, 0, 0, 0.8, 0.8, 0, \ldots), & \quad \sigma_1 = 0.03, & \omega_2 = 0.3.
\end{aligned}
$$

MGFA again demonstrates superior performance with an average RI of 0.94, substantially outperforming alternative methods ($\leq 0.72$) across 100 trials (Table 3).

## F.4. Robustness in misspecified settings

In the last experiment, we examine the robustness of MGFA over a variety of misspecified models. In particular, we consider misspecified factor dimension in models A and B, and misspecified noise type in model C.

(A) We use the following MGFA model on $\mathbb{S}^{100}$ with $K = 3$ groups and zero latent factor:

$$
\begin{aligned}
\text{(G1)} \quad & \alpha_1 = (1, 0, \ldots), & v_1 = \mathbf{0}, & \quad \sigma_1 = 0.15, & \omega_1 = 0.2, \\
\text{(G2)} \quad & \alpha_2 = (0, 1, 0, \ldots), & v_2 = \mathbf{0}, & \quad \sigma_2 = 0.15, & \omega_2 = 0.3, \\
\text{(G3)} \quad & \alpha_3 = (0, 0, 1, 0, \ldots), & v_3 = \mathbf{0}, & \quad \sigma_3 = 0.15, & \omega_3 = 0.5.
\end{aligned}
$$

(B) We use the following MGFA model on $\mathbb{S}^{100}$ with $K = 2$ groups and two latent factors:

$$
\begin{aligned}
\text{(G1)} \quad & \alpha_1 = (1, 0, \ldots), & v_{11} = (0, 0, 0.6, 0.6, 0, \ldots), & \quad \sigma_1 = 0.03, & \omega_1 = 0.4, \\
& & v_{12} = (0, 0, 0, 0, 0.6, 0.6, 0, \ldots), & & \\
\text{(G2)} \quad & \alpha_2 = (0, 1, 0, \ldots), & v_{21} = (0, 0, 0.6, 0.6, 0, \ldots), & \quad \sigma_2 = 0.03, & \omega_2 = 0.6, \\
& & v_{22} = (0, 0, 0, 0.6, 0.6, 0, \ldots). & &
\end{aligned}
$$

(C) We use the following MGFA model on $\mathbb{S}^{100}$ with $K = 2$ groups and one latent factor:

$$
\begin{aligned}
\text{(G1)} \quad & \alpha_1 = (1, 0, \ldots), & v_1 = (0, 0, 0.8, 0.8, 0, \ldots), & \quad \kappa_1 = 1000, & \omega_1 = 0.4, \\
\text{(G2)} \quad & \alpha_2 = (0, 1, 0, \ldots), & v_2 = (0, 0, 0.8, 0.8, 0, \ldots), & \quad \kappa_2 = 1000, & \omega_2 = 0.6,
\end{aligned}
$$

where we use vMF distributions rather than Riemannian Gaussian distributions for noise modeling in MGFA, and $\kappa_1, \kappa_2$ are the concentration parameters of vMF distributions.

For each of these models, we draw $n = 500$ samples and then cluster them into $K$ groups using MGFA with $q = 1$, MoRN, Ward clustering, movMF, and skmeans. We run the experiments for 100 times and report the average RIs and ARIs of different clustering methods in Table 3. The results show that when the actual factor dimension is zero (A), all methods work very well including the misspecified MGFA with $q = 1$. In addition, when the actual factor dimension is larger than 1 (B), the MGFA with $q = 1$, though misspecified, still achieves better performance than all the rest methods. Moreover, the MGFA method is robust to the noise type as demonstrated in case C. Overall, these results demonstrate the robustness of MGFA under different factor dimensions and noise types.

*Table 4.* The mean (and standard deviation) of RIs and ARIs for Ward clustering, MoRN, and MGFA on $\mathbb{CP}^m$ in various settings. Numbers in bold indicate the best results in the corresponding settings.

| Setting | Criteria | Ward clustering | MoRN | MGFA |
|---------|----------|-----------------|------|------|
| $\mathbb{CP}^3$ | RI | 0.70 (0.08) | 0.59 (0.10) | **0.89** (0.06) |
| | ARI | 0.40 (0.16) | 0.18 (0.20) | **0.79** (0.12) |
| multi-class | RI | 0.86 (0.05) | 0.75 (0.05) | **0.95** (0.03) |
| | ARI | 0.70 (0.11) | 0.46 (0.09) | **0.89** (0.06) |
| mis-spec - A | RI | 0.86 (0.03) | **0.98** (0.01) | 0.97 (0.01) |
| | ARI | 0.68 (0.06) | **0.95** (0.02) | 0.93 (0.02) |
| mis-spec - B | RI | 0.67 (0.07) | 0.54 (0.07) | **0.71** (0.04) |
| | ARI | 0.33 (0.15) | 0.08 (0.15) | **0.43** (0.08) |
| mis-spec - C | RI | 0.86 (0.07) | 0.94 (0.06) | **0.96** (0.02) |
| | ARI | 0.73 (0.13) | 0.88 (0.13) | **0.93** (0.04) |

# G. Simulation on Shape spaces $\mathbb{CP}^m$

This subsection investigates data clustering on shape spaces $\mathbb{CP}^m$ under various scenarios, comparing MGFA with two alternative approaches that do not incorporate factor structures: Ward clustering and MoRN. To evaluate and compare the performance of these methods, we compute RI and ARI.

## G.1. Clustering in $\mathbb{CP}^3$

In the first experiment, we examine binary clustering in $\mathbb{CP}^3$. Specifically, we consider the following MGFA model with two groups and one factor dimension:

$$\text{(G1)} \quad \alpha_1 = (1,0,0,0) \in \mathbb{C}^4, \quad v_1 = (0,0.7,0.7,0) \in \mathbb{C}^4, \quad \sigma_1 = 0.05, \quad \omega_1 = 0.4,$$
$$\text{(G2)} \quad \alpha_2 = (0,0,0,1) \in \mathbb{C}^4, \quad v_2 = (0,0.7,0.7,0) \in \mathbb{C}^4, \quad \sigma_2 = 0.05, \quad \omega_2 = 0.6,$$

where the equivalent class $[\alpha_k] = \{c \cdot \alpha_k \mid c \in \mathbb{S}^1\} \in \mathbb{CP}^3$ is the base point and $\bar{v}_k \in T_{[\alpha_k]}\mathbb{CP}^3$ is the tangent vector associated with $v_k$. Readers may refer to Section E.3.1 for a review of these concepts. In each trial, we draw $n = 500$ samples from the MGFA model, and then apply MGFA with $q = 1$, Ward clustering, and MoRN to produce two clusters. The average RIs and ARIs across 100 repeated trials are reported in Table 4. The results demonstrate that MGFA, with an average RI (0.89), significantly outperforms Ward clustering and MoRN, whose average RIs are at most 0.7.

## G.2. Multi-class clustering in $\mathbb{CP}^{30}$

In the second experiment, we focus on multi-class clustering on $\mathbb{CP}^{30}$, and consider the following MGFA model with three groups and one latent factor, whose parameters $\{[\alpha_k], \bar{v}_k, \sigma_k, \omega_k\}_{k=1}^K$ are given by

$$\text{(G1)} \quad \alpha_1 = (1,0,\ldots) \in \mathbb{C}^{31}, \quad v_1 = (0,0,0.7,0,\ldots) \in \mathbb{C}^{31}, \quad \sigma_1 = 0.05, \quad \omega_1 = 0.4,$$
$$\text{(G2)} \quad \alpha_2 = (0,1,0,\ldots) \in \mathbb{C}^{31}, \quad v_2 = (0,0,0.7,0,\ldots) \in \mathbb{C}^{31}, \quad \sigma_2 = 0.05, \quad \omega_2 = 0.3,$$
$$\text{(G3)} \quad \alpha_3 = (0,0,1,0,\ldots) \in \mathbb{C}^{31}, \quad v_3 = (0,0,0,0.7,0,\ldots) \in \mathbb{C}^{31}, \quad \sigma_3 = 0.05, \quad \omega_3 = 0.3,$$

Here $[\alpha_k] \in \mathbb{CP}^{30}$ stands for the equivalent class $\{c \cdot \alpha_k \mid c \in \mathbb{S}^1\}$ and $\bar{v}_k \in T_{[\alpha_k]}\mathbb{CP}^{30}$ is the tangent vector associated with $v_k$. In each trial, we generate $n = 500$ samples from the model and then cluster them into three groups using MGFA with $q = 1$, Ward clustering, and MoRN. We repeat the experiment for 100 times and calculate the mean and standard deviation of the RIs and ARIs for these three methods. The results, as displayed in Table 4, show the superiority of MGFA over Ward clustering and MoRN.

## G.3. Misspecified models

In the third experiment, we examine the robustness of MGFA across diverse misspecified settings: A and B misspecify the factor dimension and C misspecifies the noise type.

(A) We use the following MGFA model on $\mathbb{CP}^{30}$ with $K = 3$ groups and zero factor:

$$
\begin{array}{llll}
\text{(G1)} & \alpha_1 = (1, 0, \ldots) \in \mathbb{C}^{31}, & v_1 = \mathbf{0}, \quad \sigma_1 = 0.2, \quad \omega_1 = 0.4, \\
\text{(G2)} & \alpha_2 = (0, 1, 0, \ldots) \in \mathbb{C}^{31}, & v_2 = \mathbf{0}, \quad \sigma_2 = 0.2, \quad \omega_2 = 0.3, \\
\text{(G3)} & \alpha_3 = (0, 0, 1, 0, \ldots) \in \mathbb{C}^{31}, & v_3 = \mathbf{0}, \quad \sigma_3 = 0.2, \quad \omega_3 = 0.3.
\end{array}
$$

Here the equivalent class $[\alpha_k] = \{c \cdot \alpha_k \mid c \in \mathbb{S}^1\} \in \mathbb{CP}^{30}$ denotes the base point.

(B) We use the following MGFA model on $\mathbb{CP}^{30}$ with $K = 2$ groups and two factors:

$$
\begin{array}{llll}
\text{(G1)} & \alpha_1 = (1, 0, \ldots) \in \mathbb{C}^{31}, & v_{11} = (0, 0, 1, 0, \ldots) \in \mathbb{C}^{31} & \sigma_1 = 0.05, \quad \omega_1 = 0.4, \\
& & v_{21} = (0, 0, 0, 0.5, 0, \ldots) \in \mathbb{C}^{31}, & \\
\text{(G2)} & \alpha_2 = (0, 1, 0, \ldots) \in \mathbb{C}^{31}, & v_{21} = (0, 0, 1, 0, \ldots) \in \mathbb{C}^{31}, & \sigma_2 = 0.05, \quad \omega_2 = 0.6, \\
& & v_{22} = (0, 0, 0, 0.5, 0, \ldots) \in \mathbb{C}^{31}, &
\end{array}
$$

where $[\alpha_k] = \{c \cdot \alpha_k \mid c \in \mathbb{S}^1\}$ denotes the base point.

(C) We use the following MGFA model on $\mathbb{CP}^{50}$ with $K = 2$ groups and one factor:

$$
\begin{array}{llll}
\text{(G1)} & \alpha_1 = (1, 0, \ldots) \in \mathbb{C}^{51}, & v_1 = (0, 0, 0.7, 0, \ldots) \in \mathbb{C}^{51}, & \sigma_1 = 0.005, \quad \omega_1 = 0.4, \\
\text{(G2)} & \alpha_2 = (0, 1, 0, \ldots) \in \mathbb{C}^{51}, & v_2 = (0, 0, 0.7, 0, \ldots) \in \mathbb{C}^{51}, & \sigma_2 = 0.005, \quad \omega_2 = 0.3.
\end{array}
$$

where we use Riemannian Laplacian distributions rather than Riemannian Gaussian distributions for noise modeling in MGFA, and $\sigma_k$ is the parameter of Riemannian Laplacian distributions.

For each setting, we generate $n = 500$ samples from the specified model and then cluster the data into $K$ groups using MGFA with $q = 1$, Ward clustering, and MoRN. Each experiment is repeated 100 times, and we report the mean and standard deviation of the RIs and ARIs in Table 4. The results indicate that, for model A, the correctly specified MoRN method achieves the best performance, while MGFA with $q = 1$ remains competitive, demonstrating robustness despite the misspecified factor dimension. For model B, MGFA outperforms the other methods, as its modeling assumptions closely align with the true underlying structure. Lastly, in model C, MGFA again achieves superior performance, highlighting its robustness to misspecifications in the noise distribution.

# H. Simulation on hyperbolic spaces $\mathbb{H}^m$

This section investigates data clustering on hyperbolic spaces $\mathbb{H}^m$, comparing MGFA with two alternative methods—Ward clustering and MoRN—that do not account for factor structures. To assess the performance of each method, we use RI and ARI as evaluation metrics.

## H.1. 2-dimensional hyperbolic data clustering

For the purpose of visualization, we first consider data clustering on $\mathbb{H}^2$. In this experiment, we consider the following MGFA model with two groups and one latent factor:

$$
\begin{array}{llll}
\text{(G1)} & \alpha_1 = (3, 2, 2), & v_1 = (2, 1, 2), & \sigma_1 = 0.1, \quad \omega_1 = 0.4, \\
\text{(G2)} & \alpha_2 = (1, 0, 0), & v_2 = (0, 1, 0), & \sigma_2 = 0.1, \quad \omega_2 = 0.6,
\end{array}
$$

where $\alpha_k \in \{x \in \mathbb{R}^3 \mid x_1^2 = 1 + x_2^2 + x_3^2, x_1 > 0\}$ uses the hyperboloid representation and $v_k \in T_{\alpha_k}\mathbb{H}^2 = \{v \in \mathbb{R}^3 \mid v_1 x_1 = v_2 x_2 + v_3 x_3\}$ is the tangent vector. In a single trial, we draw $n = 100$ samples from the model and then cluster them into two groups using MGFA with $q = 1$, MoRN, or Ward clustering. The visualization of the clustering results is displayed in Figure 5. For each method, we compute the RI and ARI after clustering. The average RI and ARI across 100 repeated trials are shown in Table 5. As displayed in Figure 5, MGFA successfully recovers the latent factor structure while the other two methods fail to recover the hidden factor pattern. As a result, MGFA achieves a significantly higher RI (0.99) than its competitors ($\leq 0.76$).

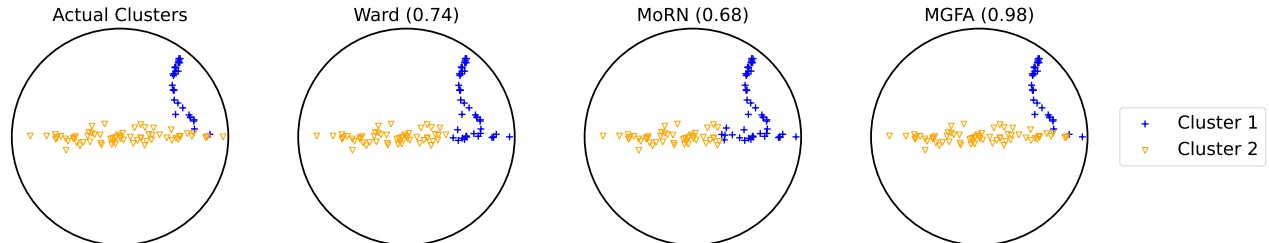

*Figure 5.* Visualization of clustering results on $\mathbb{H}^2$. The first panel displays the actual clusters, while the rest panels display clustering results for Ward clustering, MoRN and MGFA, respectively. The RIs for these methods are provided in parentheses. The data are displayed using Poincaré representation.

*Table 5.* The mean (and standard deviation) of RIs and ARIs for Ward clustering, MoRN, and MGFA on $\mathbb{H}^m$ in various settings. Numbers in bold indicate the best results in the corresponding settings.

| Setting | Criteria | Ward clustering | MoRN | MGFA |
|---------|----------|-----------------|------|------|
| $\mathbb{H}^2$ | RI | 0.76 (0.12) | 0.71 (0.10) | **0.99** (0.01) |
| | ARI | 0.51 (0.25) | 0.41 (0.20) | **0.98** (0.03) |
| multi-class | RI | 0.76 (0.04) | 0.75 (0.03) | **0.90** (0.06) |
| | ARI | 0.49 (0.09) | 0.47 (0.08) | **0.78** (0.12) |
| mis-spec - A | RI | 0.90 (0.08) | **0.95** (0.03) | 0.89 (0.05) |
| | ARI | 0.80 (0.15) | **0.89** (0.07) | 0.79 (0.11) |
| mis-spec - B | RI | 0.62 (0.08) | 0.64 (0.08) | **0.70** (0.09) |
| | ARI | 0.23 (0.17) | 0.28 (0.17) | **0.39** (0.19) |
| mis-spec - C | RI | 0.82 (0.23) | 0.70 (0.21) | **0.99** (0.07) |
| | ARI | 0.64 (0.46) | 0.40 (0.42) | **0.97** (0.14) |

## H.2. Multi-class clustering on $\mathbb{H}^{10}$

In the second experiment, we consider multi-class clustering on $\mathbb{H}^{10}$, and examine the following MGFA model with three groups and one latent factor:

$$
\begin{aligned}
\text{(G1)} \quad &\alpha_1 = (3, 2, 2, 0, \ldots) \in \mathbb{R}^{11}, & v_1 = (2, 1, 2, 0, \ldots) \in \mathbb{R}^{11}, & \quad \sigma_1 = 0.1, \quad \omega_1 = 0.4, \\
\text{(G2)} \quad &\alpha_2 = (1, 0, \ldots) \in \mathbb{R}^{11}, & v_2 = (0, 1, 0, \ldots) \in \mathbb{R}^{11}, & \quad \sigma_2 = 0.1, \quad \omega_2 = 0.3, \\
\text{(G3)} \quad &\alpha_3 = (2, -\sqrt{2}, -1, 0, \ldots) \in \mathbb{R}^{11}, & v_3 = (1, 0, -2, 0, \ldots) \in \mathbb{R}^{11}, & \quad \sigma_3 = 0.1, \quad \omega_3 = 0.3,
\end{aligned}
$$

Here $\alpha_k \in \{x \in \mathbb{R}^{11} \mid x_1^2 = 1 + \sum_{i \geq 2} x_i^2, x_1 > 0\}$ uses the hyperboloid representation and $v_k \in T_{\alpha_k} \mathbb{H}^{10} = \{v \in \mathbb{R}^{11} \mid v_1 x_1 = \sum_{i \geq 2} v_i x_i\}$ denotes the tangent vector. For any single trial, we generate $n = 100$ data points from the MGFA model, and then cluster data into three clusters using MGFA with $q = 1$, MoRN, and Ward clustering. The average RIs and ARIs across 100 repeated trials are reported in Table 5. The average RI of MGFA (0.90) is significantly greater than the average RIs of the other methods ($\leq 0.76$), demonstrating the superiority of MGFA in multi-class clustering.

## H.3. Robustness in misspecified settings

In the third experiment, we examine the robustness of MGFA across diverse misspecified settings: A and B misspecify the factor dimension and C misspecifies the noise type.

(A) We use the following MGFA model on $\mathbb{H}^{10}$ with two groups and zero factor:

$$
\begin{aligned}
\text{(G1)} \quad &\alpha_1 = (3, 2, 2, 0, \ldots) \in \mathbb{R}^{11}, & v_1 = (2, 1, 2, 0, \ldots) \in \mathbb{R}^{11}, & \quad \sigma_1 = 0.5, \quad \omega_1 = 0.4, \\
\text{(G2)} \quad &\alpha_2 = (2, -\sqrt{2}, -1, 0, \ldots) \in \mathbb{R}^{11}, & v_2 = (1, 0, -2, 0, \ldots) \in \mathbb{R}^{11}, & \quad \sigma_2 = 0.5, \quad \omega_2 = 0.6,
\end{aligned}
$$

where $\alpha_k \in \{x \in \mathbb{R}^{11} \mid x_1^2 = 1 + \sum_{i \geq 2} x_i^2, x_1 > 0\}$ uses the hyperboloid representation.

(B) We use the following MGFA model on $\mathbb{H}^{10}$ with two groups and two factors:

(G1)    $\alpha_1 = (3, 2, 2, 0, ...) \in \mathbb{R}^{11},$      $v_{11} = (4, 2, 4, 0, ...) \in \mathbb{R}^{11},$      $\sigma_1 = 0.1,$   $\omega_1 = 0.4,$

          $v_{12} = (0, \frac{1}{2}, -\frac{1}{2}, \frac{1}{2}, 0, ...) \in \mathbb{R}^{11}$

(G2)    $\alpha_2 = (2, -\sqrt{2}, -1, 0, ...) \in \mathbb{R}^{11},$   $v_{21} = (2, 0, -4, 0, ...) \in \mathbb{R}^{11},$    $\sigma_2 = 0.1,$   $\omega_2 = 0.6,$

          $v_{22} = (0, 0, \frac{1}{2}, \frac{1}{2}, 0, ...) \in \mathbb{R}^{11}$

where $\alpha_k \in \{x \in \mathbb{R}^{11} \mid x_1^2 = 1 + \sum_{i \geq 2} x_i^2, x_1 > 0\}$ uses the hyperboloid representation.

(C) We use the following MGFA model on $\mathbb{H}^{10}$ with two groups and one factor:

(G1)    $\alpha_1 = (3, 2, 2, 0, \ldots) \in \mathbb{R}^{11},$      $v_1 = (2, 1, 2, 0, \ldots) \in \mathbb{R}^{11},$    $\sigma_1 = 0.005,$   $\omega_1 = 0.4,$

(G2)    $\alpha_2 = (2, -\sqrt{2}, -1, 0, \ldots) \in \mathbb{R}^{11},$   $v_2 = (1, 0, -2, 0, \ldots) \in \mathbb{R}^{11},$   $\sigma_2 = 0.005,$   $\omega_2 = 0.6,$

where we use Riemannian Laplacian distributions rather than Riemannian Gaussian distributions for noise modeling in MGFA, and $\sigma_k$ is the parameter for Riemannian Laplacian distributions.

For each setting, we generate $n = 100$ samples and cluster them into two groups using MGFA with $q = 1$, MoRN, and Ward clustering. The average RI and ARI across 100 repetitions are presented in Table 5. In Model A, MoRN achieves the highest performance as it correctly matches the underlying data distribution, while MGFA also attains a comparably high accuracy, highlighting its robustness despite model misspecification. In Model B, MGFA yields the highest RI (0.70), outperforming both Ward clustering and MoRN (RI $\leq$ 0.64), further confirming MGFA's resilience to misspecifications in factor dimensionality. Similarly, in Model C, MGFA significantly surpasses the alternatives, achieving an RI of 0.99, compared to 0.82 or lower for Ward clustering and MoRN, underscoring MGFA's robustness against noise-type misspecification.

# I. Real data analysis

This paper conduct real data analysis for ADNI research. As the aging population grows, dementia—particularly Alzheimer's disease (AD)—has become a significant societal challenge (Prince et al., 2015; Mirzaei & Adeli, 2022). Despite substantial research, the cause of AD remains unclear, making early diagnosis and intervention critical (Mirzaei & Adeli, 2022; Rasmussen & Langerman, 2019). Mild cognitive impairment (MCI), an intermediate stage between normal aging and AD, is a key target for early detection (Petersen, 2016). Advances in neuroimaging and cerebrospinal fluid (CSF) biomarkers increasingly support accurate MCI diagnosis (Frisoni et al., 2017). However, the inherent heterogeneity of MCI poses challenges for precise diagnosis and effective treatment (Alashwal et al., 2019). Our objective is to use novel clustering methods in AD-related research.

## I.1. Corpus callosal shape analysis

We analyze the corpus callosal shape dataset from the ADNI study. The corpus callosum (CC), the largest white matter structure in the brain, connects the cerebral hemispheres and plays a critical role in brain function, making it a focus of neuroimaging studies (Paul et al., 2007). Examining CC shape provides insights into neurological disorders and aids in developing therapeutic approaches. To address this, we use MGFA to cluster MCI subjects into more homogeneous subgroups based on CC shapes and analyze cluster features, including cognitive abilities, CSF biomarkers, and demographics. Given known sex differences in MCI (Au et al., 2017; Li & Singh, 2014; Laws et al., 2016; Ferretti et al., 2018), we stratify the cohort by sex and apply MGFA separately to female and male groups to explore sex-specific patterns. Section I.1.1 introduces CC shape geometry for MGFA. We then perform clustering on late MCI (LMCI) subjects, analyzing female and male cohorts separately (Sections I.1.2 and I.1.3), followed by a comparative analysis.

### I.1.1. KENDALL'S SHAPE SPACES

The Kendall's shape space is used to model the shapes of planar contours (Kendall, 1984), where the shape means the information that remains when location, scale, and orientation are removed (Dryden & Mardia, 2016). For any contour in $\mathbb{R}^2$, it can be represented by a configuration of $m$ points in $\mathbb{R}^2$, or a complex $m$-vector, $y \in \mathbb{C}^m$. Location information can be removed by pre-multiplying by the Helmert submatrix $H \in \mathbb{R}^{(m-1) \times m}$, $y_H = Hy \in \mathbb{C}^{m-1}$, where the $j$-th row of $H$ is given by

$$(h_j, \ldots, h_j, -jh_j, 0, \ldots, 0), \quad h_j = -j(j+1)^{-1/2}, \tag{I.1}$$

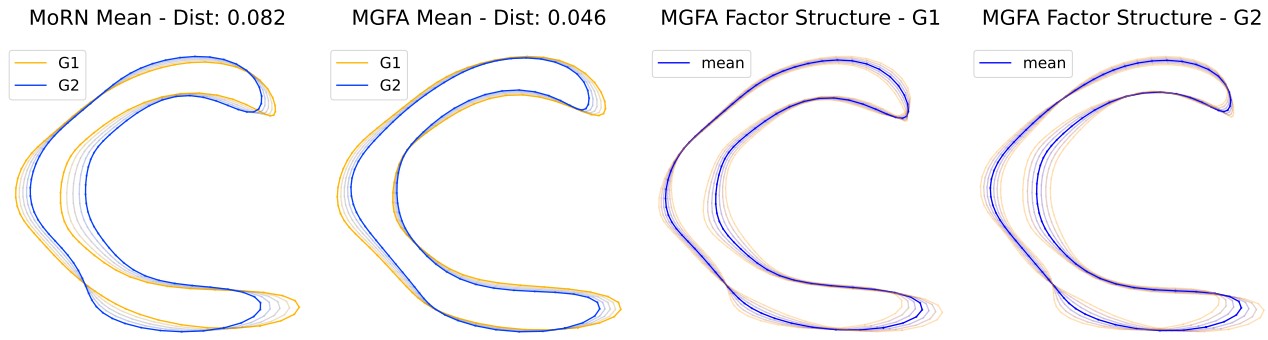

*Figure 6.* Clustering results for the female cohort. The first two panels display the mean CC shapes of two groups identified by MoRN and MGFA, along with the geodesics between these mean shapes. The geodesic distances between mean shapes are 0.082 for MoRN and 0.046 for MGFA. The rest two panels visualize the factor structures of two groups identified by MGFA. The thick lines represent the mean CC shapes, while the thin lines stand for $\mathrm{Exp}(\alpha, \mathrm{length} \cdot v)$, where $\alpha$ is the mean shape, $v$ is a unit tangent vector modeling the factor structure, $\mathrm{length}$ is chosen from $[-l, l]$ equidistantly, and $l$ is half the maximum norm of estimated latent factors.

*Table 6.* FDR-adjusted p-values for subgroup differences in the female LMCI cohort.

| | AGE | EDU | $A\beta$ | PTAU | TAU | ADAS11 | ADAS13 |
|---|---|---|---|---|---|---|---|
| MoRN | 0.635 | 0.442 | 0.442 | 0.052 | 0.094 | 0.690 | 0.794 |
| MGFA | **0.000** | 0.631 | 0.855 | 0.806 | 0.690 | 0.631 | 0.549 |

| | MMSE | immediate | learning | percent forgetting | ST2SV | ST3SV | ST4SV |
|---|---|---|---|---|---|---|---|
| MoRN | 0.638 | 0.962 | 0.690 | 0.794 | 0.613 | 0.442 | 0.503 |
| MGFA | 0.631 | 0.631 | 0.351 | 0.893 | **0.028** | **0.013** | **0.008** |

| | ST5SV | ST6SV | ST103CV | ST88SV | ST44CV | ST29SV | ICV |
|---|---|---|---|---|---|---|---|
| MoRN | 0.442 | 0.442 | 0.535 | 0.442 | 0.613 | 0.442 | 0.094 |
| MGFA | **0.013** | **0.028** | 0.855 | 0.193 | 0.879 | 0.549 | 0.855 |

Bold numbers represent statistically significant differences ($p < 0.05$).

where $h_j$ are repeated $j$ times. The scale information is then removed by normalizing $y_H$, leading to $x = y_H / \|y_H\| \in \mathbb{CS}^{m-2}$. Finally, the orientation information can be removed by taking the equivalent class $[x] = \{c \cdot x \mid c \in \mathbb{C}\} \in \mathbb{CP}^{m-2}$. Consequently, a planar contour's shape is conceptualized as a point in a shape space, and the associated shape analysis simplifies to data analysis in a shape space.

### I.1.2. CLUSTERING FEMALE LMCI SUBJECTS

In this section, we use MGFA to uncover meaningful subgroups within the female LMCI cohort. After excluding those with missing data, our final sample consists of 46 female LMCI subjects, aged 70.89 years on average (range: 55.1 - 84.0 years) and with an average education length of 15.34 years (range: 8.0-20.0 years). For each subject, we apply `FreeSurfer` (Fischl, 2012) to process T1-weighted MRI and then use `CCseg` (Vachet et al., 2012) to extract the planar CC contour data on the midsagittal slice. The output CC contour is given as a configuration of 100 points in $\mathbb{R}^2$, and we will model its shape in the Kendall's shape spaces, as introduced in Section I.1.1. Besides the CC shape and demographic information, each subject is further characterized by brain measures (cf. Table 8, Appendix I.2.1), CSF variables ($A\beta$, TAU, and PTAU), and cognitive test scores, including the Mini-Mental State Examination (MMSE) (Tombaugh & McIntyre, 1992), Alzheimer's Disease Assessment Scale (ADAS) subscores (ADAS13 and ADAS11) (Kueper et al., 2018), and Rey Auditory Verbal Learning Test (RAVLT) (Rey, 1958). Our objective is to stratify the female LMCI cohort into distinct subgroups and examine the subgroup differences in characteristics and within-group variable correlations.

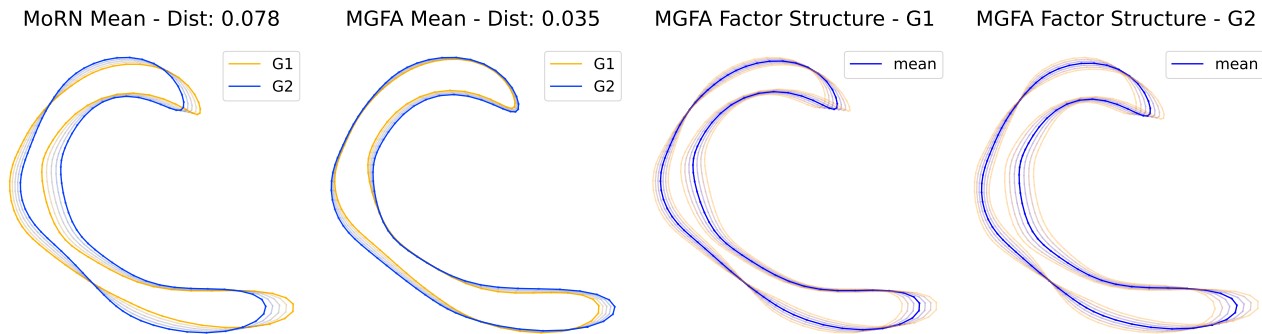

*Figure 7.* Clustering results for male LMCI subjects. The first two panels display the mean CC shapes of two groups identified by MoRN and MGFA, and visualize the geodesics between these mean shapes. The geodesic distances between mean shapes are 0.078 for MoRN and 0.035 for MGFA. The rest two panels visualize the factor structures of two groups identified by MGFA. The thick lines represent the mean CC shapes, while the thin lines stand for $\text{Exp}(\alpha, \text{length} \cdot v)$, where $\alpha$ is the mean shape, $v$ is a unit tangent vector modeling the factor structure, $\text{length}$ is chosen from $[-l, l]$ equidistantly, and $l$ is half the maximum norm of estimated latent factors.

Our study explores two methods for clustering the female LMCI cohort: MoRN, and MGFA with a latent dimension of $q = 1$. We use these methods to produce two clusters, yielding sizes of 25 and 21 for MoRN, and 21 and 25 for MGFA. Figure 6 visualizes the mean CC shapes and connecting geodesics for each subgroup, revealing that MoRN-identified subgroups exhibit larger differences in mean CC shapes compared to MGFA-identified subgroups, primarily in the trunk and splenium regions. The reduced between-cluster variations in MGFA can be attributed to the within-cluster factor structure, implying that the differences between clusters may be explained by the underlying factors. The third and fourth panels in Figure 6 further illustrate this point, showing a stronger factor structure in the second MGFA-identified subgroup, particularly in the trunk and splenium regions.

Next, we compare the subgroup differences in age, education length, brain measures, CSF variables, and cognitive scores, and use the Benjamini-Hochberg procedure (Benjamini & Hochberg, 1995) to control the false discovery rate (FDR). As shown in Table 6, MoRN-identified groups show significant difference in TAU and PTAU, while MGFA-identified clusters exhibit significant differences in age and sub-cortical volumes of the CC, including anterior, central, mid-anterior, mid-posterior, and posterior CC regions. To further characterize these subgroups, we examine the correlations between CSF variables and cognitive test scores, aiming to identify potential patterns within individual subgroup. Figure 8 in the Appendix visualizes these correlations and FDR-adjusted p-values in heatmaps. Given the significance level $0.05$, we observe the following.

- The first MoRN-identified cluster exhibits significant correlations between $A\beta$ and several cognitive test scores (MMSE, RAVLT percent forgetting, and RAVLT immediate). This cluster also shows significant correlations between RAVLT percent forgetting and two other CSF variables (TAU and PTAU). In contrast, the second MoRN-identified cluster does not show any significant correlations between CSF variables and cognitive test scores.

- The first MGFA-identified cluster shows significant correlations between $A\beta$ and all cognitive scores (MMSE, ADAS13, ADAS11, RAVLT percent forgetting, RAVLT immediate, and RAVLT learning). The second MGFA-identified cluster does not show any significant correlations between CSF variables and cognitive test scores.

These findings suggest that both MoRN and MGFA identify two distinct subgroups: one with significant correlations between CSF biomarkers and cognitive performance, and another without. The first subgroup, identified by both models, comprises individuals more sensitive to CSF variables, which may indicate a higher risk of AD progression (Wolk et al., 2009; Hansson et al., 2006; Blennow, 2004). A key distinction between MoRN and MGFA lies in the specific correlations within this subgroup. MGFA reveals significant associations between $A\beta$ and multiple cognitive scores (ADAS11, ADAS13, RAVLT learning), whereas MoRN identifies significant correlations between RAVLT percent forgetting and tau pathology (TAU and PTAU). Notably, MGFA, which accounts for a one-dimensional latent factor, better captures the effects of $A\beta$ accumulation, a major pathogenic event in Alzheimer's disease (Haass & Selkoe, 1993). These results underscore the importance of incorporating latent factor structures to enhance the understanding of biomarker-cognition relationships.

*Table 7.* FDR-adjusted p-values for subgroup differences in the male LMCI cohort.

| | AGE | EDU | $A\beta$ | PTAU | TAU | ADAS11 | ADAS13 |
|---|---|---|---|---|---|---|---|
| MoRN | 0.700 | 0.477 | 0.477 | **0.007** | **0.007** | 0.528 | 0.670 |
| MGFA | 0.751 | 0.751 | 0.751 | 0.388 | 0.552 | 0.927 | 0.751 |
| | MMSE | immediate | learning | percent forgetting | ST2SV | ST3SV | ST4SV |
| MoRN | 0.840 | 0.477 | 0.533 | 0.477 | 0.931 | **0.032** | 0.233 |
| MGFA | 0.751 | 0.126 | 0.938 | 0.751 | 0.388 | 0.239 | 0.382 |
| | ST5SV | ST6SV | ST103CV | ST88SV | ST44CV | ST29SV | ICV |
| MoRN | 0.273 | 0.059 | 0.784 | 0.477 | 0.784 | 0.670 | **0.007** |
| MGFA | 0.258 | 0.126 | 0.388 | 0.938 | 0.938 | 0.751 | 0.751 |

Bold numbers represent statistically significant differences ($p < 0.05$).

### I.1.3. CLUSTERING MALE LMCI SUBJECTS

We now apply MGFA to identify meaningful subgroups within the male LMCI cohort. After excluding individuals with missing values, our final sample comprises 105 male LMCI subjects, characterized by an average age of 75.82 years (range: 54.4-88.3 years) and an average education length of 16.20 years (range: 6.0-20.0 years). Using the same procedure as for the female cohort, we extract planar CC contour data for each male subject, generating a 100-point configuration in $\mathbb{R}^2$ that is subsequently mapped to Kendall's shape space. We then stratify the male LMCI cohort into two distinct subgroups and examine subgroup differences, using the same approach as for the female cohort.

Our study still examines two models for clustering the male LMCI cohort: MoRN and MGFA with a latent dimension of $q = 1$. We apply these models to stratify data into two subgroups, resulting in sizes of 56 and 49 for MoRN, and 63 and 42 for MGFA. We visualize the mean CC shapes and connecting geodesics for each subgroup in Figure 7, indicating that MGFA-identified subgroups have more similar mean CC shapes compared to MoRN-identified subgroups. This reduced between-cluster variation in MGFA can be attributed to the effects of underlying latent factors. The last two panels in Figure 9 show the factor structure within each MGFA-identified cluster. It turns out that the second subgroup exhibits a stronger factor structure than the first subgroup, particularly in the trunk and splenium regions.

Next, we examine the subgroup differences in age, education length, brain measures, CSF variables, and cognitive scores, employing the Benjamini-Hochberg correction to control the false discovery rate. As shown in Table 7, MGFA-identified subgroups do not display significant difference in these characteristics. In contrast, MoRN-identified subgroups show significant difference in TAU, PTAU, brain size (ICV), and the subcortical volume of the central CC. To further elucidate subgroup differences, we examine the correlations between CSF variables and cognitive test scores, apply the Benjamini-Hochberg procedure to adjust the p-values for multiple comparisons, and identify significant pairs. These adjusted p-values are visualized using heatmaps in Figure 9 in the Appendix. Given the significance level 0.05, we find the following significant correlations:

- The second MoRN-identified cluster exhibits significant correlations between $A\beta$ and ADAS13, whereas the first cluster shows no significant correlations between CSF variables and cognitive test scores.

- The second MGFA-identified subgroup shows significant correlations between RAVLT percentage forgetting and tau pathology (TAU, PTAU). The first MGFA-identified subgroup exhibits significant correlations between $A\beta$ and ADAS test scores (ADAS11, ADAS13).

These findings show that both MoRN and MGFA identify one subgroup that exhibit significant correlations between $A\beta$ and cognitive scores (ADAS13). However, the subgroup identified by MGFA also show significant correlations between $A\beta$ and ADAS11, better capturing the effects of $A\beta$ accumulation. Moreover, the other MoRN-identified subgroup does not show significant correlations between CSF variables and cognitive performance, while the other MGFA-identified cluster exhibits

significant associations between tau pathology and memory performance. These results suggest that MGFA may be more sensitive to the nuances of A$\beta$ and tau pathology, key features of AD progression (Blennow, 2004).

Finally, we compare the female and male cohorts, examining their corresponding subgroups. Notably, the first female subgroup identified by MGFA exhibits significantly stronger correlations between A$\beta$ and cognitive test scores, compared to the second male subgroup identified by MGFA. Specifically, in the female subgroup, A$\beta$ shows significant associations with all cognitive test scores (MMSE, ADAS11, ADAS13, RAVLT percentage forgetting, RAVLT learning, RAVLT immediate), whereas in the male subgroup, A$\beta$ only correlates with ADAS11 and ADAS13. This sex difference in A$\beta$'s effects on cognitive aligns with previous findings (Koran et al., 2017; Li & Singh, 2014; Hirata-Fukae et al., 2008), which indicate that females are more vulnerable to the harmful effects of A$\beta$ accumulation than males. Furthermore, this sex difference may contribute to the higher prevalence of LMCI and AD in females compared to males (Au et al., 2017; Li & Singh, 2014), suggesting a critical need for sex-specific considerations in AD research and clinical practice.

## I.2. Supplementary to corpus callosal shape analysis

In this section, we present additional materials for corpus callosal shape analysis.

### I.2.1. IDs FOR VARIOUS BRAIN MEASURES

Table 8 summarizes various brain measures and their corresponding IDs used in Section 7. For further information, one may refer to https://adni.bitbucket.io/reference/ucsffsx51.html.

*Table 8.* IDs of various brain measures

| ID | Meaning |
|---|---|
| ST2SV | Subcortical Volume of CorpusCallosumAnterior |
| ST3SV | Subcortical Volume of CorpusCallosumCentral |
| ST4SV | Subcortical Volume of CorpusCallosumMidAnterior |
| ST5SV | Subcortical Volume of CorpusCallosumMidPosterior |
| ST6SV | Subcortical Volume of CorpusCallosumPosterior |
| ST103CV | Cortical Volume of RightParahippocampal |
| ST88SV | Subcortical Volume of RightHippocampus |
| ST44CV | Cortical Volume of LeftParahippocampal |
| ST29SV | Subcortical Volume of LeftHippocampus |
| ICV | Intracranial Volume |

### I.2.2. WITHIN-SUBGROUP CORRELATION HEATMAPS

In this section, we provide heatmaps of within-subgroup variable correlations. Figure 8 gives heatmaps for subgroups within the female cohort, while Figure 9 gives heatmaps for the male cohort. The inputs for these heatmaps are the FDR-adjusted p-values from the pairwise Pearson correlation test. Since the matrix is symmetric, we only visualize the lower-diagonal part of the matrix.

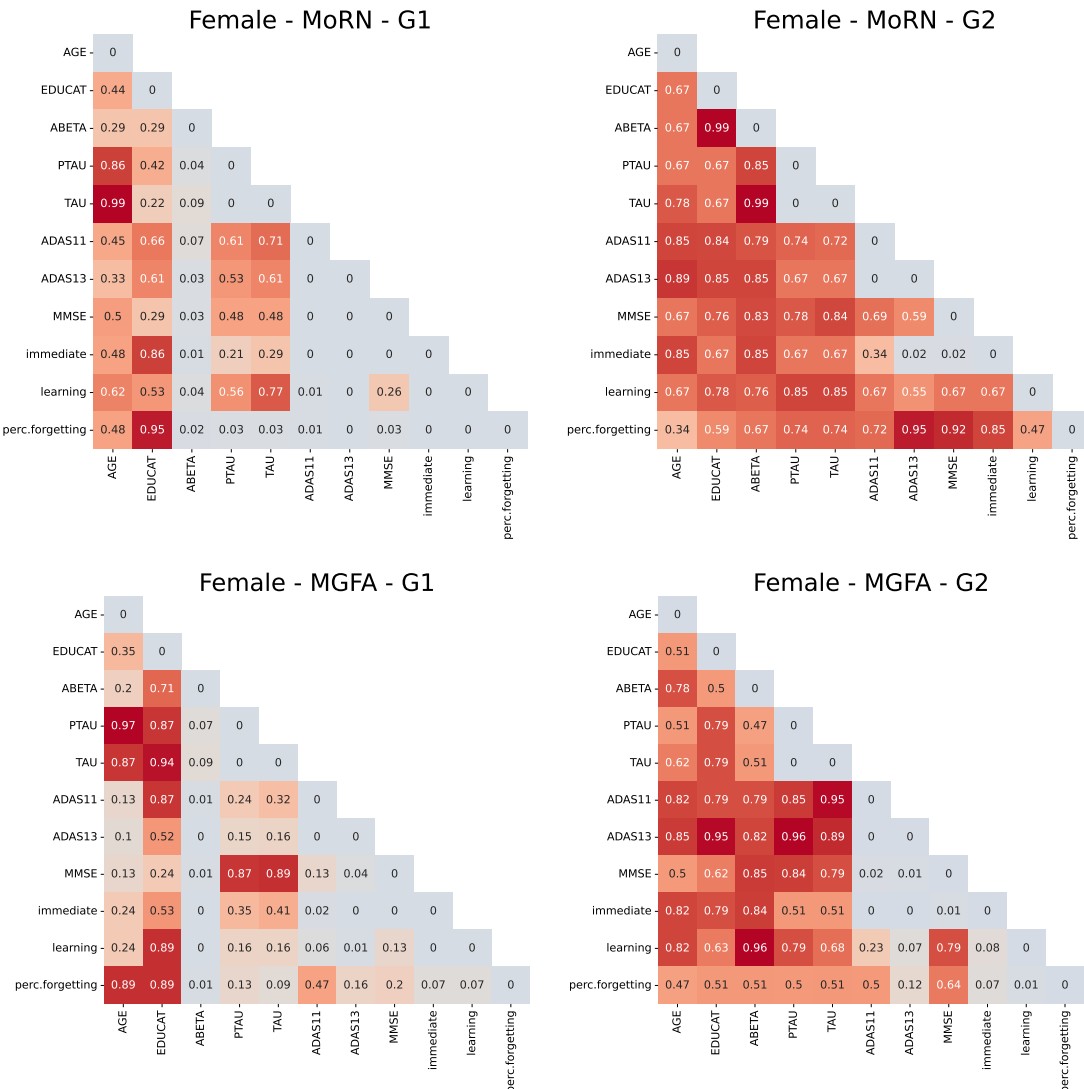

*Figure 8.* Within-subgroup correlation tests in the female cohort. The first row shows results for two MoRN-identified subgroups while the second row corresponds to two MGFA-identified subgroups. For each panel, we provide the heatmap of variable correlation tests, measured by FDR-adjusted p-values from the Pearson correlation test. The variables of interest include AGE, EDU, CSF (A$\beta$, TAU, PTAU), and cognition scores.

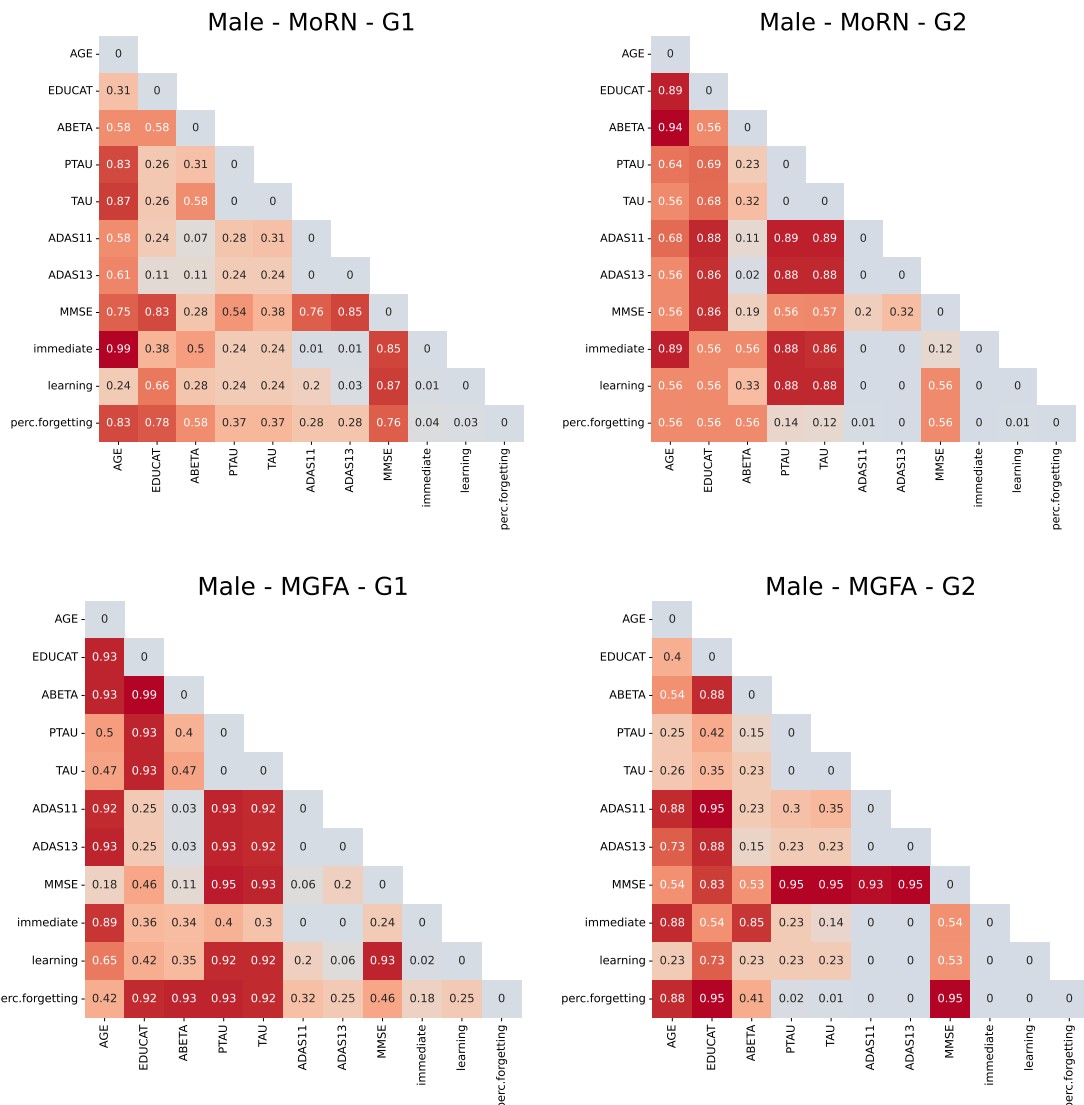

*Figure 9.* Within-subgroup correlation tests in the male cohort. The first row shows results for MoRN-identified subgroups while the second row corresponds to MGFA-identified subgroups. For each panel, we display the Pearson correlation tests in a heatmap, measured by FDR-adjusted p-values. The variables of interest include AGE, EDU, CSF (A$\beta$, TAU, PTAU), and cognitive scores.

## I.3. Additional results in left hippocampal shape analysis

This section presents additional results in left hippocampal shape analysis. First, Table 9 summarizes the demographic information of identified subgroups.

Next, we investigate the factor structures within these subgroups, as depicted in Figure 10. The analysis reveals distinct factor structures across groups. Notably, although G1 and G3 share similar mean shapes, their factor structures differ significantly, distinguishing them as separate clusters. Additionally, G2 shows a strong factor structure, indicating considerable variability in shapes across subjects within this group. Also, the factor pattern of G2 is different from its surface distance pattern in Figure 4, demonstrating that the reference shape does not belong to this group.

Moreover, we compare G1 and G2, which have the largest geodesic distance of 0.0659 and the highest and lowest MMSE scores, respectively. Figure 11 illustrates geodesic between their mean shapes, highlighting key structural differences. Notably, the most pronounced variations occur in the top and bottom regions, which may correspond to functionally distinct areas of the hippocampus.

*Table 9.* Demographic information and the mean variable values for each subgroup in the hippocampal shape analysis. The standard deviations of the variables are provided in the parentheses.

| Group | $n$ | Sex (F/M) | Age (years) | Education Length (years) | MMSE | Hipp / ICV (1e-4) | ICV (1e5) |
|-------|-----|-----------|-------------|--------------------------|------|-------------------|-----------|
| G1 | 145 | 58 / 87 | 73.91 (7.13) | 15.30 (3.15) | 27.17 (1.72) | 40.58 (7.12) | 15.69 (1.75) |
| G2 | 14 | 4 / 10 | 71.39 (7.23) | 14.85 (3.16) | 26.00 (1.47) | 40.21 (4.89) | 16.39 (1.50) |
| G3 | 79 | 27 / 52 | 74.27 (7.62) | 15.50 (2.77) | 26.80 (1.88) | 42.03 (7.03) | 15.48 (1.50) |
| G4 | 82 | 29 / 53 | 74.54 (7.56) | 16.02 (3.26) | 26.95 (1.86) | 40.40 (5.77) | 15.93 (1.77) |

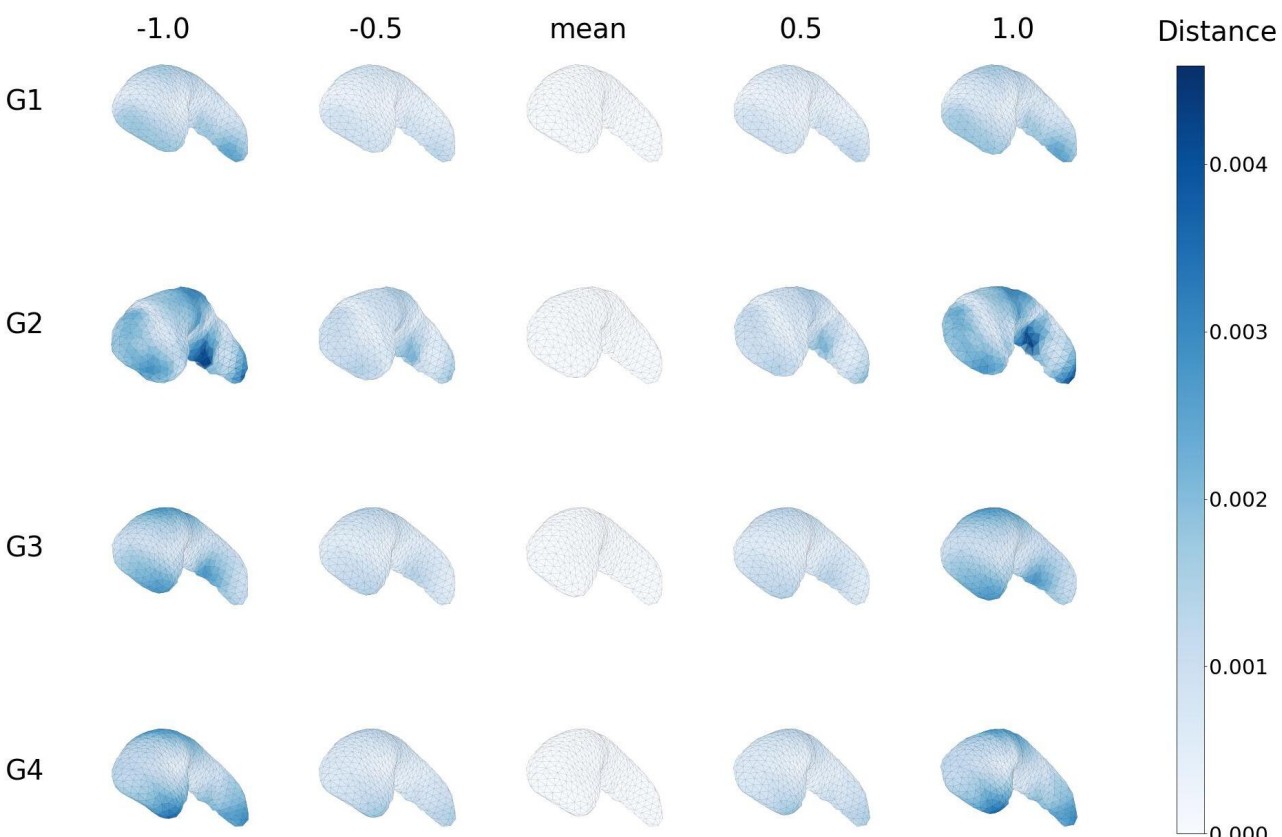

*Figure 11.* Visualization of the connecting geodesics between the mean shapes for group one and group two. These panels correspond to $\mathrm{Exp}_{\alpha_1}(r \cdot \mathrm{Log}_{\alpha_1}(\alpha_2))$ with $r \in [0, 1/4, 2/4, 3/4, 1]$, where $\alpha_1$ and $\alpha_2$ are the mean shapes of G1 and G2. The color gradient corresponds to the surface distance between the shape and the mean shape of G1. The geodesic distance between G1 and G2 is 0.0659.

*Figure 10.* Visualization of the factor structures for each subgroup. For each group, we depict $\mathrm{Exp}_{\alpha_k}(r \cdot v_k)$ with $r \in [-1, -1/2, 0, 1/2, 1]$, where $\alpha_k$ is the mean shape of the $k$-th group and $v_k \in T_{\alpha_k}\mathcal{M}$ is the estimated loading vector. The color gradient corresponds to the surface distance from each panel to its corresponding mean shape.

