## J. Discussion

In this section, we present several intriguing questions for future research.

- **Parameter estimation and identifiability:** Our paper establishes the convergence rate of the MLE in terms of the Hellinger distance. However, whether model parameters can be estimated at the same rate remains an open question. This challenge is closely linked to the identifiability of the MGFA model. In Euclidean spaces, previous works have established convergence rates for parameter estimation under strong identifiability conditions (Ho & Nguyen, 2016a;b; Heinrich & Kahn, 2018). Extending these results to Riemannian homogeneous spaces poses a compelling challenge.

- **Model selection:** Determining the appropriate number of clusters and latent factors in the MGFA model is critical. A key question is whether the clusters identified via heuristic model selection techniques correspond to genuine subgroups or are merely artifacts. In the Euclidean case, hypothesis-testing-based approaches have been proposed (Lo et al., 2001; Nylund et al., 2007; Liu et al., 2008). Extending these techniques to manifolds would be an interesting avenue for research.

- **Clustering high-dimensional manifold-valued data:** High-dimensional manifold clustering presents unique challenges, including computational efficiency and the curse of dimensionality. Developing an efficient algorithm for clustering large-scale, high-dimensional datasets is crucial. Additionally, exploring whether certain regularization techniques can mitigate the curse of dimensionality is an important direction.

- **Extending MGFA models:** Further exploration of MGFA models is promising. For instance, incorporating covariates (e.g., age or gender in medical imaging) into mixture models may provide new insights. Another interesting extension involves assuming shared parameters across clusters, such as a common variance parameter ($\sigma$), which could simplify model complexity and improve interpretability.