# OpenReview forum: "Mixtures of geodesic factor analyzers on Riemannian homogeneous spaces"
_ICML.cc/2026/Conference — ICML 2026 regular_

### Official Review · Reviewer_yo8x · 2026-03-06

**Soundness:** 3
**Presentation:** 3
**Significance:** 2
**Originality:** 3
**Overall Recommendation:** 5
**Confidence:** 3

**Summary:**

The paper introduces a new model for approximating distributions and clustering on Riemannian (homogeneous) manifolds. To this end, the authors combine a geodesic regression model for manifold-valued observations using Riemannian Gaussian noise with a mixture approach. They introduce a practical algorithm to estimate the parameters of the resulting mixture model on a wide array of manifolds. Furthermore, they provide theoretical insights into the convergence rates of the MLE estimator for their model.

**Compliance With Llm Reviewing Policy:**

Affirmed.

**Final Justification:**

After reading the rebuttal, my positive opinion about the paper remains. When incorporating the suggested extended discussion, I believe the paper could be a valuable contribution to the community. As raised also by the other authors, there are certainly some limitations, however, these are things that can be addressed in future work from my perspective.

**Key Questions For Authors:**

- Motivation: What are central challenges in clustering of manifold-valued data? Which specific applications can benefit from your approach? Should a practitioner care about your method, and if yes, why?
- Runtimes: What are the runtimes and resource needs of your method?

**Limitations:**

No issues with  potential negative societal impact, however, the authors do not discuss the limitations of their method at all (see also weaknesses in questions).

**Strengths And Weaknesses:**

Strengths:
- I found the paper easy to follow.
- The paper includes theoretical results showing a similar behavior to Euclidean spaces
- The method shows convincing and improved clustering results on synthetic data.
- The method yields plausible results on real data.

Weaknesses:
- The paper lacks in discussion of the method (the section "Discussion" is just a summary) and placement into the wider context.
- The real world evaluation is limited to one dataset, where the significance of the results is hard to judge.
- I would assume that the method is not easy to implement from scratch.

---

> ### Author Rebuttal · Authors · 2026-03-28
>
> We sincerely thank the reviewer for their encouraging feedback and for the meticulous evaluation of our manuscript. Here are our responses to the reviewer's questions.
>
> ## Question 1. Motivation: What are central challenges in clustering of manifold-valued data? Which specific applications can benefit from your approach? Should a practitioner care about your method, and if yes, why?
>
> The primary challenges in clustering manifold-valued data arise from its inherent **non-Euclidean geometry, structural heterogeneity, and directional anisotropy**. Addressing these complexities requires significant **innovation in statistical modeling, algorithmic design, and numerical optimization**. For instance, an effective algorithm must accommodate overlapping subgroups with anisotropic dispersion—a task where standard Expectation-Maximization (EM) approaches often falter due to the intricate nature of the likelihood landscape. Our proposed iterative framework successfully overcomes these hurdles. Furthermore, as distinct manifolds possess unique geometric properties, we **provide extensive mathematical derivations for their implementation, with detailed treatments of specific geometries provided in Appendices E.3, E.4, and E.5**. By leveraging advanced Riemannian geometric tools, we establish a unified clustering framework with formal theoretical guarantees, providing a clear and rigorous methodology for researchers in the field.
>
> Our approach is uniquely suited to identifying anisotropic subgroups in complex biological data. In neuroimaging, for example, **longitudinal cohorts of patients and healthy controls often exhibit distinct clustering patterns, where individual subgroups possess latent structures influenced by aging processes.** By accounting for these anisotropic variations, our method provides a more nuanced lens for clinical research, potentially revealing insights into disease progression and age-related brain morphology that traditional isotropic models might overlook.
>
> We highly recommend that practitioners account for the inherent complexities of manifold-valued data, as real-world datasets seldom conform to simple unimodal or isotropic assumptions. By providing a framework that explicitly models multimodality and anisotropic structures, our method offers a more robust and accurate alternative for complex geometric data analysis.
>
> ## Question 2. Runtimes: What are the runtimes and resource needs of your method?
>
> While implementing mixtures of isotropic distributions (equivalent to MGFA without latent structures) is highly efficient, the full MGFA model—which incorporates latent factor structures—entails an order-of-magnitude increase in computational complexity. However, **the framework remains computationally tractable; for datasets of reasonable scale, the algorithm can be executed efficiently on a standard laptop,** demonstrating its practical viability despite the added model depth.

---

> > ### Author Rebuttal · Reviewer_yo8x · 2026-04-03
> >
> > I was already quite positive in my review, so I appreciate the authors time to address my questions. I suggest a clearer discussion of the method (as long with placement in context, see e.g. the points raised by reviewer Whjg) and its motivation should be added to the paper as well.

---

> > > ### Author Response · Authors · 2026-04-05
> > >
> > > Thank you for your thoughtful follow-up and positive assessment. We are pleased that our responses have satisfactorily addressed your concerns. We will revise the paper to include a clearer explanation of the method and its motivation.

---

### Official Review · Reviewer_N4bF · 2026-03-06

**Soundness:** 3
**Presentation:** 2
**Significance:** 3
**Originality:** 3
**Overall Recommendation:** 4
**Confidence:** 4

**Summary:**

The paper proposes Mixtures of Geodesic Factor Analyzers (MGFA) that generalizes factor analysis onto Riemannian homogeneous space for clustering manifold-valued data. The proposed model introduces a latent geodesic factor structure within each mixture component so that clusters have anisotropic variation, rather than the isotropic behavior of standard Riemannian mixtures. The paper also gives statistical theory in forms of entropy bounds and Hellinger-based rate guarantees for the constrained MLE where mixtures of radial distributions can be viewed as a special case with zero latent dimension. It also presents an alternating algorithm with implementations on popular manifolds like spehres, CP shape spaces, and hyperbolic spaces. The method is evaluated on synthetic and neuroimaging data, the latter of which includes well-known examples of corpus callosum and hippocampal shape analyses.

**Compliance With Llm Reviewing Policy:**

Affirmed.

**Final Justification:**

I believe this paper is worth acceptance. The authors have successfully addressed my (and others, at least from what I read) concerns successfully.

**Key Questions For Authors:**

- Relationship between the algorithm and the theoretical estimator: The theoretical analysis studies a constrained maximum likelihood estimator, while the algorithm used in practice appears to be a hard-assignment alternating procedure involving Frechet means, tangent-space PCA, and Monte Carlo likelihood approximation. Could the authors clarify the precise relationship between the algorithm and the estimator analyzed in the theory? For example, does the algorithm optimize the same objective, or is it intended as a practical approximation to the MLE?
- Clarification of theoretical guarantees: The paper discusses $\sqrt{n}$ convergence results for the MLE, but the theoretical statements appear to focus on convergence of density estimates in Hellinger distance rather than parameter estimation. Could the authors clarify the exact interpretation of the theoretical guarantees and how they relate to parameter estimation and clustering performance?
- Initialization and model-selection procedures: The manuscript appears to describe different criteria for selecting the best run or model (e.g., likelihood vs. residual error). Could the authors clarify which criterion is actually used in the experiments and provide more details on the practical model selection process for choosing $K$ and $q$?
- Evaluation against stronger baselines:  The experiments compare MGFA with several clustering approaches, but the baselines are relatively limited. Could the authors comment on how the proposed method compares with other anisotropic manifold clustering approaches or tangent-space factor-analysis–based methods?
- Stability and robustness of discovered clusters in real data:  For the neuroimaging applications, could the authors provide additional evidence on the stability of the discovered clusters contingent on random initializations, bootstrap samples, or different hyperparameter choices? Such analysis would help assess the robustness of the subgroup findings.

**Limitations:**

In my opinion, the discussion of limitations may not be adequate for a methodology paper when the main discipline it seeks its adoption is neuroimaing. I would like the paper to discuss at least cluster instability and spurious subgroup discovery, demographic/site confounding, the risk of over-interpreting unsupervised subgroups as clinically meaningful, and the fact that these clusters should not be used for medical decision-making without external validation.

**Strengths And Weaknesses:**

**Strengths**
- The paper addresses a foundational problem of clustering heterogeneous manifold-valued data where subpopulations may exhibit anisotropic variation, the direction where standard models cannot capture effectively.
- The modeling contribution is appealing. Specifically, extending mixtures of factor analyzers to Riemannian homogeneous spaces through geodesic factor models provides a natural geometric analogue of classical latent-variable models.
- This paper attempts to present both methodological innovation with accompanying theoretical guarantees.
- The theory part incorporates core geometric tools, e.g., Jacobi fields and comparison theorems, to control model complexity on manifolds. This is technically interesting and I expect it to have borader relevance beyond this work.
- Riemannian homogeneous space is a relatively general setting, covering a wide range of important manifolds that practitioners often find relevant.
- Both synthetic experiments and real-data applications are contined, which is a definite plus. Also, the authors attempt to evaluate robustness under model misspecification.

**Weaknesses**
- There appears to be a gap between the theoretical estimator analyzed in the paper (a constrained MLE) and the practical algorithm used for estimation, which is an alternating procedure with hard assignments and Monte Carlo approximations.
- The theoretical guarantees focus on density estimation in Hellinger distance rather than parameter estimation, and the connection between the theoretical results and practical estimation objectives is not fully clarified.
- Some methodological descriptions are inconsistent across sections, particularly regarding initialization procedures and criteria for selecting the best model fit.
- Model selection for the number of clusters and latent dimensions relies on heuristic approaches (e.g., elbow method on residual errors), which may limit reproducibility and statistical justification.
- The comparison baselines are somewhat limited and do not include some potentially competitive anisotropic manifold clustering approaches. For instance, one may consider tangential-based clustering easily with some arbitrary reference as a base point, the procedure of which often delivers good performance.

---

> ### Author Rebuttal · Authors · 2026-03-28
>
> We appreciate the reviewer’s thorough reading and the valuable effort invested in this evaluation. We have incorporated the suggested improvements and addressed all specific comments. Our detailed responses follow below.
>
> ## Question 1. Relationship between the algorithm and the theoretical estimator.
>
> In the statistical theory, we assume the estimator is the **global maximum likelihood estimator**. But in the algorithm, we did not use the likelihood as the objective function and directly optimize it to find the global solution. This is due to the non-convexity and extreme complexity of the likelihood function, which involves both integration on manifolds, i.e., the normalizing constant $Z(\sigma)$, and integration of latent variables. To address this challenge, we adopt a heuristic approach to approximately estimate the MLE, similar to the classic Expectation-Maximization algorithm. Our simulation studies have demonstrated its effectiveness.
>
> ## Question 2. Clarification of theoretical guarantees.
>
> We thank the reviewer for this observation. Our current theory provides convergence guarantees quantified by the Hellinger distance between the estimated and true distributions. Although this confirms the statistical consistency of the MGFA model, we recognize that deriving explicit bounds for parameter convergence would require additional identifiability assumptions and proofs. We have noted this distinction in the revised manuscript and view it as a valuable path for extending our theoretical analysis.
>
> ## Question 3. Initialization and model-selection procedures.
>
> In our current experiments, we use the residual error to select the best model configuration ($K$ and $q$). This is a special example of likelihood-based model selection method under the assumption that all subgroups have the same dispersion parameters $\sigma_k=\sigma$.
>
> ## Question 4. Evaluation against stronger baselines:
>
> Thanks to Reviewer Whjg, we find the paper on *the mixture-of-PPGA model by Zhang, Xing, and Zhang, MMBIA 2019* is relevant to ours. However, as we explained in the response to Reviewer Whjg, the implementation of mixture-of-PPGA exhibits a fundamental theoretical inconsistency. Specifically, they treat $x_{nk}$ as the latent variable in equation (5); but, Section 3.1 treats $x_{nk}$ as a parameter to be estimated within $\theta$. By treating only the assignment variable $z_{nk}$ as the latent variable, the approach departs from standard EM rigor. This stands in contrast to the original PPGA paper, where $x$ was correctly treated as a latent variable requiring Hamiltonian Monte Carlo for estimation. Consequently, the mixture-of-PPGA algorithm lacks a rigorous theoretical foundation.
>
> Moreover, **the simulations presented in the mixture-of-PPGA study fail to adequately evaluate the model's performance in complex scenarios, as the tested subgroups remain non-overlapping.** As shown in Figure 3(c) in their paper, the disjoint nature of the actual clusters simplifies the mixture structure. This allows for clustering and estimation to be treated as decoupled tasks.
>
> **In contrast, our simulations incorporate significantly overlapping subgroups** (see Figure 2 in our paper), representing a more rigorous benchmark for latent structure recovery. Our proposed method is uniquely capable of identifying these overlapping clusters. Moreover, during our initial framework development, we evaluated an EM-based approach similar to that used in the PPGA studies; however, it failed to recover the true latent structures, instead collapsing into solutions resembling mixtures of isotropic distributions. **This motivated us to propose a heuristic yet effective approach to implement the MGFA, that is, using tangent space PCA in an iterative algorithm.**
>
> ## Question 5. Stability and robustness of discovered clusters in real data & Limitations
>
> We thank the reviewer for this valuable suggestion regarding model stability and clinical interpretation. In our current experimental setup, we observed that the identified subgroups remain relatively stable. To further validate these findings, we intend to systematically evaluate stability across more diverse conditions, including the use of bootstrap resampling to quantify uncertainty. Additionally, we have incorporated a detailed discussion of the method’s limitations into the revised manuscript, as suggested.

---

> > ### Author Rebuttal · Reviewer_N4bF · 2026-04-03
> >
> > Thanks for the rebuttal and I acknowledge that the authors seem to have managed their responses well. I'll keep the score as is.

---

> > > ### Author Response · Authors · 2026-04-05
> > >
> > > Thank you for your acknowledgment and for your careful consideration of our rebuttal. We are very glad that our responses were able to address your concerns. We appreciate your thoughtful evaluation, and we are grateful for your continued consideration of the paper in light of these clarifications.

---

### Official Review · Reviewer_zNa4 · 2026-03-12

**Soundness:** 3
**Presentation:** 3
**Significance:** 2
**Originality:** 2
**Overall Recommendation:** 4
**Confidence:** 3

**Summary:**

This paper introduces a geodesic factor model and extend it to clustering via Mixtures of Geodesic Factor Analyzers (MGFA) on Riemannian homogeneous spaces. It derives entropy bounds for MGFA and extend the analysis to mixtures of Riemannian radial distributions. It also proposes an iterative procedure for MGFA estimation and clustering. MGFA is evaluated on spheres, shape spaces, and hyperbolic spaces for low- and high-dimensional regimes, multiple classes, and model misspecification.

**Compliance With Llm Reviewing Policy:**

Affirmed.

**Final Justification:**

Some of my major concerns regarding the range of applicability of the proposed method have been resolved.

**Key Questions For Authors:**

- It seems like the focus of the experiments is Riemannian manifolds with constant curvature which greatly simplifies the theoretical analysis and computational aspect in real wolrd applications. Are there any challenges for the proposed method to be applied to other Riemannian manifolds like SPD ?
- Related to the previous question, the iterative procedure for MGFA estimation and clustering includes Fréchet mean computation which is generally expensive. Moreover, the Fréchet mean is not easily obtained on several Riemannian homogeneous spaces (see examples in Helgason, 1979). How does the proposed procedure cope with that ?
- Simulations on hyperbolic spaces are designed for relatively low-dimensional regime ($\mathbb{H}^{10}$). Does the proposed method show a similar behavior in higher dimensional hyperbolic spaces ?

**Limitations:**

I did not find a discussion on the limitations of their work. Please see the weaknesses and the questions for suggestions for improvement.

**Strengths And Weaknesses:**

Strengths:
- The paper is well-written
- The paper provides entropy estimates for MGFA and a root-n non-asymptotic convergence rate for the maximum likelihood estimator (MLE)
- Experimental results demonstrate the effectiveness of the proposed method


Weaknesses:
- It is not clear how effective the proposed model is for more complicated Riemannian homogeneous spaces like SPD manifolds
- MGFA estimation relies on Fréchet mean computation which is generally expensive and may not be easily implemented on several Riemannian homogeneous spaces.

---

> ### Author Rebuttal · Authors · 2026-03-28
>
> We thank the reviewer for their careful reading and the constructive feedback provided in this comprehensive review. Our point-by-point responses to the specific comments are detailed below.
>
> ## Question 1. It seems like the focus of the experiments is Riemannian manifolds with constant curvature which greatly simplifies the theoretical analysis and computational aspect in real wolrd applications. Are there any challenges for the proposed method to be applied to other Riemannian manifolds like SPD ?
>
> We address this point from three distinct perspectives:
>
> **First,** the scope of our framework extends beyond manifolds of constant curvature (such as spheres and hyperbolic spaces) to **encompass Riemannian manifolds with non-constant curvature**. Specifically, our analysis includes shape spaces, which correspond to complex projective spaces. This structural diversity ensures that our method remains applicable to a broad range of real-world geometries encountered in practical applications.
>
> **Second,** our algorithmic framework is directly applicable to **Symmetric Positive Definite (SPD) matrices**, provided the normalization constant $Z(\sigma)$ is properly addressed. As shown in Equation (3.8), the estimation step of MGFA requires calculating both the normalization constant $Z(\sigma)$ and its derivative $Z'(\sigma)$. While the complex closed-form formulation provided in Said et al. (2017) presents a non-trivial implementation challenge, its existence confirms the **mathematical feasibility** of our approach for SPD manifold data.
>
> **Third,** while our theoretical results apply to the general class of **Riemannian homogeneous spaces**, we acknowledge that the algorithm requires manifold-specific implementations of the dispersion constant. We provide a rigorous treatment of this challenge in **Appendix E.3**, using complex projective spaces as a non-trivial case study. This example illustrates the unique geometric complexities inherent to different manifolds, which preclude a universal implementation. Nevertheless, our current work **prioritizes the most computationally relevant and widely used manifold types**, providing a robust foundation for further extensions.
>
> ## Question 2. Related to the previous question, the iterative procedure for MGFA estimation and clustering includes Fréchet mean computation which is generally expensive. Moreover, the Fréchet mean is not easily obtained on several Riemannian homogeneous spaces (see examples in Helgason, 1979). How does the proposed procedure cope with that ?
>
> **First, regarding computational efficiency and implementation**: For general Riemannian manifolds, we recommend the **Riemannian gradient-based method** for Fréchet mean estimation. While we acknowledge the associated computational overhead, this remains the standard approach for ensuring convergence on non-Euclidean structures. However, for computationally demanding applications on specific manifolds like spheres, we propose a more efficient **one-step heuristic estimator**: computing the Euclidean mean followed by a projection onto the manifold. This significantly reduces the per-iteration cost while maintaining sufficient accuracy for clustering purposes.
>
> **Second, regarding the necessity of the Fréchet mean in latent variable models**: It is important to note that the requirement for Fréchet mean computation is not unique to MGFA; it is a fundamental necessity for even simpler frameworks, such as mixtures of isotropic distributions on manifolds. Since MGFA generalizes these isotropic models to account for complex latent factor structures, the inclusion of the Fréchet mean is thus a methodological necessity.
>
> ## Question 3. Simulations on hyperbolic spaces are designed for relatively low-dimensional regime (${\mathcal H}^{10}$). Does the proposed method show a similar behavior in higher dimensional hyperbolic spaces ?
>
> **First, regarding scalability**: In high-dimensional regimes, intrinsic models on hyperbolic spaces often encounter **numerical instability** due to **floating-point precision limits** (e.g., the rapid growth of the hyperbolic volume element). To mitigate this, we have incorporated specific numerical stabilizing transformations within our iterative procedure. While these methods significantly extend the stable range of the algorithm, further specialized precision-handling may be required for extreme-dimensional scenarios.
>
> **Second, regarding practical utility**: We emphasize that even low-dimensional hyperbolic spaces (such as 2D or 5D) possess immense representational capacity. Due to the exponential growth of volume in hyperbolic geometry, low-dimensional embeddings are often sufficient to represent complex hierarchical structures, such as large-scale trees or networks, with minimal distortion. Consequently, while our model scales, lower-dimensional hyperbolic spaces typically provide the most efficient and expressive embedding environment for real-world applications.

---

> > ### Author Rebuttal · Reviewer_zNa4 · 2026-04-03
> >
> > I would like to thank the authors for their rebuttal. However, some of my major concerns in questions 1, 2, and 3 still remain.
> > - Question 1:  The authors state that "its existence confirms the mathematical feasibility of our approach for SPD manifold data". I am not convinced by this argument. If the formula of the normalization constant exists, but it is challenging to compute it or to obtain a good approximation of it, then there is no guarantee that it can be used in practical applications.
> >
> > - Question 2: Riemannian gradient-based method for Fréchet mean estimation is only applicable when closed-forms of some geometric maps (exponential or retraction map) exist. This is not the case for several Riemannian homogeneous spaces. So in practice, it seems that the proposed method can only be applied to some specific Riemannian homogeneous spaces. Based on the responses for questions 1 and 2, I can't quite see how the range of applicability of the proposed method can go beyond the considered context, i.e., spaces of constant curvature. Since the authors claim that the method is designed for Riemannian homogeneous spaces, I think it is important to demonstrate for at least one case of non-constant curvature spaces.
> >
> > - Question 3: I understand that some methods can be very effective in low-dimensional regime, but if they show their best performance in 2D or 5D, then they might not be able to fully capture the underlying structures of geometric spaces in many real-world problems.

---

> > > ### Author Response · Authors · 2026-04-04
> > >
> > > Thanks for these follow-up questions. We would like to address all of them in greater detail.
> > >
> > > 1. For question 1, we now provide three additional lines of evidence, demonstrating the practicality of MGFA in the applications of SPD matrix spaces.
> > >
> > >     **(1) A practical estimation method for the dispersion parameter $\sigma$ has been provided in the paragraph following Proposition 7** in the paper Said et al. (2017). Specifically, the estimation problem is to find the root of a non-linear equation in one real-valued variable, i.e., the dispersion parameter $\sigma$. Said et al. (2017) proposed to use the standard Newton's method to estimate the dispersion parameter $\sigma$. With this method, we can implement our MGFA efficiently.
> > >
> > >     **(2) The approximation of the normalizing constant has been addressed in the paragraph with equation (21) in Said et. al (2017) .** Specifically, when the dimension is lower than 50, the normalizing constant has been easily evaluated using Monte Carlo integration. When the dimension is 2, the normalizing constant has an analytic expression.
> > >
> > >     **(3) Said et al. (2017) has implemented the mixtures of Riemannian Gaussian distributions on SPD matrix spaces. Extending their algorithm to our MGFA algorithm is fully plausible.** Notably, the underlying model estimation strategies remain compatible. We can use a similar model estimation strategy, such as the method in point (1). For the assignment, we can use Monte Carlo integration. Combing these two steps, we can implement MGFA in SPD matrix spaces.
> > >
> > >     - Said et al. (2017) Riemannian Gaussian Distributions on the Space of Symmetric Positive Definite Matrices.
> > >
> > > 2. We have addressed Question 2 in our paper. Specifically, **the shape space**, also known as the **complex projective space $\mathbb{CP}^{n}$,** **is a manifold of non-constant curvature. We have already implemented MGFA on this manifold  of non-constant curvature in our submitted manuscript with real-world data applications.**
> > >
> > > 3. Regarding Question 3, our framework is highly scalable for spherical and complex projective spaces, maintaining effectiveness in hundreds of dimensions. Only for hyperbolic spaces, the current implementation is restricted to lower-dimensional settings due to numerical instabilities. This is already useful in applications of hyperbolic embeddings. For example, **the expeirments in Sala et al. (2018) use 2-dimensional hyperbolic embeddings. The experiments in Nickle (2017) embed WorldNet datasets in a 2-dimensional hyperbolic space. Both papers have hundreds of citations, demonstrate that the low-dimensional hyperbolic embedding is one of the mainstream embedding method. This is because a tree can be embedded into a 2-dimensional hyperbolic space with arbitrarily low distortion (Sarkar, 2011).**
> > >
> > >     Furthermore, if the numerical stability can be addressed in future works, then our method is applicable to higher-dimensional hyperbolic spaces.
> > >
> > >     - Sala et. al (2018). Representation tradeoffs for hyperbolic embeddings.
> > >
> > >     - Nickel et. al (2017) Poincaré embeddings for learning hierarchical representations
> > >
> > >     - Sarkar, R (2011). Low distortion Delaunay embedding of trees in hyperbolic plane.
> > >
> > > We appreciate your thoughtful evaluation, and we are grateful for your continued consideration of the paper in light of these clarifications.

---

### Official Review · Reviewer_Whjg · 2026-03-13

**Soundness:** 4
**Presentation:** 3
**Significance:** 3
**Originality:** 3
**Overall Recommendation:** 5
**Confidence:** 5

**Summary:**

This paper presents a mixture of geodesic factor analyzers (MGFA) model that generalizes mixtures of linear factor analysis from Euclidean space to Riemannian manifolds (particularly homogeneous spaces). The authors present an iterative algorithm for estimating model parameters. There are two main theoretical results: one regarding the model complexity through entropy bounds and one regarding the convergence rate. Finally, the experiments demonstrate the model fit on synthetic manifold data and 2D and 3D shapes from brain MRI (these are modeled as the manifold that is the "preshape" sphere, where translation and scale are removed.)

**Compliance With Llm Reviewing Policy:**

Affirmed.

**Final Justification:**

The proposed MGFA model is rigorously developed and evaluated. The model formulation is equivalent to the previously proposed mixture-of-PPGA model, but the proposed MGFA has a more rigorous theoretical and algorithmic development.

**Key Questions For Authors:**

The key question is the relationship to PPGA/mixture-PPGA described above. The authors should thoroughly discuss the relationship of these models with their proposed MGFA and explain how it is different. I believe the theoretical results of the paper are strong enough to still be publishable even if the MGFA model is equivalent to a mixture-of-PPGA.

How does the proposed estimation procedure compare to the expectation-maximization algorithm that is used in PPGA? Are there advantages or disadvantages to either approach? And, as asked above, is the MGFA estimation algorithm guaranteed to converge to the MLE?

Can the authors bring at least some highlights of real data results into the main body of the paper?

**Limitations:**

I didn't see a discussion of limitations. (I don't see a need for a discussion of negative societal impact).

**Strengths And Weaknesses:**

Strengths:
* The MGFA model is rigorously based on an appropriate generalization of the normal distribution to Riemannian homogeneous spaces. The hierarchical modeling with latent mixtures and factors is well developed and nicely reflects the analogous model in Euclidean space.
* The theoretical contributions are the strongest aspect of the work. The entropy bracket on the model complexity and the large sample convergence are both excellent results. The fact that they behave similarly to the analogous results in Euclidean space is a bit surprising at first glance.
* The MFGA model performs favorably compared to competing approaches (Ward, mixture of von Mises, spherical k-means, and mixture of Riemannian normals). Although, this is not too surprising, as none of the competing models would handle geodesic factors.

Weaknesses:
* The biggest weakness is the similarity of the proposed factor analysis to probabilistic principal geodesic analysis (PPGA) by Zhang and Fletcher, NeurIPS 2013, and the mixture-of-PPGA model by Zhang, Xing, and Zhang, MMBIA 2019. The formulation of the PPGA latent geodesic factors looks equivalent to me to the proposed MGFA (with slightly different, but equivalent, parameterizations). These papers should be cited and the similarities and differences with the proposed approach should be discussed. It does seem like the estimation procedures are different, and the PPGA/mixture-PPGA papers do not develop the complexity and convergence theory that is in the present paper.
* The PPGA paper develops an expectation-maximization algorithm for parameter estimation, which appears to avoid a projection step that is needed in the MGFA estimation algorithm. I would expect that projection step and linear PCA (step (b) in Section 3.2.1) is a suboptimal approximation. Is this iterative procedure guaranteed to converge to the true MLE? It seems that this is a linear approximation, similar to how tangent space PCA approximates PGA.
* The real data results are almost an afterthought in the main text, and they are mostly relegated to the appendix. The paper is 52 pages with appendices, and I don't think it is reasonable that this should all be required reading for a conference review process. Thus, to make the paper self-contained, more of the real data experiments should be promoted to the main text -- perhaps just the tables that highlight the main comparison metrics? (I do think that relegating the background on Riemannian geometry and the detailed theorem proofs to the appendix is appropriate.)

---

> ### Author Rebuttal · Authors · 2026-03-27
>
> We would like to express our gratitude for the time and effort invested in providing such a comprehensive review. Here are our responses.
>
> ## Question 1. Similarity with PPGA and mixture-of-PPGA & Our contributions.
>
> We thank the reviewer for bringing these two closely related studies to our attention. After a thorough comparative analysis, we acknowledge that the underlying MGFA model is equivalent to the mixture-of-PPGA model. However, we maintain that **this does not diminish the primary contributions of our manuscript. Our work provides distinct algorithmic and theoretical advancements that remain unaddressed in prior literature**, as detailed in the following points:
>
> (1) **First, the algorithm in the mixture-of-PPGA study departs from the standard EM framework and exhibits a fundamental theoretical inconsistency.** Specifically, they treat $x_{nk}$ as the latent variable in equation (5); but, Section 3.1 treats $x_{nk}$ as a parameter to be estimated within $\theta$. By treating only the assignment variable $z_{nk}$ as the latent variable, the approach departs from standard EM rigor. This stands in contrast to the original PPGA paper, where $x$ was correctly treated as a latent variable requiring Hamiltonian Monte Carlo for estimation. Consequently, the mixture-of-PPGA algorithm lacks a rigorous theoretical foundation.
>
> (2) **The simulations presented in the mixture-of-PPGA study fail to adequately evaluate the model's performance in complex scenarios, as the tested subgroups remain non-overlapping.** Specifically, as shown in Figure 3(c) in their paper, the disjoint nature of the actual clusters simplifies the mixture structure. This allows for clustering and estimation to be treated as decoupled tasks.
>
> **In contrast, our simulations incorporate significantly overlapping subgroups** (see Figure 2 in our paper), representing a more rigorous benchmark for latent structure recovery. Our proposed method is uniquely capable of identifying these overlapping clusters. Moreover, during our initial framework development, we evaluated an EM-based approach similar to that used in the PPGA studies; however, it failed to recover the true latent structures, instead collapsing into solutions resembling mixtures of isotropic distributions. **This motivated us to propose a heuristic yet effective approach to implement the MGFA, that is, using tangent space PCA in an iterative algorithm.**
>
> Based on our previous experimental experience, we expect the mixture-of-PPGA algorithm to encounter similar limitations. Their current simulation results, which rely on disjoint subgroups, do not demonstrate the robustness required to handle the complex, non-separable settings addressed in our work.
>
> (3) **The theoretical contributions of this work are both novel and significant.** As noted by the reviewer, our manuscript extends beyond algorithmic development to establish a rigorous theoretical framework for model complexity and convergence rates. We introduce specialized techniques—including **comparison theorems and Jacobi field analysis from Riemannian geometry**—that have been largely overlooked in previous manifold data studies. By leveraging these geometric tools, we provide a deeper analytical understanding of manifold-based data structures, offering a robust methodology that we anticipate will serve as a foundation for future research in non-Euclidean data analysis.
>
> ## Question 2. The MGFA algorithm & the parameter estimation method.
>
> **During our initial framework development, we evaluated an EM-based approach similar to the PPGA studies; however, it failed to recover the true latent structures, consistently collapsing into suboptimal solutions resembling mixtures of isotropic distributions.** Consequently, we moved away from the standard EM/HMC framework in favor of a more robust iterative approach involving a projection step and tangent space PCA (TS-PCA). While this may appear as a heuristic simplification, our simulations demonstrate that **it is more effective at resolving complex latent structures than traditional EM.**
>
> Furthermore, by employing TS-PCA as a linear approximation of Principal Geodesic Analysis (PGA), we significantly reduce computational overhead without compromising accuracy. Regarding theoretical guarantees, while we do not provide a proof for global convergence to the Maximum Likelihood Estimator (MLE), this remains a fundamental challenge in non-convex optimization. Consistent with the standard EM algorithm—which is typically guaranteed only to reach a stationary point or local optimum—our method prioritizes practical stability and efficiency. We view the pursuit of global MLE convergence in manifold-based models as a significant direction for future research.
>
> ## Question 3. Insights of real data studies & paper organization.
>
> Thanks for the suggestion. We will move more contents of the real data studies to the main body of the paper.

---

> > ### Author Rebuttal · Reviewer_Whjg · 2026-04-07
> >
> > Thank you for the response. My concerns, particularly about the relationship to mixture-of-PPGA, has been resolved. I will raise my score with the assumption that this discussion will make it into a revised paper if accepted.

---

> > > ### Author Response · Authors · 2026-04-07
> > >
> > > Thank you for your acknowledgement and for carefully reading our rebuttal. We appreciate your thoughtful follow-up and positive feedback.
> > >
> > > We are glad that our clarifications helped resolve your concerns. We will ensure that these points, especially the discussion of the relationship to mixture-of-PPGA, are clearly incorporated into the revised paper.
> > >
> > > Thank you again for your constructive feedback throughout the review process. We also appreciate your encouraging reassessment of the paper.

---

### Decision · Program_Chairs · 2026-04-30

**Decision:**

Accept (regular)

**Comment:**

The paper proposes a Mixture of Geodesic Factor Analyzers (MGFA) model that generalizes mixtures of linear factor analysis from Euclidean space to Riemannian homogeneous spaces. An iterative algorithm for estimating model parameters is presented, along with theoretical results on model complexity through entropy bounds and convergence rate of the estimated density function. Numerical experiments are presented to demonstrate the effectiveness of the proposed approach.

Reviewers are generally positive about the theoretical contributions of the paper. Reviewer Whjg notes that the formulation of MGFA is equivalent to the previous work Mixture-of-PPGA (Zhang, Xing, and Zhang, MMBIA 2019), which the authors agree, but Reviewer Whjg also acknowledges that the current work is more rigorous theoretically. Reviewer zNa4 is concerned about the feasibility of the proposed method on more complicated manifolds such as SPD matrices, since the experiments were reported on manifolds with constant curvature.
Reviewer N4bF notes that the presented theoretical guarantees concern the convergence of the estimated density rather than parameter estimation. Reviewer yo8x notes that only one real word dataset was tested, thus the practical significance of the proposed method is not yet clear. Overall, reviewers agree that the current work is promising but there are many aspects for improvement.

The scores are Accept, Weak Accept, Weak Accept, Accept.